# Predictive validity in middle childhood of short tests of early childhood development used in large scale studies compared to the Bayley-III, the Family Care Indicators, height-for-age, and stunting: A longitudinal study in Bogota, Colombia

**Marta Rubio-Codina**[1]*, **Sally Grantham-McGregor**[2]

**1** Social Protection and Health Division, Inter-American Development Bank, Washington, D.C., United States of America, **2** Faculty of Population Health Sciences, Institute of Child Health, University College London, London, United Kingdom

* martarubio@iadb.org

## Abstract

There is increasing global commitment to establish early childhood interventions that promote the development of the millions of disadvantaged children in low- and middle-income countries not reaching their developmental potential. However, progress is hindered by the lack of valid developmental tests feasible for use at large scale. Consequently, there is an urgent need for such tests. Whilst screeners and single-domain tests ('short tests') are used as alternatives, their predictive validity in these circumstances is unknown. A longitudinal study in Bogota, Colombia began in 2011 when 1,311 children ages 6–42 months were given the Bayley Scales of Infant and Toddler Development (Bayley-III) by psychologists and randomized to receive one of two batteries of short tests under survey conditions. Concurrent validity of the short tests with the Bayley-III ('gold standard') was reported. In 2016, at 6–8 years, 940 of these children were given tests of IQ (Wechsler Intelligence Scale for Children, WISC-V) and school achievement (arithmetic, reading, and vocabulary) by psychologists. We compared the ability of the short tests, the Family Care Indicators (FCI), height-for-age, stunting (median height-for-age <-2 SD), and the Bayley-III to predict IQ and achievement in middle childhood. Predictive validity increased with age for all tests, and cognition and language were usually the highest scales. At 6–18 months, all tests had trivial predictive ability. Thereafter, the Bayley-III had the highest predictive validity, but the Denver Developmental Screening Test was the most feasible and valid short test and could be used with little validity loss compared with the Bayley-III. The MacArthur-Bates Communicative Development Inventory at 19–30 months and the FCI under 31 months predicted IQ and school achievement as well as the Bayley-III. The FCI had higher predictive validity than stunting and height-for-age, and could be added to stunting for use as a population-based indicator of child development.

**Data Availability Statement:** The analysis data are available to download from the IDB Data Portal following the IDB data documentation standards (https://data.iadb.org/DataCatalog/Dataset#DataCatalogID=sjty-9qzs). As the informed consent that study participants had signed before agreeing to participate in the study indicated that the data can only be used for research purposes, users will have to adhere to using the data for non-commercial research purposes.

**Funding:** Data collection was funded by Funds RG-T1907 and CO-T1419 from the Inter-American Development Bank (IDB). MRC is employed by the IDB. Hence, the funder (IDB) provided support in the form of salaries to author MRC but did not have any additional role in the study design, data collection and analysis, decision to publish, or preparation of the manuscript. SGM has no financial relationship relevant to this article to disclose. The opinions expressed in this publication are those of the authors and do not necessarily reflect the views of the IDB, its Board of Directors, or the countries they represent.

**Competing interests:** The authors have declared that no competing interests exist. The commercial affiliation of author MRC with the IDB does not alter the authors' adherence to PLOS ONE policies on sharing data and materials, which will be shared as indicated in the Data Availability Statement.

## Introduction

In low- and middle-income countries (LMICs), an estimated 250 million children under five years fail to reach their developmental potential [1]. With the recognition that critical building blocks for adult health and well-being are established early in life, there is an increasing global commitment to implement large scale early childhood development (ECD) interventions to address the problem and to promote the development of disadvantaged children [1]. However, progress of these efforts is impeded by the lack of reliable, valid, and easy-to-collect measures of ECD, particularly for children under age three years [1–3]. Such measures are essential both to monitor and evaluate the effectiveness of interventions, as well as to assess ECD levels at the population level.

Efforts are currently underway to develop global (i.e. culturally-neutral or very easy-to-adapt), valid, feasible, freely accessible population-based instruments, as well as individual-level instruments suitable to evaluate interventions, for children 0–3 years [3,4]. Whilst existing full developmental tests such as the Bayley Scales [5,6] are sensitive to ECD interventions [2,7–9], according to our experience and that of many other researchers, they are very expensive, take a long time to administer and require highly trained testers with a certain level of technical expertise [2]. These aspects make their use at scale very difficult [10]. Screener tests (which are designed to identify children at risk of developmental delay) or tests assessing one particular domain (language, for example) are more and more often used as alternatives, both for large scale surveys [11] and program evaluations [12,13], since they are cheaper, quicker, and much easier to administer. Importantly, they are readily available. However, their reliability and validity when used at large scale to detect differences in developmental levels within the normal range and/or to track developmental progress at the population level, rather than to screen for high-risk children, is unknown and must be determined [2]. More generally, it is critical to identify readily available, reliable, valid and feasible ECD measures for use in large samples until the population-based and individual-level instruments currently under development become available.

We previously assessed five tests commonly used in large surveys and evaluations for reliability, feasibility and concurrent validity with the Bayley Scales of Infant and Toddler Development (third edition, Bayley-III) [6] in children aged 6–42 months in Bogota, Colombia [10]. The tests ('short tests' henceforth) included three multi-dimensional screeners—the Ages and Stages Questionnaires (third edition, ASQ-3) [14], the Denver Developmental Screening Test (second edition, Denver-II) [15,16], the Battelle Developmental Inventory screener (second edition, BDI-2) [17]—and two single-domain tests—the World Health Organization Gross Motor Milestones (WHO-Motor) [18,19] and the MacArthur-Bates Communicative Development Inventories Short-Forms I and II (SFI, SFII) [20,21]. The short tests were given in the home by fieldworkers, who were sufficiently trained but had no specific background on ECD. Therefore, they were administered under conditions feasible to implement in large scale studies. In contrast, the Bayley-III, which we considered our 'gold standard', was administered by psychologists at a center to minimize distractions and standardize the test experience as far as possible. It was therefore administered under preferable conditions.

The Bayley-III was found to be the most expensive and longest test to give. Whilst also long and expensive, the BDI-2 took, on average, 20 minutes less to administer than the Bayley-III; and the Denver-II and the ASQ-3 took a third of the time or less, being intermediate in time and cost. The single-domain tests were quickest, taking no more than 8 minutes, on average, and were considerably less expensive (the WHO-Motor was free). Concurrent validity of their cognitive, language, and fine motor scales with matching (i.e. same domain) Bayley-III scales increased with age: correlations were low under 19 months, low-to-moderate at 19–30 months,

and moderate-to-high over 30 months. Whilst the ASQ-3 had a poor performance for children younger than 31 months, the SFII (expressive language by caregiver reports) correlated relatively well with the Bayley-III language scale under 30 months. Combining feasibility and validity considerations, the Denver-II was found to be the preferred multi-dimensional short test [10].

Before choosing a test, it is essential to know its ability to predict future function [22]. Therefore, we reassessed the children 5.5 years later in order to determine the predictive validity of the short tests with IQ, as assessed on the Wechsler Intelligence Scale for Children (fifth edition, WISC-V) [23], and with an index of school achievement, which combined scores on arithmetic, reading, and vocabulary, at ages 6–8 years, compared with that of the Bayley-III. We also examined the predictive ability of later functioning of children's height-for-age and stunting (height-for-age <-2 standard deviations (SD) of the WHO reference median) in early childhood, since the latter has been repeatedly used as a global ECD proxy indicator [1,24]; and of the Family Care Indicators (FCI), a measure of the home environment quality that has been widely assessed in international surveys as a relevant protective factor of ECD [25–27]. As with concurrent validity [10], and given the broad age range of the children in the study sample (36 months) and fast pace of development from 6 to 42 months, the analysis of predictive validity was performed by 12-months-of-age intervals (6–18, 19–30 and 31–42 months-of-age). These intervals were the smallest we thought possible, given the available sample sizes.

There are limited data on the predictive validity of short tests administered in early childhood in the context of large household surveys. In rural Bangladesh, low ($r$ = [0.21–0.25]) but significant correlations between monthly maternal reports of age of attainment of motor milestones (walking and standing alone) and IQ at age 5 years were found [28]. Similarly, a maternal report language test for children 12–18 months, developed locally and based on the MacArthur-Bates Communicative Development Inventories [29], significantly predicted IQ at age 5 ($r$ = [0.37–0.41]) [30].

We hypothesized that predictive validity would increase with age of the child in the early assessment, as observed for concurrent validity [10]; the Bayley-III would have highest predictive validity as it is a full diagnostic test; cognitive and language scales would perform better than other scales since measures of IQ are comprised of cognitive and language functions only [23]; and the home environment, height-for-age, and stunting would be less predictive than the short tests since they do not assess developmental domains per se but might serve as proxies, given their association with child development [1,24–26].

The study was neither designed nor powered to examine the sensitivity or specificity of the screeners in identifying high risk children [10]. Furthermore, obviously disabled children were excluded as well as children with Bayley-III scores below 70. Therefore, our results have no implications regarding the use of screeners to identify potentially high-risk children in need of further assessment. The study aimed to investigate the ability of the screeners and other short tests and measures, frequently used to evaluate interventions or measure child development at population level, given at three different age ranges, to predict later functioning in intelligence and school attainment.

## Materials and methods

### Study design and participants

The study was conducted in the poorest three, out of six, socio-economic sectors ('*estratos*' in Spanish) of Bogota. These three sectors, defined by location and the quality of housing and infrastructure, comprise low- and lower-middle-income households and account for 85% of the city's population. In 2011, we enrolled 1,533 children aged 6–42 months living in blocks in

these sectors. Within each sector stratum, blocks were selected by random probability, weighting by the proportion of women in fertile age. Within each selected block, all children 6–42 months were identified through a door-to-door census and a subsample was randomly selected for study inclusion, stratifying by age. Strata sizes (originally, 4 sector strata and 4 age strata: 6–14, 15–23, 24–32 and 33–42 months-of-age) were computed to allow for detection of differences in child development between socio-economic sector and age groups. The final study sample included 12 children from a fourth sector (middle-income), initially included in the study but subsequently dropped due to high refusal to study participation. Twins and children with obvious disabilities were excluded; and in households with more than one eligible child, one was randomized into the study. Further sample details were provided elsewhere [31].

On enrollment, all 1,533 children were randomly assigned to one of two batteries of short tests to increase the number of tests examined whilst minimizing test weariness: approximately, half the children were assessed on the short tests in battery A and the remaining half on those in battery B. Bayley-III scores were collected on 1,330 of these children immediately afterwards and 1,311 were analyzed. In 2016, we tracked and reassessed, at ages 6–8 years, as many of these children as possible. Fig 1 details the study design and participant flow.

## Procedures

**Enrollment assessments.** Battery A tests included the ASQ-3 [14], the Denver-II [15,16], and the vocabulary checklists in the SFI and SFII [20,21]. Battery B tests comprised the BDI-2 [17] and the WHO-Motor [18,19].

The ASQ-3, Denver-II and BDI-2 screeners were multi-dimensional and covered the entire age range. They all combined receptive and expressive language in one communication/language scale. Similarly, the BDI-2 motor scale combined fine and gross motor items; and the Denver-II fine-motor adaptive scale included both fine motor and cognitive items. The ASQ-3 problem solving scale comprises cognition. As reported earlier [10], all tests were administered following manual instructions except the ASQ-3, which we modified as follows. Given the low literacy levels of some caregivers, caregiver-completed items were administered by interview in order to ensure that all mothers understood the questions similarly. Furthermore, the child was tested if the caregiver did not know the answer. In addition, whenever the scale ceiling was reached in the appropriate questionnaire for the age of the child, we added the next three more difficult items from the subsequent questionnaires excluding items that had already occurred in the age-appropriate test. This reduced the number of children on the test ceiling by 10.5–15.5%, to levels of 1.7–4.8%, depending on the domain, thus increasing the variability in development captured by the test. Similar adaptations have been used elsewhere [11]. Following test manuals, the Denver-II and BDI-2 were mostly collected by direct child assessment; although up to 39% of the items in the Denver-II (mainly in the personal-social and language scales) can be obtained by caregiver report and a few BDI-2 items may also be collected by caregiver report or tester observation.

The WHO-Motor and SFs were single-domain tests and covered a limited age range. The WHO-Motor comprised six gross motor milestones directly assessed in children 6–18 months, although analysis was limited to children 6–15.9 months because most older children (91.9%) attained all milestones. The SFI and SFII collected caregiver reports on receptive and expressive language (words the child 'understands' and words the child 'understands and says') for children 8–18 months and expressive language (words the child 'says') for children 19–30 months, respectively.

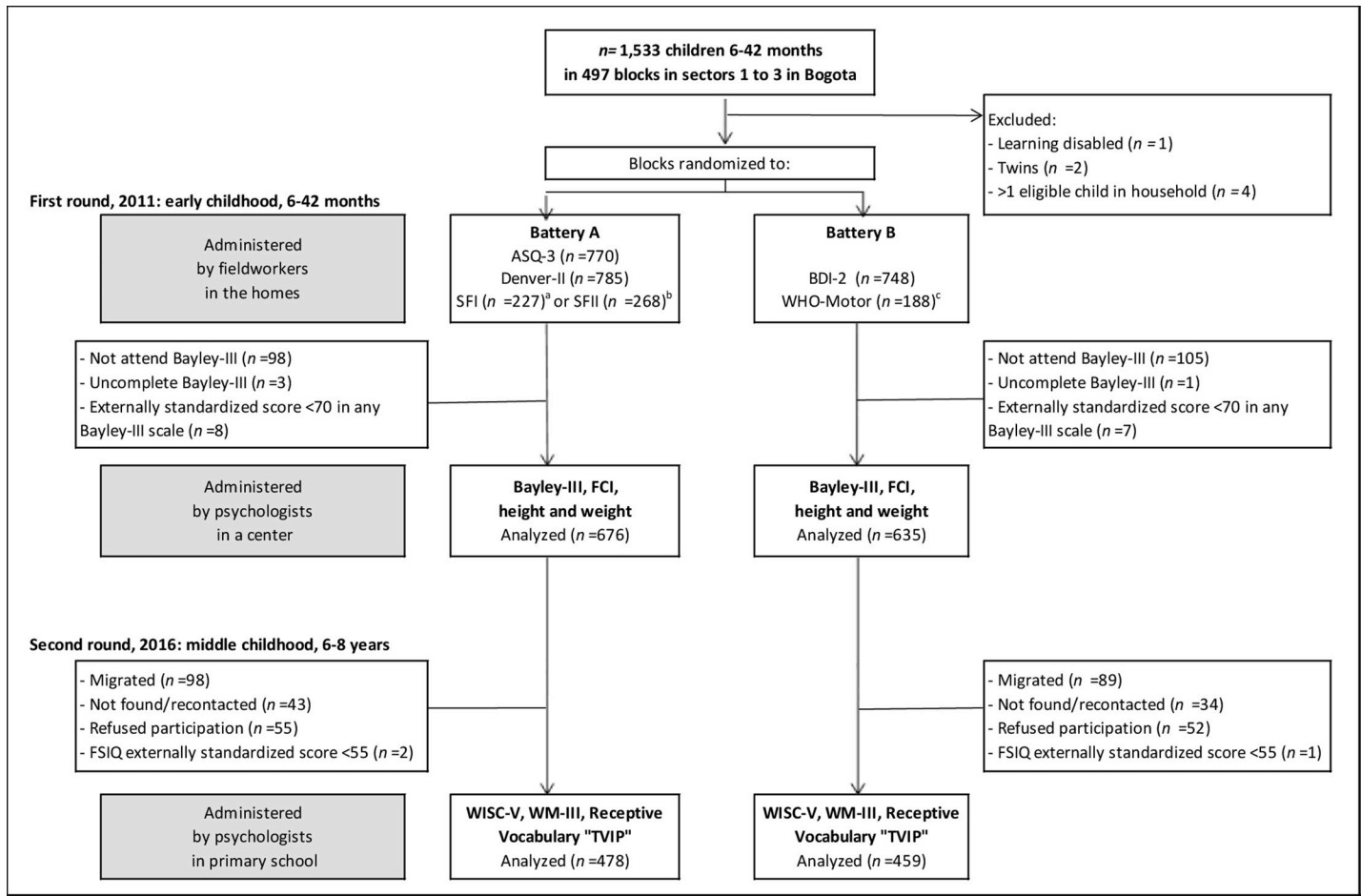

**Fig 1. Study design and flow diagram of study participants.** ASQ-3 (Ages and Stages Questionnaires, third edition); Denver-II (Denver Developmental Screening Test, second edition); SFI (MacArthur-Bates Communicative Development Inventories, Short-Form I in Spanish); SFII (MacArthur-Bates Communicative Development Inventories, Short-Form II in Spanish); BDI-2 (Battelle Developmental Inventory screener, second edition); WHO-Motor (World Health Organization Gross Motor Milestones); Bayley-III (Bayley Scales of Infant and Toddler Development, third edition); FCI (Family Care Indicators); WISC-V (Wechsler Intelligence Scale for Children, fifth edition); WM-III (Woodcock-Muñoz Test of Achievement, third edition); "TVIP" (Peabody Picture Vocabulary Test-Revised, Spanish version, adapted/selected subset of words). [a] Children 8–18 months. [b] Children 19–30 months. [c] Children 6–15 months.

All short tests were administered by non-specialized interviewers in the homes. After 5–14 days, psychologists (testers henceforth), who were blind to children's performance on the short tests, administered the Bayley-III [6] to all children in a center. All Bayley-III scales were collected by direct child assessment, except the socio-emotional one, which is by caregiver report. The testers also measured children's height following standard procedures [32]. Scores were standardized and stunting (height-for-age <-2SD of the median WHO reference standards) was computed using the WHO Anthro software, 2011. Testers/interviewers were trained for six weeks, including practices.

In preparation for testing, we translated and back-translated the Bayley-III; as well as the BDI-2 manual, and the WHO-Motor report forms and manual. Short tests in battery A were all available in Spanish. All translations and official Spanish versions were piloted and, subsequently, minor wording/phrasing modifications were made in order to reflect Colombian Spanish. Similarly, a few images had to be contextualized. Full adaptation details were provided earlier [33].

The Bayley-III was the most expensive test ($1,025 US per kit; $4.89 US per child fee, at the time of assessment) and its administration took 83 minutes on average. The BDI-2 took 63 minutes and was also expensive ($405.70 US per kit; $3.08 US per child fee), whilst the Denver-II (27 minutes) and ASQ-3 (20 minutes) were intermediate both in time and cost (Denver-II: $200 US per kit; $0.45 US per child fee and ASQ-3: $275 US per kit; no per child fee). As expected, the single-domain tests were quickest (6–8 minutes) and cheapest (SFs: $90 US per kit including both forms, $1 US per child fee; WHO: free). More details on the enrollment tests, their costs, and on the administration and training procedures were provided previously [10, 33].

**Middle childhood assessments.**   In 2016, children were tracked and reassessed. Their IQ was measured on the WISC-V using the seven subtests that constitute the Full Scale Intelligence Quotient (FSIQ): block design, similarities, matrix reasoning, digit span, coding, vocabulary, and figure weights [23]. School achievement was assessed using the arithmetic (calculations) and reading comprehension subtests in the Woodcock-Muñoz Test of Achievement (third edition, WM-III) [34], the Spanish version of the Woodcock-Johnson [35]; and a subset of 75 words from the *Test de Vocabulario en Imágenes de Peabody* (TVIP) [36], the Spanish version of the Peabody Picture Vocabulary Test-Revised [37]. These words were selected using existing data from urban children of the same age and socio-economic status from a longitudinal nationally representative survey (ELCA) [38]. We first selected the words in the relevant age range that showed sufficient variability and ordered them by difficulty; we then made final decisions after piloting. The aim was to simplify test administration and minimize testing time.

All middle childhood tests were administered following manual instructions, except the subset of 75 words in the TVIP, which were given in difficulty order until three consecutive errors were made. Tests were administered individually by psychologists at the child's primary school (91.5%), and occasionally, at another center (2.5%) or the child's home (6%). The mother, teacher or another familiar adult was sometimes present (5.8%). Total testing time was kept under 90 minutes in all cases, including a 5–10 minutes break halfway through.

We translated the WISC-V report forms and manuals and piloted the translations. All other test materials were available in Spanish and minor wording/phrasing modifications were made, after piloting, to reflect Colombian Spanish. No other adaptations were found necessary.

Twelve Psychology graduates were trained for five weeks and each carried out 15–20 practice administrations per test, until inter-observer reliabilities between trainees reached >90% item-level agreement on each test. The trainer observed 2% of study assessments, with mean agreement >95% (range = [85–100%]), and corrective feedback was given when appropriate.

**Household survey.**   In both rounds, children's homes were visited to collect household composition and other socio-economic information. On enrollment, the quality of the home environment was measured using UNICEF's Family Care Indicators (FCI) [39] for play materials and play activities. The caregivers were asked about the play activities the child engaged in with an adult over the week prior to the survey and the type of toys the child usually played with were observed. In middle childhood, an adaptation of the Middle Childhood Home Observation for Measurement of the Environment (MC-HOME) [40,41] was collected.

**Ethics.**   Ethical approval was obtained from the *Instituto de Ortopedia Infantil Roosevelt* in Bogota. Before each assessment, caregivers gave their written informed consent.

## Statistical analysis

Wealth indexes on assets and housing were constructed using polychoric principal component analyses for both rounds [31]. FCI and MC-HOME total scores were constructed by adding up binary indicators, with cutoffs varying depending on the empirical prevalence for each indicator for the FCI. The FCI score included play activities (reading/looking at picture books, telling stories, singing songs, taking child outside the home place/go for a walk, playing with toys, scribbling/drawing/coloring, naming/counting things) and play materials (toys to make/play music, things to draw/write/paint, coloring books, picture books, toys to play pretend games, toys for moving around, things meant for stacking/constructing/building, toys to learn shapes and colors).

The probability of being reassessed in 2016 was estimated using logistic regression on enrollment characteristics and its inverse was used as a weight in robustness analyses that corrected for sample loss. We also examined differences among tested participants who had been administered batteries A and B using t-tests.

For all tests, scales were administered and scored independently, and continuous raw scores were constructed following test manuals. Since the Denver-II has no raw score, we added items passed to items preceding the basal level, following general scoring principles and as done in previous work [10]. Similarly, for the WHO-Motor, we added all items the child had passed to construct the raw score [10].

No test had norms for Colombia. Therefore, we internally standardized the raw scores over age using age-conditional means and SDs, computed non-parametrically, after removing testers'/interviewers' effects, as done in previous work [10]. This is, for each value of the residual of the raw scores on tester or interviewer dummies, we constructed a z-score by subtracting the age-conditional mean and dividing by the age-conditional SD, both computed using local polynomial regressions. Unlike using norms from the reference populations (i.e. externally standardized scores) for each test, this standardization method handles age consistently across tests, which facilitates comparisons. It is also less sensitive to outliers/small sample sizes than methods traditionally used to internally standardize scores, which usually use interval-specific (for example, monthly) means and SDs to compute z-scores. FSIQ was constructed adding the internally standardized scores of the WISC-V subtests; and the school achievement score was constructed adding the arithmetic, reading and vocabulary internally standardized scores.

We examined test internal consistency using Cronbach's alpha ($\alpha$); test-retest reliability using intraclass correlations (*ICCs*); and associations among tests and with socio-economic background variables using Pearson correlations (*r*), overall and by 12-months-of-age groups. The ability of all tests, the FCI, height-for-age, and stunting to predict FSIQ and school achievement was analyzed by 12-months-of-age intervals, computing Pearson correlations within each age range. In all correlations, the internally standardized scores were used, which is equivalent to computing partial correlations controlling for testers/interviewers and age flexibly. We used the Denver-II fine motor-adaptive scale for both cognition and fine motor analyses, as there is some evidence that the scale includes both fine motor and cognitive items. Similar items to some of the Denver-II items appear in the Bayley-III cognitive scale; and, in prior analysis of concurrent validity [10], somewhat higher correlations were found between the Denver-II fine motor-adaptive scale and the Bayley-III cognitive scale than with the Bayley-III fine motor scale in children under 30 months. We used the ASQ-3, Denver-II and BDI-2 communication/language scales for both receptive and expressive language analyses; and the BDI-2 motor scale for both fine and gross motor analyses. We refer to scales predominantly measuring cognition or language by those names, hereafter. Bootstrapped *P* values [42], computed stratifying by the design strata (socio-economic sector and age), were used in all

inference and to compare whether the predictive validity of a test differed significantly across the three age groups examined. Similarly, bootstrapped *P* values were used to compare the predictive validity of the Bayley-III, the short tests, the FCI, height-for-age, and stunting. Infant tests measure a range of developmental domains whereas FSIQs only measure cognitive and language functions. Therefore, for the short tests, we only compare the predictive validity of the cognitive and language scales but present predictive validity values of other scales for interest.

To further explore the effect of age under 19 months in the Bayley-III, the FCI, height-for-age, and stunting, which were available for children in both batteries, we repeated the analyses by 6-months-of-age intervals on enrollment. For the other tests, given to children in either battery A or battery B, sample sizes were considered too small to subdivide.

We classify correlations as very low ($r = 0.10$–$0.19$), low ($r = 0.20$–$0.39$), moderate ($r = 0.40$–$0.59$), and high ($r = 0.60$–$0.79$) [43].

Statistical analyses were performed using Stata 14.2 (StataCorp, College Station, TX).

## Results

We reassessed 940 (71.7%) of the 1,311 children with Bayley-III scores on enrollment, of which 3 scored < -3 SD in the externally standardized FSIQ scores and were dropped (Fig 1). Main reasons for sample loss were migration (50.4%), inability to recontact the household (20.7%), refusal to continue study participation (21.6%), and refusal to be administered the WISC-V (7.3%). Migrants who could be located and lived within 1-hour from Bogota (by bus) were tracked. Higher attrition occurred among poorer households, girls, and younger/less educated mothers. Children tested in middle childhood, who were originally given either battery A or battery B, were comparable in terms of their characteristics, except for mother's age ($P = 0.037$) (Table 1), which had significant but very low associations ($r < 0.127$, $P < 0.001$) with outcomes (not shown).

Raw scores of all middle childhood tests and subtests, reported in S1 Table for all children and by test battery given at enrollment, increased with age and school grade (S2 Table). The average WISC-V externally standardized scores were below those of the norming sample (Table 1) and internal consistency was good (all α's >0.6, except two) and stable over time (not shown). Test-retest reliabilities after 6–14 days were also good ([$ICC = 0.39$–$0.88$], $ICC > 0.6$ for most tests). FSIQ and school achievement were associated with household characteristics and with each other as theoretically expected (Table 2); these correlations were higher for children 7 and 8 years than for the 6-year olds (not shown). S1 Table also reports raw scores for the short tests and raw and externally standardized scores for the Bayley-III, for all children and by test battery given at enrollment.

Fig 2 shows correlations of the short tests and the Bayley-III with FSIQ and S1 Fig with school achievement. Table 3 reports correlations among all enrollment and middle childhood scores by domain/scale and age group, and the bottom panel shows the FCI, height-for-age, and stunting. Table 4 shows the significantly different correlations with the language and cognition scales only and with the FCI and stunting. S3 Table reports comparisons across age groups. Cognition, language and, to a lesser extent, fine motor scales were the most predictive. The Bayley-III generally had the highest correlations; and for all tests, correlations increased with age of initial test, except for the ASQ-3 cognition which decreased at 19–30 months. FSIQ and school achievement were highly correlated ($r = 0.706$, $P < 0.001$, Table 2) and had similar patterns of correlations (S1 Fig, Table 3). We therefore focus on the FSIQ.

**Table 1. Characteristics of children tested in middle childhood and their families, by test battery given at enrollment.**

| | Battery A ($n_A$ = 478) | Battery B ($n_B$ = 459) | P value of difference between batteries |
|---|---|---|---|
| **I. Child characteristics** | | | |
| Child's age at enrollment, % | | | |
| 6–18 months | 31.6 | 33.1 | 0.640 |
| 19–30 months | 36.2 | 35.5 | 0.822 |
| 31–42 months | 32.2 | 31.4 | 0.787 |
| Child's age in middle childhood, % | | | |
| 6 years | 33.1 | 33.1 | 0.985 |
| 7 years | 35.4 | 36.6 | 0.698 |
| 8 years | 31.6 | 30.3 | 0.674 |
| Girls, % | 45.4 | 49.7 | 0.173 |
| Premature (gestational age <37 weeks), % | 14.6 | 15.5 | 0.754 |
| Height-for-age[a] at enrollment, z-score, mean (SD) | -1.1 (1.1) | -1.1 (1.1) | 0.319 |
| Stunted[a] (z-score height-for-age <-2SD) at enrollment, % | 16.6 | 17.9 | 0.595 |
| **II. Parental characteristics in middle childhood** | | | |
| Mother's age[a], y, mean (SD) | 33.3 (6.9) | 32.3 (6.5) | 0.037 |
| Mother's education, y, mean (SD) | 10.9 (3.2) | 11.2 (3.3) | 0.241 |
| Father's education[a], y, mean (SD) | 8.9 (4.1) | 9.3 (4.0) | 0.087 |
| **III. Household characteristics and home environment** | | | |
| Socio-economic sector (strata) in middle childhood, % | | | |
| 1. Lowest | 27.2 | 27.5 | 0.930 |
| 2. | 39.7 | 42.5 | 0.422 |
| 3. | 32.0 | 28.5 | 0.329 |
| 4. Highest | 1.0 | 1.5 | 0.598 |
| Household size in middle childhood, mean (SD) | 4.5 (1.4) | 4.3 (1.5) | 0.133 |
| Household wealth index in middle childhood, mean (SD) | -0.01 (1.03) | 0.03 (0.95) | 0.520 |
| Number of varieties of play materials (FCI) at enrollment, mean (SD) | 5.0 (2.3) | 5.0 (2.3) | 0.143 |
| Number of varieties of play activities (FCI) at enrollment, mean (SD) | 3.9 (1.9) | 3.7 (1.8) | 0.955 |
| Total FCI score (play materials and activities) at enrollment, internally standardized, mean (SD) | 0.16 (1.7) | 0.13 (1.7) | 0.786 |
| Total MC-HOME score in middle childhood, internally standardized, mean (SD) | 0.05 (0.9) | -0.05 (1.0) | 0.145 |
| **IV. Child development at enrollment** | | | |
| Bayley-III, internally standardized scores, mean (SD) | | | |
| Cognitive | 0.03 (0.10) | 0.05 (0.99) | 0.763 |
| Receptive Language | 0.08 (0.99) | 0.02 (1.02) | 0.346 |
| Expressive Language | 0.04 (0.99) | 0.02 (1.00) | 0.713 |
| Fine Motor | 0.05 (0.98) | 0.00 (0.99) | 0.390 |
| Gross Motor | 0.03 (1.02) | 0.01 (0.10) | 0.758 |
| Socio-emotional | 0.02 (0.98) | 0.00 (0.97) | 0.702 |
| **V. Child development in middle childhood** | | | |
| FSIQ, WISC-V, externally standardized, mean (SD) | 88.6 (12.1) | 88.6 (12.6) | 0.991 |
| FSIQ, WISC-V, internally standardized, mean (SD) | 0.12 (4.40) | -0.04 (4.50) | 0.605 |
| Achievement score, internally standardized, mean (SD) | 0.03 (2.39) | 0.01 (2.39) | 0.903 |

[a] Missing data for some variables. Sample sizes for these are: height-for-age and stunted ($n_A$ = 477); mother's age ($n_A$ = 453, $n_B$ = 433); father's education ($n_A$ = 456, $n_B$ = 424). SD is Standard Deviation.

**Table 2.  Correlations of middle childhood tests with concurrent socio-economic variables and with each other.**

|  | FSIQ WISC-V | Achievement score | Arithmetic WM-III | Reading comprehension WM-III | Receptive vocabulary, based on TVIP |
|---|---|---|---|---|---|
| **Socio-economic variables** |  |  |  |  |  |
| Mother's education, y | 0.315*** | 0.288*** | 0.172*** | 0.292*** | 0.226*** |
| Household wealth index | 0.308*** | 0.300*** | 0.212*** | 0.285*** | 0.221*** |
| Total MC-HOME score, internally standardized | 0.331*** | 0.342*** | 0.266*** | 0.289*** | 0.264*** |
| **Middle childhood tests, internally standardized** |  |  |  |  |  |
| FSIQ, WISC-V | 1 |  |  |  |  |
| Achievement score | 0.706*** | 1 |  |  |  |
| Arithmetic, WM-III | 0.531*** | 0.798*** | 1 |  |  |
| Reading comprehension, WM-III | 0.600*** | 0.824*** | 0.512*** | 1 |  |
| Receptive vocabulary, based on TVIP | 0.563*** | 0.776*** | 0.402*** | 0.461*** | 1 |

N = 937 children. Pearson correlations on internally standardized test scores (net of testers'/interviewers' effects). Standard Errors (SE) computed using bootstrap methods, stratifying by the design strata: socio-economic sector and age (n = 2,000 replications).

## Multi-dimensional tests

Under 19 months, all tests had very low predictive validity (all $r < 0.185$), although some were significant (Bayley-III cognition and receptive language, and ASQ-3 cognition).

At 19–30 months, the Bayley-III, Denver-II and BDI-2 cognitive and language scales had similar low but significant correlations with FSIQ ($r = 0.330$, $P < 0.001$; $r = 0.229$, $P < 0.01$; $r = 0.221$, $P < 0.01$, respectively for cognition; and $r = 0.307$, $P < 0.001$; $r = 0.214$, $P < 0.01$; $r = 0.244$, $P < 0.01$, for receptive language/language). All three tests' cognitive scales had significantly higher correlations than the ASQ-3 cognition (Table 4), which was not significantly related to FSIQ (Table 3). The ASQ-3 language also had very low correlations ($r = 0.180$, $P < 0.05$) but not significantly different from other scales.

At 31–42 months, correlations of the four multi-dimensional tests with FSIQ increased and were significant. The Bayley-III cognition ($r = 0.474$, $P < 0.001$) and receptive language ($r = 0.409$, $P < 0.001$) and the Denver-II cognition ($r = 0.422$, $P < 0.001$) had the highest predictive validity with moderate levels, whereas the others were low ($r = 0.271$–$0.386$, $P < 0.05$). The Bayley-III cognition was not different from the Denver-II language or cognition but was significantly higher than the BDI-2 and ASQ-3 language, and than the ASQ-3 cognition (Table 4).

The increase of the correlations across age groups was significant between the youngest age group (children 6–18 months) and the oldest (children 31–42 months) for the Bayley-III and the Denver-II cognitive and language scales (S3 Table). For the Bayley-III scales the increase was also significant between the youngest and the middle age group (children 19–30 months).

Fine motor scales were generally less predictive of FSIQ than language or cognition (all $r < 0.353$), except for the Denver-II where it is combined with cognition. Gross motor scales showed very little association with FSIQ (all $r < 0.228$), with the highest values for the BDI-2 where it is combined with fine motor.

## Single-domain tests

Similar to the other short tests, the SFII was not significantly correlated with FSIQ below 19 months. At 19–30 months, however, this correlation ($r = 0.301$, $P < 0.001$) was low and similar

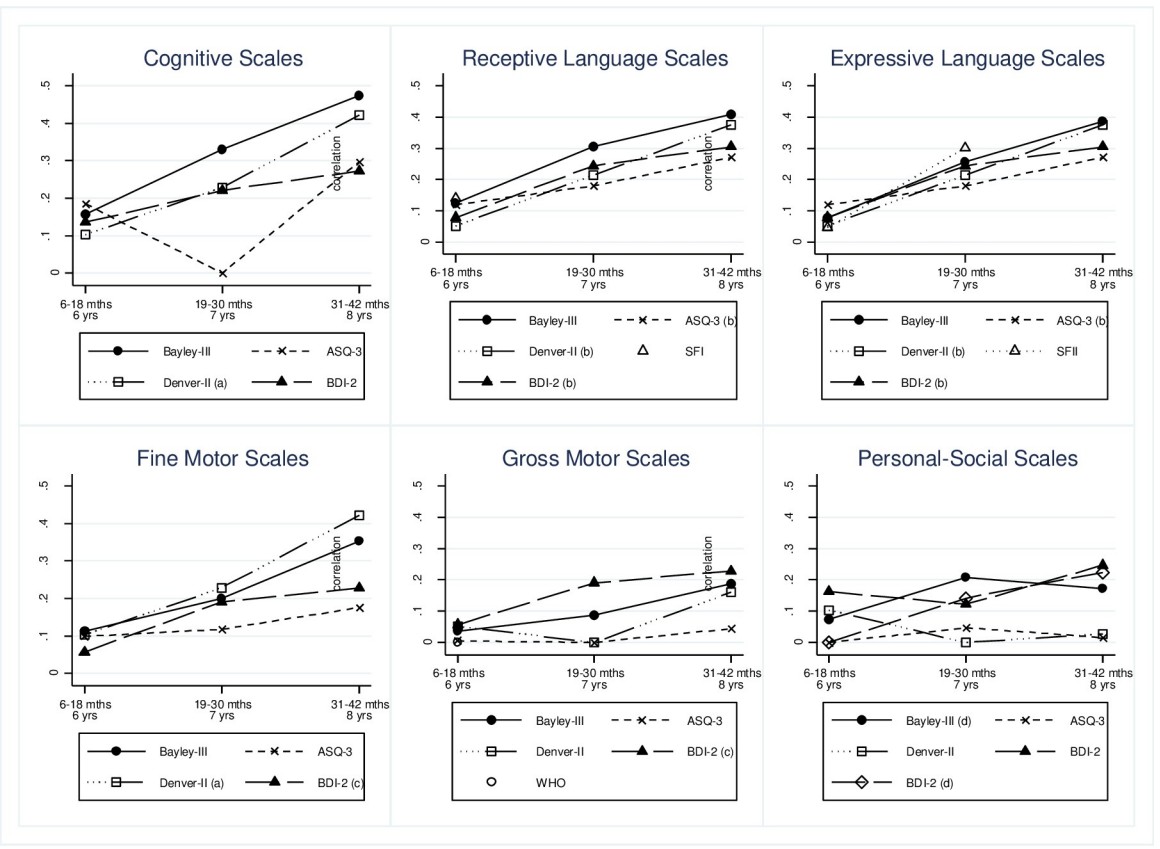

**Fig 2. Predictive validity of Bayley-III and short tests at 6–42 months with the FSIQ at 6–8 years, by age group and domain.** Each plot in Fig 2 shows, for each developmental domain, the average correlation between scores in early childhood with FSIQ scores in middle childhood, by age group. The youngest age group are children 6–18 months in early childhood and 6 years in middle childhood; the middle age group are children 19–30 months in early childhood and 7 years in middle childhood; and the oldest age group are children 31–42 months in early childhood and 8 years in middle childhood. Sample sizes for each correlation (point depicted) are in Table 2. See Fig 1 footnote for the definition of the acronyms used. [a]Denver-II is the fine-motor adaptive scale; [b]ASQ-3, Denver-II and BDI-2 are the communication/language scales; [c]BDI-2 motor combines fine and gross motor items; [d]Bayley-III is the socio-emotional scale; BDI-2 is the adaptive skills scale.

to those of the Bayley-III cognitive and language scales and slightly higher than those in the other tests, although only significantly higher than the ASQ-3 cognition ($P < 0.000$, Table 4). The increase in predictive validity between the SFI (children 8–18 months) and the SFII (children 19–30 months) was statistically significant (S3 Table).

The WHO-Motor did not significantly predict FSIQ.

## Family Care Indicators (FCI), height-for-age, and stunting

The FCI had a very low but significant correlation with FSIQ below 19 months ($r = 0.183$, $P < 0.01$), which increased to low at 19–30 months ($r = 0.362$, $P < 0.001$), similar to the correlations of the Bayley-III cognitive and language scales and significantly higher than those in the ASQ-3 and the Denver-II cognition. This increase was statistically significant (S3 Table). At 31–42 months, the FCI correlation with FSIQ remained low ($r = 0.329$, $P < 0.001$) and was comparable to those in other short tests. It was, however, significantly lower than that of the Bayley-III cognition ($P = 0.035$, Table 4).

The predictive validity of height-for-age and stunting increased with age, although the increase was not statistically significant (S3 Table). Neither were significant at 6–18 months.

**Table 3. Predictive validity of the Bayley-III, the short tests, the FCI, height-for-age, and stunting at 6–42 months with the FSIQ and school achievement at 6–8 years, by age at enrollment.**

| Tests at Enrollment (6–42 months) | 6–18 Months at Enrollment | | 19–30 Months at Enrollment | | 31–42 Months at Enrollment | |
|---|---|---|---|---|---|---|
| | FSIQ | Achievement | FSIQ | Achievement | FSIQ | Achievement |
| **Bayley-III** | n = 303 | | n = 336 | | n = 298 | |
| Cognitive | 0.157** | 0.135* | 0.330*** | 0.309*** | 0.474*** | 0.436*** |
| Receptive language | 0.126* | 0.150** | 0.307*** | 0.348*** | 0.409*** | 0.397*** |
| Expressive language | 0.079 | 0.089 | 0.257*** | 0.308*** | 0.386*** | 0.398*** |
| Fine motor | 0.113* | 0.122* | 0.201*** | 0.213*** | 0.353*** | 0.311*** |
| Gross motor | 0.036 | -0.071 | 0.087 | 0.143** | 0.188** | 0.175** |
| Socio-emotional | 0.073 | 0.019 | 0.207*** | 0.208*** | 0.172** | 0.119* |
| **ASQ-3 (adapted)** | n = 145 | | n = 172 | | n = 153 | |
| Problem solving | 0.185** | 0.027 | -0.122 | -0.008 | 0.297*** | 0.310*** |
| Communication | 0.120 | 0.112 | 0.180* | 0.223** | 0.271*** | 0.317*** |
| Fine motor | 0.099 | 0.038 | 0.118 | 0.167* | 0.176** | 0.247*** |
| Gross motor | 0.005 | -0.065 | -0.099 | -0.110 | 0.044 | 0.038 |
| Personal-social | -0.115 | -0.021 | 0.047 | 0.057 | 0.016 | 0.009 |
| **Denver-II** | n = 148 | | n = 169 | | n = 148 | |
| Language | 0.052 | -0.026 | 0.214** | 0.286*** | 0.375*** | 0.373*** |
| Fine motor-adaptive | 0.103 | 0.116 | 0.229** | 0.146* | 0.422*** | 0.438*** |
| Gross motor | 0.052 | -0.082 | -0.005 | 0.035 | 0.160* | 0.190* |
| Personal-social | 0.104 | 0.015 | -0.031 | 0.049 | 0.028 | 0.058 |
| **BDI-2 (Battelle)** | n = 152 | | n = 163 | | n = 144 | |
| Cognitive | 0.136 | 0.175* | 0.221** | 0.238** | 0.272* | 0.305*** |
| Communication | 0.079 | 0.095 | 0.244** | 0.326*** | 0.305*** | 0.318*** |
| Motor | 0.057 | -0.010 | 0.191* | 0.136 | 0.228* | 0.220* |
| Personal-social | 0.163* | 0.125 | 0.122 | 0.181** | 0.247** | 0.224** |
| Adaptive skills | -0.058 | -0.033 | 0.142 | 0.083 | 0.224** | 0.192* |
| **SFI & SFII (MacArthur)** | n = 126[a] | | n = 172 | | | |
| Receptive language | 0.140 | 0.181* | | | | |
| Expressive language | 0.047 | 0.055 | 0.301*** | 0.298*** | | |
| **WHO-Motor** | n = 110[b] | | | | | |
| Gross Motor | -0.033 | -0.068 | | | | |
| **FCI, Height-for-age, Stunting** | n = 303 | | n = 336 | | n = 298 | |
| FCI | 0.183** | 0.180** | 0.362*** | 0.384*** | 0.329*** | 0.310*** |
| Height-for-age | 0.085 | 0.088 | 0.164** | 0.172** | 0.203*** | 0.206*** |
| Stunting | -0.054 | -0.113* | -0.179*** | -0.207*** | -0.200*** | -0.247*** |

Pearson correlations on internally standardized scores (net of testers'/interviewers' effects), except for height-for-age and stunting. *P* values computed using bootstrap methods, stratifying by the design strata: socio-economic sector and age (n = 2,000 replications). FCI includes play materials and play activities. Stunting is defined as height-for age <-2 SD of the WHO reference median.

* p<0.05

** p<0.01

*** p<0.001.

[a] Children 8–18 months

[b] Children 6–15 months.

At 19–30 months, both were very low (*r* = 0.164 for height-for-age; *r* = -0.179 for stunting; both *P* <0.001) and significantly less predictive than the FCI (*P* = 0.012 for stunting, *P* = 0.004 for height-for-age, Table 4). At 31–42 months, these predictive validities increased slightly to

**Table 4. All significantly different ($p$ <0.05) correlations of cognitive and language scales of all short tests, Bayley-III, FCI, height-for-age, and stunting on enrollment with later FSIQ and school achievement.**

| | FSIQ | Achievement |
|---|---|---|
| **19–30 months** | Bayley-III cognitive > ASQ-3 problem solving ($P$ <0.0001) | Bayley-III cognitive > ASQ-3 problem solving ($P$ = 0.001) |
| | Bayley-III cognitive > Height-for-age ($P$ = 0.037) | Bayley-III cognitive > Denver-II fine motor-adaptive ($P$ = 0.033) |
| | Denver-II fine motor-adaptive > ASQ-3 problem solving ($P$ <0.0001) | Bayley-III language expressive > Height-for-age ($P$ = 0.037) |
| | BDI-2 cognitive > ASQ-3 problem solving ($P$ = 0.002) | BDI-2 cognitive > ASQ-3 problem solving ($P$ = 0.030) |
| | SFII expressive language > ASQ-3 problem solving ($P$ <0.0001) | SFII expressive language > ASQ-3 problem solving ($P$ = 0.003) |
| | FCI > ASQ-3 problem solving ($P$ <0.0001) | FCI > ASQ-3 problem solving ($P$ <0.0001) |
| | FCI > ASQ-3 communication ($P$ = 0.012) | FCI > ASQ-3 communication ($P$ = 0.035) |
| | FCI > Denver-II fine motor-adaptive ($P$ = 0.049) | FCI > Denver-II fine motor-adaptive ($P$ = 0.002) |
| | FCI > Stunting ($P$ = 0.012) | FCI > Stunting ($P$ = 0.014) |
| | FCI > Height-for-age ($P$ = 0.003) | FCI > Height-for-age ($P$ = 0.002) |
| | Stunting > ASQ-3 problem solving ($P$ = 0.003) | |
| | Height-for-age > ASQ-3 problem solving ($P$ = 0.005) | |
| **31–42 months** | Bayley-III cognitive > ASQ-3 problem solving ($P$ = 0.038) | Bayley-III cognitive > FCI ($P$ = 0.027) |
| | Bayley-III cognitive > ASQ-3 communication ($P$ = 0.033) | Bayley-III cognitive > Stunting ($P$ = 0.007) |
| | Bayley-III cognitive > BDI-2 communication ($P$ = 0.016) | Bayley-III expressive language > Stunting ($P$ = 0.022) |
| | Bayley-III cognitive > FCI ($P$ = 0.035) | Bayley-III cognitive > Height-for-age ($P$ = 0.001) |
| | Bayley-III cognitive > Stunting ($P$ <0.0001) | Bayley-III receptive language > Height-for-age ($P$ = 0.017) |
| | Bayley-III receptive language > Stunting ($P$ = 0.010) | Bayley-III expressive language > Height-for-age ($P$ = 0.003) |
| | Bayley-III expressive language > Stunting ($P$ = 0.007) | Denver-II fine motor-adaptive > Stunting ($P$ = 0.012) |
| | Bayley-III cognitive > Height-for-age ($P$ <0.0001) | Denver-II fine motor-adaptive > Height-for-age ($P$ = 0.004) |
| | Bayley-III receptive language > Height-for-age ($P$ = 0.010) | |
| | Bayley-III expressive language > Height-for-age ($P$ = 0.008) | |
| | Denver-II fine motor-adaptive > Stunting ($P$ = 0.004) | |
| | Denver-II language > Stunting ($P$ = 0.046) | |
| | Denver-II fine motor-adaptive > Height-for-age ($P$ = 0.007) | |
| | Denver-II language > Height-for-age ($P$ = 0.045) | |

Number of observations for each correlation compared as in Table 3. $P$ values of comparisons computed using bootstrap methods, stratifying by the design strata: socio-economic sector and age (n = 2,000 replications).

low levels ($r$ = 0.203 for height-for-age; $r$ = -0.200 for stunting, both $P$ <0.001). Height-for-age remained significantly lower than the Bayley-III and Denver-II cognition and language ($P$ = 0.001, Table 4), but not than the FCI.

## Six-monthly age intervals analyses

Analyses of Bayley-III, FCI, height-for-age, and stunting in 6-months-of-age intervals under 19 months showed that none significantly predicted FSIQ before 12 months. At 13–18 months, the correlations of Bayley-III cognition, receptive language and fine motor with FSIQ increased ($r$ = 0.231, $P$<0.05; $r$ = 0.204, $P$<0.1; $r$ = 0.190, $P$<0.1, respectively). The FCI was also predictive ($r$ = 0.240, $P$<0.05) but not height-for-age nor stunting.

Analyses were repeated using FSIQ externally standardized scores, scores internally standardized using traditional methods, ASQ-3 scores using the original 6-item questionnaires, dropping outliers <-2 SD in the FSIQ distribution, and weighting by the inverse probability of being assessed in middle childhood to correct for loss at follow-up. Results were little altered in all cases.

## Discussion

As expected, the Bayley-III generally had the highest predictive validity, but was not significant under 12 months, it was very low at 13–18 months, low from 19–30, and became moderate at 31–42 months. It is recognized that standard tests given before 24 months have low associations with abilities in later childhood [22,44] and our findings are comparable to those reported elsewhere [2,28,30,45,46].

The sample was well balanced across the three 12 months-of-age groups of analysis. Predictive validity of the tests generally increased with age, the differences being particularly significant between 6–18 and 31–42 months for the Bayley-III and the Denver-II. All short tests had very low predictive validity before 19 months and hence little value as predictors of future functioning. At 19–30 months, the Denver-II and BDI-2 cognitive and language scales had similar low correlations to the Bayley-III and the correlations of all three cognitive scales were significantly higher than that of the ASQ-3. At 31–42 months, the Denver-II and Bayley-III had similar moderate levels, whereas the BDI-2 and ASQ-3 were significantly lower than the Bayley-III.

The WHO-Motor was not predictive of IQ or school achievement. In contrast, in Bangladesh, age of attainment of motor milestones showed significant but low associations with IQ at 5 years [28]. However, the age of attainment of milestones was recorded by the mothers throughout the first year which would probably be more accurate than a one-off examination. In high-income countries, early motor scores were also weaker predictors of later function than language and cognitive scores [47].

The SFII's predictive validity was comparable to that of the Bayley-III language and cognition, at ages 19–30 months, when vocabulary acquisition is rapidly increasing. In Bangladesh, a similar vocabulary test given at age 18 months also had comparable predictive ability with IQ at 5 years [30]. Maternal reports do not require the child to engage with the tester, which is an advantage as young children in LMICs are often inhibited. Nonetheless, while the SFII is available in many languages [48], new inventories would have to be developed for new languages and regional adaptations may be required for existing inventories. In the US, the SFs at 24 months were also as predictive of later language as the Bayley-III, although predictive validity varied by social background [49].

The choice of which specific test to use should be informed by the cost of the test, administration time and skill required, as well as concurrent and predictive validity. Over age 18 months, of the short tests investigated, the Denver-II appeared to be the best candidate for use at scale, showing the closest predictive ability to the Bayley-III, although it was low-to-moderate, as indicated earlier. The BDI-2 took too long; and the ASQ-3 had poor validity under 31 months. The Denver-II administration took around 27 minutes, approximately one third of the time for a Bayley-III test. However, this time may still be too long in large scale studies. Although multi-dimensional tests are generally desirable [50], particularly if resources are limited, a possible compromise might be to use only the cognitive and language scales of the selected test to shorten assessments since these scales are the most likely to be affected by poverty [31,51] and have the highest predictive ability. Another alternative is to use single-domain tests. The SFII, for example, can be useful at 19–30 months if available in the local language. Research is however needed to extend findings to 36 months and to develop SFs versions in new languages.

The choice of the test also depends on the objectives of the survey [10]. Caregiver reports might be better suited for the evaluation of population-based indicators and less convenient to evaluate psychosocial stimulation programs as they may suffer from "observation" bias, if mothers in the treatment group spend more time with the child and are more aware of the

process of development/achievement of milestones as a result of the intervention. Moreover, intervened mothers may be biased towards making optimistic claims of their children's development to report on intervention success ("desirability" bias). Similarly, the evaluation of nutritional interventions might favor the use of a gross motor scale, which would not be advised to evaluate psychosocial stimulation interventions given the lower predictive ability of later intellectual functioning of the gross motor scales investigated compared to the cognitive and language ones. Whilst both the Denver-II and the SFII have been found to be sensitive to the impact of cash transfer programs in Nicaragua [12] and Ecuador [13], respectively, further investigation of sensitivity to interventions for all short tests would be helpful.

It is remarkable that under 31 months the FCI showed similar or higher predictive ability than any test including the Bayley-III. It is free, quick (10–15 min) and easy to give, provides information on useful activities for parents (although not on responsive caregiving) and has been widely used in international surveys—most notably, UNICEF's Multiple Indicator Cluster Surveys [27]. Whilst individual item performance can vary depending on the context and children's age, home environment quality has been identified as a relevant protective factor [25,26] and the FCI has often improved with ECD interventions [52–54]. We hypothesize that adding the FCI to the Denver-II cognitive and language scales as suggested above might improve the sensitivity of a program evaluation and needs to be investigated elsewhere.

Furthermore, although stunting has been used as an indicator of inadequate child development globally [1,24], the FCI was a better predictor of future overall intellectual functioning and school achievement in this population. If these findings are replicated in countries with different home environments and stunting severity and prevalence, combining both indicators would be a more effective population-based indicator.

Finally, the above findings could be extrapolated to urban Colombia and possibly urban areas in other Latin American countries. Following our earlier study of these tests' concurrent validity [10], it was shown that the Denver-II was also appropriate for Brazil [55]. Further studies would be required before extrapolating to rural areas and other LMICs.

The number of short tests included in the current study was limited due to time and budgetary constraints. Short tests likely to be more suitable for Africa and Asia, such as the Malawi Developmental Assessment Tool (MDAT) [56] could be similarly evaluated. Moreover, several tests are currently under development including population-based and individual-level instruments for children 0–36 months [3,4] using a new approach, the D-score, which summarizes overall development using an interval scale [57]. After further piloting, these may be appropriate for use globally [4].

An important point is that some of the short tests have been shown to have good sensitivity and specificity when identifying children in LMICs at high-risk of disability [58]. By design, our study did not address this issue: the study sample was insufficient, not representative of high-risk populations (e.g. premature or low birthweight children), and obviously disabled children were excluded [10]. Therefore, our findings and recommendations are not generalizable to these subpopulation groups.

Study limitations include the high attrition and the lack of standardization of the middle childhood tests for Colombia. However, weighting the analyses for sample loss did not change the findings, and the middle childhood tests showed good reliability and correlated with socioeconomic characteristics, with each other, and with earlier measures of development, thus appearing to be valid in this population. Another problem is that the Denver-II does not have a separate cognitive scale per se. However, the fine motor-adaptive scale combines cognitive and fine motor items and we have shown that it correlated well with later IQ, better or similar to the cognitive scales in other short tests and similar to the Bayley-III cognition for children 19 months and older. A further limitation is that the predictive validity of the early tests might

be partly confounded with the performance of the middle childhood tests since, in middle childhood, all children were tested 5.5 years after the early assessment. Performance of the middle childhood tests seems to increase with age according to some reliability and validity indicators (correlations of tests with each other and with socio-economic variables) but not to some others (internal consistency).

Study strengths are the large, population-based (albeit urban) sample of children assessed in early and in middle childhood, the number of tests evaluated, and the quality of the 'gold standard'.

## Conclusions

The predictive validity of all tests generally increased with age. Language and cognition were the scales with the highest predictive validity. No short test had useful predictive ability under 19 months and, after this age, predictive validity values were only low-to-moderate at best. The SFII from 19–30 months and the Denver-II from 19–42 months were the most feasible and valid short tests of those investigated. Under 31 months, the FCI was as good a predictor as the Bayley-III and better than stunting. These findings suggest that it may be worth piloting a combination of the FCI and the language and cognitive scales of the Denver-II to evaluate large ECD interventions with children under 36 months. Adding the FCI to stunting may also improve the global estimate of children at risk of poor development (population-based indicator). These findings need to be examined in other regions, when recently developed tests, could be included.

## Supporting information

**S1 Table. Raw scores for the Bayley-III, short tests, WISC-V and school achievement measures, and composite (Externally Standardized) scores for the Bayley-III. all children and by test battery given at enrollment.**
(DOCX)

**S2 Table. Correlations of WISC-V and school achievement raw scores with age and school grade in middle childhood.**
(DOCX)

**S3 Table. Test of significance (*P* values) of Correlations between age groups: Cognitive and language scales of the Bayley-III, the short tests, the FCI, height-for-age, and stunting on enrollment with later FSIQ and school achievement.**
(DOCX)

**S1 Fig. Predictive validity of the Bayley-III and the short tests at 6–42 months of school achievement at 6–8 years, by age group and domain.**
(TIF)

**S1 File. STROBE checklist.**
(PDF)

**S2 File. Approved study protocols.** Study protocols approved by IRB at *Instituto de Ortopedia Infantil Roosevelt* in Bogota, Colombia (in Spanish).
(PDF)

## Acknowledgments

We thank all the families who participated in the study; and the primary schools for lending us their facilities for testing. We are also grateful to all testers and interviewers; to Andrea Solano and Marlenny Solano for providing excellent training and field coordination; to Paula Bernal for invaluable technical advice on tests and adaptations; and to María Adelaida Martínez and Marta Dormal for research assistance. This work would not have been possible without the encouragement and support from María Caridad Araujo and Ana Lucía Muñoz.

## Author Contributions

**Conceptualization:** Marta Rubio-Codina, Sally Grantham-McGregor.

**Data curation:** Marta Rubio-Codina.

**Formal analysis:** Marta Rubio-Codina.

**Funding acquisition:** Marta Rubio-Codina.

**Investigation:** Marta Rubio-Codina.

**Methodology:** Marta Rubio-Codina, Sally Grantham-McGregor.

**Project administration:** Marta Rubio-Codina.

**Supervision:** Sally Grantham-McGregor.

**Writing – original draft:** Marta Rubio-Codina, Sally Grantham-McGregor.

**Writing – review & editing:** Marta Rubio-Codina, Sally Grantham-McGregor.

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
