## [Decision Letter · Decision Letter 0]

2 Sep 2019

PONE-D-19-20029

Predictive validity in middle childhood of short tests of early childhood development (ECD) used in large scale studies compared to the Bayley-III, the Family Care Indicators and stunting: A longitudinal study in Bogota, Colombia

PLOS ONE

Dear Dr Rubio-Codina,

Thank you for submitting your manuscript to PLOS ONE. After careful consideration, we feel that it has merit but does not fully meet PLOS ONE’s publication criteria as it currently stands. Therefore, we invite you to submit a revised version of the manuscript that addresses the points raised during the review process.

We would appreciate receiving your revised manuscript by Oct 17 2019 11:59PM. To enhance the reproducibility of your results, we recommend that if applicable you deposit your laboratory protocols in protocols.io, where a protocol can be assigned its own identifier (DOI) such that it can be cited independently in the future. For instructions see: http://journals.plos.org/plosone/s/submission-guidelines#loc-laboratory-protocols

We look forward to receiving your revised manuscript.

Kind regards,

Thach Duc Tran, M.Sc., Ph.D.

Academic Editor

PLOS ONE

Journal Requirements:

1. We noticed you have some minor occurrence of overlapping text with the following previous publication, which needs to be addressed: Rubio-Codina, Marta, et al. "Concurrent validity and feasibility of short tests currently used to measure early childhood development in large scale studies." PLoS One 11.8 (2016): e0160962. In your revision ensure you cite all your sources (including your own works), and quote or rephrase any duplicated text outside the methods section. Further consideration is dependent on these concerns being addressed.

Reviewers' comments:

Reviewer's Responses to Questions

**Comments to the Author**

1. Is the manuscript technically sound, and do the data support the conclusions?

Reviewer #1: Partly

Reviewer #2: Yes

Reviewer #3: Yes

2. Has the statistical analysis been performed appropriately and rigorously? 

Reviewer #1: Yes

Reviewer #2: Yes

Reviewer #3: Yes

3. Have the authors made all data underlying the findings in their manuscript fully available?

Reviewer #1: Yes

Reviewer #2: Yes

Reviewer #3: Yes

4. Is the manuscript presented in an intelligible fashion and written in standard English?

Reviewer #1: Yes

Reviewer #2: Yes

Reviewer #3: Yes

5. Review Comments to the Author

Reviewer #1: PONE-D-19-20029 Predictive validity in middle childhood of short tests of 1 early childhood development (ECD) used in large scale studies compared to the Bayley-III, the Family Care Indicators and stunting: A longitudinal study in Bogota, Colombia

Abstract first sentence and Line 60. It is not clear why interventions are hindered by large-scale measurement of child development. This implies that large-scale interventions require that all children be tested, rather than a representative sample, as is usual. Otherwise, the most convincing rationale for this study was the value in having population-level indicators. Perhaps reference to evaluations of programs can be deleted.

Abstract. The data do not really support the first part of the final statement, namely that the FCI is an appropriate indicator for evaluations of programmes promoting mental development. However, I do agree with the second part, that the FCI and stunting together could be used as population-level proxies of mental development.

Introduction

Instead of the #2 Frongillo reference (are any of these authors experts in educational/psychological measurement of child development?) you might include the 2017 World Bank toolkit of Lia Fernald and Beth Prado.

Lines 67 – 69. They Bayley III scales are characterized as “extremely expensive, take a long time to administer and require highly trained professionals. This makes them infeasible for use at scale.” This characterization is inaccurate and will result in researchers avoiding direct child assessment in favor of biased caregiver reports. The statements should be revised. The Bayley is expensive only in some countries, notably the US and UK; they do not need to be used in their entirety, and if cognitive and language are used without motor or socio-emotional, the time is cut in half; most researchers in low-income countries do not use professionals, though they are trained to standard. The Bayley is being used at scale when a representative subsample is assessed, rather than all treated children (as in Bangladesh).

Methods

1,533 children living in urban Colombia aged 6-42 months, 1311 of whom received the Bayley along with two or more shorter tests, were followed up when 6 to 8 years of age and tested on the WISC, picture receptive vocabulary, and school achievement. It is hard to believe that children of 6 to 42 months would sit for an 80-minute testing and that children of 6 – 8 years would sit for a 90-minute test with a 5-10 minute break! Fortunately the HOME was used for older children in addition to the Family Care Indicator.

The tests are mainly caregiver report and only under certain conditions is the child directly tested. So the phrase should be “mainly caregiver report and less direct child assessment”. Even the Bayley socio-emotional score is probably the caregiver report; this should be noted as different from the other Bayley scales.

Line 163, 261. The Denver fine motor-adaptive subscale does not really include cognitive items in the usual sense of problem solving, matching and memory -- stacking cubes and scribbling with a pencil don’t reflect cognitive development, which is the important facet of mental development. This is the downside of the Denver; the other being a caregiver report which entails too many potential biases to be used in most nurturing care programs.

Lines from 186 are a repeat of lines 92 – 95.

The inter-observer reliability data suggest that there was no practice effect for children from test to re-test. Is this so?

Results

The tables are useful for comparing correlations of WISC and Achievement with earlier measures.

I did not see correlations of the measures with child’s age and grade level for the older children. These correlations would also address concurrent validity. This is different from partialling age in the analyses (or standardizing within age).

It is problematic not to be able to see all raw scores. In a few cases, raw scores were given but in other cases only internally standardized scores were provided. Raw scores with ranges are useful in knowing the nature of this sample and how it compares with other samples in LMIC. Please provide raw scores for variables in table 1.

Could you present the correlations of height-for-age, not only stunting, with other measures?

Discussion

Any researcher looking at Table 3 would zoom in on the Bayley cognitive subscale as showing the strongest correlations. Some of the others may not be significantly different in correlation size, but they are noticeably lower.

The Discussion para starting line 403 could be more balanced. The Denver lacks a cognitive scale and this reduces its appeal. The suggestion to use only cognitive and language (receptive if children are inhibited) scales is helpful and will simplify the decision for many researchers (did people think otherwise?). This cuts the time to 40 min for the Bayley. Many researchers already use non-professionals with no/little psychology background; this also simplifies the decision. Also, you did not address the important fact that parents participating in a psychosocial stimulation program will be influenced by this and so their reports of child cognition and language will be biased. So caregiver report screeners can be used only in health, nutrition and cash transfer programs. These points need to be raised here.

Line 417. There is a problem with the FCI when used with children 6 to 48 months of age. Bornstein (2015) showed that prevalence for several items asymptotes at a high rate at 4 months of age, and other items like counting and drawing asymptote at a low rate at 4 years. So not all items are relevant for low-income countries at all these ages.

Line 423. FCI is a continuous score and stunting is dichotomized. Why not make both continuous or both dichotomous?

Limitations should be expanded. The Malawi Developmental Assessment measure was excluded here. Limitations of caregiver reports could be expanded. The nature of this sample – urban and involved in programs.

Reviewer #2: The authors have done an interesting study and collected a lot of data in a longitudinal project for which congratulations are in order! Important is also that they collected data on child development, but also on family care indicators,

I have only a few comments on the paper.

The ASQ is developed to be answered by parents, but an interview was used in stead. Did the authors also have some data available to check the reliability of the interview procedure with a sole parent report? Could there be a difference? Could the information also be provided by a day care worker? Perhaps the authors could pay attention to this aspect in their discussion.

No Columbian norms were available for any of the assessments; did all the assessments have norms based on the USA population? Would the authors expect stronger relationships over time when Columbian norms would be available?

An important reason to use early developmental tests or screeners, is to identify children at risk for later developmental problems. The authors state that their study was not designed to result in sensitivity and specificity data, as they did not study subgroups at risk. Indeed, population specific norms are important to base such information on. In the discussion they do make suggestions for combinations of assessments to be used for further pilot studies to acquire population based indicators of poor development. Do they think that the same assessment batteries could provide indicators of poor development in general, even global, populations, or that specific instruments would be neededin subgroups of children at risk?

Minor points

The legenda in Table 1 regarding missing data is not clear: I read as if 477 children had missing data for being stunted: is that correct?

ASQ cognition probably reflect the Problem solving dimension?

Reviewer #3: This manuscript focuses on the important topics of: (1) the lack of measures of children's early childhood development for use in low and middle income countries and (2) the uncertain predictive validity of existing measures, including the inexpensive short form or single domain measures often used in survey research. The study therefore followed a large (> 1,000) sample of children from Bogota Columbia from 6-42 months in 2011 to ages 6-8 in 2016.

Broadly, the study is well designed, executed and reported, although addressing several issue would strengthen the manuscript.

1) The authors note in several places that some measures are “expensive” and others fall in the middle or are “cheapest.” Can numbers be attached to these statements, even if rounded or a range, to help the reader benchmark what is expensive and what is cheap?

2) The Bayley-III is considered the gold standard. To what extent has its reliability and validity for low/middle income and Spanish speaking children?

3) What are the authors considering as “acceptable” predictive validity (p. 6)?

4) Although broadly the authors well describe the study design and implementation, including sample loss, a few things are not clear:

a) How many age/sector strata were there?

b) What defined study eligibility?

c) How many children were eligible?

d) What was the response rate among eligible children? Was this different by strata?

e) Clarifying the above will also help explain the statement in lines 143-145 regarding what seems to be one strata (sector but not age is clear) that was dropped due to low response rate.

f) The above might be incorporated into Figure 1, preceding the n=1,533. Also, are the blocks in Figure 1 synonymous with strata?

f) The authors appear to use the bootstrap methods to adjust for strata (e.g., note to Table 2). Making that clear when the use of strata in the design is presented would help the reader. Also, was there any oversampling within/across strata, suggesting sampling weights would also be needed (Heering, West & Berglund, Applied Survey Data Analysis, Chapman & Hall).

g) As I understand it, the PPVT and TVIP identify the child’s approximate word knowledge and then show words around that target level. It is thus not clear why a subset of words were selected from the TVIP? Is this allowable in the standard instructions for the TVIP, or was permission received from the TVIP developers to make that change and call the scores TVIP scores?

h) Why was principal component rather than factor analysis used? Does the MC-HOME have a standard scoring? If so, why wasn’t that standard scoring used?

i) Were results sensitive to whether raw scores or full sample standardized scores were used, rather than age-standardized scores? What is the conceptual rationale for using age-standardized scores? (This question is returned to below).

j) The authors refer to removing testers/interviewers effects “as done previously” and provide a citation. For the paper to “stand on its own” this adjustment should be briefly described so that it is transparent to the readers of this paper. Again, sensitivity to this adjustment would also be of value.

k) The point on lines 248-250 that the age-standardization process has the advantage of handling age consistently across tests makes sense, but it is not clear how the approach is “less sensitive to small samples.” By the latter do you mean that use calculated the means and SDs non-parametrically? Even if this is the meaning of the statement, it is not clear how norms from reference populations would be more sensitive to small samples, since often norming samples are large and likely as large or larger than your within-age samples. Clarifying the meaning of your statement would help.

l) Analysis within 12-month-of-age ranges is sensible, but as a reader I wanted this plan to be more explicit earlier in the paper, with the age ranges (how many, min/max age) explicitly defined and its rationale and connection to the design offered.

m) The statement on p. 258-260 seems to be combining two ideas, and relates to the request of clarifying why you chose to age-standardize. You seem to be implying that computing correlations within 12-month-age-bands is equivalent to controlling for age. However, the analysis approach assumes correlations are the same within 12-month-age bands and allows them to differ across 12-month-age-bands. It seems there would be a conceptual motivation for looking at analyses within 12-month-age-bands, as discussed below.

n) In lines 270-272 the authors note that sample sizes varied across tests. A table that summarizes the tests, and any sample loss that is built in (due to eligibility for the test) and that is not (due to refusals or data errors) would be helpful. Such a table could also contain the names, acronyms, and study-defined types of the tests (incorporating aspects of the glossary at the start of the paper). If the latter type of test-level missing was extensive in some cases, did the authors examine how those with and without scores differ and consider addressing it with multiple imputation or FIML?

o) What does “Migrants were tracked whenever feasible” mean? (line 281).

p) Table 2 lists “n = 2000 replications” which appears to correspond to bootstrapping. What was the number of children on which each value was based?

q) Did the authors test whether the correlations differed significantly across the age groups in Table 3? Adding confidence intervals would both show the uncertainty of estimation and allow readers to see which estimates overlapped across age groups.

5) The authors might attend to the following in the discussion:

a) The authors provide recommendations to readers about which tests they might use, based on the results. Doing so is helpful, but as a reader I felt at times the recommendations focused on relative rather than absolute predictive validity (e.g., p. 405-406). The authors might remind the reader that the predictive validity is low-to-moderate.

b) In lines 417-422, the authors might be careful to acknowledge that the FCI may proxy for other aspects of household or community resources that would be the better proximal cause for intervention, given this study is simply focusing on correlations.

6) Figure 2 does not well “stand on its own”

a) The acronyms should be defined in the note (or refer the reader to where to find the definitions).

b) The Y axes are not labelled.

c) The n’s on which the values in the Figure are based should also be clear either in each the legends/titles/axes or the table notes (or tell the reader where to find the n’s in another table that explicitly adds and lists them).

d) The Y axis or table note should also make clear what is being correlated with what: that each correlation is between the kids in the younger age group with their own scores 5 years later. For that reason, it would be preferable on the X axis to list the follow-up along with the initial age-range, as those together define the meaning of the correlation (e.g., B: 6-18 m, F: ~ 6y; B: 19-30 m, F ~ 7y; B: 31-42 m, F ~ 8y).

7) This point just noted about labelling the X axis relates to a broader conceptual question that should receive more attention throughout the paper. The study design starts with children in three age bands, centered on age 1, age 2, and age 3. These children are followed up 5 years later at ages 6, 7, and 8, respectively. Thus the size of the correlations depend on both how well the measures used in early childhood capture each latent construct at those early ages, and how well the measures used in middle childhood capture each latent construct at those later ages. In other words, the lower correlations for the “6-18 months” group may to some degree reflect how well the middle childhood tests measure outcomes for 6 year olds versus 8 year olds. If the study could have followed children to age 8, the meaning would have been further strengthened. Presumably that wasn’t done in 2018, but the meaning of the values presented should be clear (as in the labels of Figure 2) and the rationale for the design and its interpretation should be clearer throughout the paper. More attention to the reliability and validity of the middle childhood measures (including the WISC, but also school achievement - are 6, 7, and 8 year olds all in schools of similar type in this context?) for these ages, and especially in lower/middle income countries and for Spanish speaking children, would be of value.

6. PLOS authors have the option to publish the peer review history of their article (what does this mean?). If published, this will include your full peer review and any attached files.

Reviewer #1: No

Reviewer #2: Yes: Anneloes van Baar

Reviewer #3: No

---

## [Author Response · Author response to Decision Letter 0]

17 Oct 2019

PONE-D-19-20029

Predictive validity in middle childhood of short tests of early childhood development (ECD) used in large scale studies compared to the Bayley-III, the Family Care Indicators (FCI), height-for-age and stunting: A longitudinal study in Bogota, Colombia

Responses from the authors to the academic editor and reviewer’s comments 

We thank the editor and the reviewers for their valuable and detailed comments, which have substantially improved the clarity and flow of the Manuscript.

We respond below to the reviewers’ comments. For clarity we retain the comments and add our response in italics. We have numbered the reviewers’ comments to facilitate cross-references to our responses. Please note that the references to the line numbers were edits have been made correspond to the version of the Manuscript without tracked changes (clean version). 

Academic Editor Comments to the Authors

Please ensure that your manuscript meets PLOS ONE's style requirements, including those for file naming. The PLOS ONE style templates can be found at: http://www.journals.plos.org/plosone/s/file?id=wjVg/PLOSOne_formatting_sample_main_body.pdf and http://www.journals.plos.org/plosone/s/file?id=ba62/PLOSOne_formatting_sample_title_authors_affiliations.pdf

Response: Thanks for the links to the guidelines. We have carefully reviewed the manuscript and first page to ensure PLOS ONE’s style requirements are met. To facilitate the reading of the final Manuscript with tracked changes, we have not tracked the formatting changes we have made. 

1. We noticed you have some minor occurrence of overlapping text with the following previous publication, which needs to be addressed: Rubio-Codina, Marta, et al. "Concurrent validity and feasibility of short tests currently used to measure early childhood development in large scale studies." PLoS One 11.8 (2016): e0160962. In your revision ensure you cite all your sources (including your own works), and quote or rephrase any duplicated text outside the methods section. Further consideration is dependent on these concerns being addressed.

Response: We have contrasted the revised manuscript against the publication mentioned for duplicated text outside the Methods section and either cited or quoted/rephrased any duplication found, as suggested. Thanks for the suggestion.

Response: We understand that the manuscript will be held until the relevant accession numbers to access the data are provided and agree with this procedure. Therefore, no changes are needed to the Data Availability Statement. 

Reviewer Comments to the Author

Reviewer #1: PONE-D-19-20029. Predictive validity in middle childhood of short tests of 1 early childhood development (ECD) used in large scale studies compared to the Bayley-III, the Family Care Indicators and stunting: A longitudinal study in Bogota, Colombia.

1. Abstract first sentence and Line 60. It is not clear why interventions are hindered by large-scale measurement of child development. This implies that large-scale interventions require that all children be tested, rather than a representative sample, as is usual. Otherwise, the most convincing rationale for this study was the value in having population-level indicators. Perhaps reference to evaluations of programs can be deleted.

Response: We certainly agree with the need to identify cost-effective tools that can produce population-level indicators of child development. There are efforts underway to this end (see references in line 50 in the clean version of the Manuscript, line 66 in the old version of the Manuscript). Nonetheless, in large scale studies, testing even a representative sample of the population of interest can be expensive and, in our and other colleagues’ experience, it has limited evaluation efforts. Currently, each Bayley-III kit costs $1,159 USD and each pack of 25 record forms costs $138.75 USD (or 5.55 USD per form; this is, 5.55 USD per child tested as one form is needed for each child). In addition, one would have to add the cost of translation of the test forms and manuals (the test is available to be purchased in English and Spanish ; but not in any other language), the cost of adapting the test and any test materials, as well as the cost of piloting the adaptations. 

A further problem is that the Bayley Scales require more skilled testers than the short tests and longer training; and people with this level of skills are usually not readily available in poor or remote areas so that the testers not only require higher salaries but also need to travel further, which results in higher costs (honoraria, travel and per diems of the testers). 

The World Bank 2017 measurement toolkit concurs with these points: “administration of these tests requires a certain level of technical expertise and may be cumbersome for assessments that require a lot of manipulatives (e.g., Bayley Scales of Infant Development).” (Fernald, Prado, Kariger, & Raikes, 2017, p. 60). The authors of the toolkit also refer to time, cost, and copyright issues as limitations of the Bayley-III for administration in the context of program evaluations and hypothesis-driven research (see p. 65). Therefore, considering the above, we prefer to keep the statement about the Bayley-III being not feasible to use at scale, but have amended it to focus on the evaluation: “However, evaluation is hindered by the lack of valid developmental tests feasible for use at large scale” (lines 18-19) and “However, evaluation of these efforts is impeded by the lack of reliable, valid, and easy-to-collect measures of ECD, particularly for children under age three years” (lines 43-45).

We are not implying that all children in an intervention need to be tested. However, in large scale programs a considerable number of children need to be tested especially where there is heterogeneity in the population. As an example, the sample size for the impact evaluation of Cuna Más, the nationwide home visiting program in Peru, currently covering over 93,000 families in rural areas, was of 5,859 children 1-24 months of age in 180 districts (municipalities) (Araujo et al. 2019). These numbers were informed by rigorous power calculations. The size of this representative evaluation sample and other costs/limitations associated with the administration of the Bayley-III did prevent the government from using the test: the per child administration fee alone (this is, the cost of the record forms) would have amounted to $32,224.5 USD! The ASQ-3 was used instead for the representative evaluation sample. Owing to space restrictions, we have not added this example in the Abstract. 

2. The data do not really support the first part of the final statement, namely that the FCI is an appropriate indicator for evaluations of programmes promoting mental development. However, I do agree with the second part, that the FCI and stunting together could be used as population-level proxies of mental development.

Response: Thanks for pointing this out. Suggesting that the FCI could be used alone for evaluation was an oversight. We had already indicated in the Conclusions (previous lines 446-448, current lines 527-529), that we are suggesting piloting the use of the FCI in combination with the language and cognitive scales of the Denver-II for large ECD evaluations. However, we have now dropped the statement from the Abstract and left it in the Discussion as a hypothesis to be examined (lines 487-489): “We hypothesize that adding the FCI to the Denver-II cognitive and language scales as suggested above might improve the sensitivity of a program evaluation and needs to be investigated elsewhere”. 

Introduction

3. Instead of the #2 Frongillo reference (are any of these authors experts in educational/psychological measurement of child development?) you might include the 2017 World Bank toolkit of Lia Fernald and Beth Prado.

Response: The reference has been changed to the one suggested.

4. Lines 67 – 69. They Bayley III scales are characterized as “extremely expensive, take a long time to administer and require highly trained professionals. This makes them infeasible for use at scale.” This characterization is inaccurate and will result in researchers avoiding direct child assessment in favor of biased caregiver reports. The statements should be revised. The Bayley is expensive only in some countries, notably the US and UK; they do not need to be used in their entirety, and if cognitive and language are used without motor or socio-emotional, the time is cut in half; most researchers in low-income countries do not use professionals, though they are trained to standard. The Bayley is being used at scale when a representative subsample is assessed, rather than all treated children (as in Bangladesh).

Response: Please see our response to Comment 1 above by Reviewer 1 re the cost of administering the Bayley-III. Moreover, please note that the use of the Bayley-III remains expensive outside the US and the UK since one must not only buy the kits and record forms from the publisher—either in the US or the UK, and pay for transportation costs and customs costs in addition; or from the local branches (often a cumbersome process)—but also to additionally cover all translation and adaptation costs. Any translations/backtranslations/adaptations made are paid by the researchers/users and checked by the publisher, who retains their ownership. This is ‘equivalent’ to a translation fee. Furthermore, any future use of a previously translated test, even by the same researchers who carried out the translation, requires the payment of a usage fee. 

Given the number of manipulatives required and the complexity in administering the test, it is consistently recommended to use testers with “a certain level of technical expertise” to administer tests like the Bayley-III, both by experts (Fernald et al., 2017, p. 65) and by the test developer (see Bayley-III administration manual, Bayley, 2006). In the current study, we took 6 weeks to train 6 recent graduates in Psychology to standard on the Bayley-III (and paid them during the entire training period). 

Therefore, we have marginally edited the sentence – see current lines 50-53, which now read “Whilst existing diagnostic tests such as the Bayley Scales [5,6] are sensitive to ECD interventions [2,7–9], they are very expensive, take a long time to administer and require highly trained testers with a certain level of technical expertise [2]. These aspects make their use at scale very difficult [10].”

We do agree with the suggestion of using only cognitive and language scales when appropriate. But, with the Bayley-III, this might still involve 30-45 minutes of test, depending on the child’s age, ability and collaboration, which is likely to still be too long for large scale studies. We have already suggested using these scales but from the Denver-II rather than the Bayley-III (previous lines 447-448, current lines 662-665), which have not been modified. 

We have now added the cost of the tests in the Enrollment assessments to substantiate the claim of the Bayley-III being expensive and in response to this Reviewer and to Reviewer 3, Comment 1: “The Bayley-III was the most expensive test ($1,025 US per kit; $4.89 US per child fee, at the time of assessment) and its administration took 83 minutes on average. The BDI-2 took 63 minutes and was also expensive ($405.70 US per kit; $3.08 US per child fee), whilst the Denver-II (27 minutes) and ASQ-3 (20 minutes) were intermediate both in time and cost (Denver-II: $200 US per kit; $0.45 US per child fee and ASQ-3: $275 US per kit; no per child fee). As expected, the single-domain tests were quickest (6-8 minutes) and cheapest (SFs: $90 US per kit including both forms, $1 US per child fee; WHO: free).”, lines 197-203.

Methods

5. 1,533 children living in urban Colombia aged 6-42 months, 1311 of whom received the Bayley along with two or more shorter tests, were followed up when 6 to 8 years of age and tested on the WISC, picture receptive vocabulary, and school achievement. It is hard to believe that children of 6 to 42 months would sit for an 80-minute testing and that children of 6 – 8 years would sit for a 90-minute test with a 5-10 minute break! Fortunately the HOME was used for older children in addition to the Family Care Indicator.

Response: Children were allowed as many breaks as required during the administration of the Bayley-III and of the WISC-V/school achievement tests, especially in early childhood (Bayley-III administration). On a few occasions, the test could not be completed in one sitting and had to be suspended and resumed later in the day or on another day. In middle childhood, 90 minutes was the maximum testing time overall and over half the sample completed the assessments in 76 minutes or less. The time was recorded by the testers and 2% of the tests were observed by a supervisor. We therefore have no reason to disbelieve the data.

6. The tests are mainly caregiver report and only under certain conditions is the child directly tested. So the phrase should be “mainly caregiver report and less direct child assessment”. Even the Bayley socio-emotional score is probably the caregiver report; this should be noted as different from the other Bayley scales.

Response: Actually, many of the short tests administered in early childhood included direct testing of children. Of the short tests only the vocabulary checklists in the SFI and SFII, and the ASQ-3 are intended to be entirely by caregiver report. However, unlike in the original test, items in the ASQ-3 were directly administered to the child whenever the caregiver could not know the answer (as explained in previous lines 165-159, current lines 171-174). All other tests were administered as indicated in the test manuals (as noted in previous lines 164-165, current lines 170-171). The Denver-II is mostly by direct administration and a minority of items, up to 39%, mostly in the personal-social and language scales may be passed by caregiver report (Frankenburg et al., 1990). Similarly, the BDI-2 is mostly administered by direct assessment on the child, although a limited number of items can be collected by caregiver report or by tester observation (Newborg, 2005). Finally, the WHO-Motor was administered on the child exclusively. With regards to the Bayley-III, and as indicated in the manual, yes, the socio-emotional scale is by caregiver report (Bayley, 2006). We have clarified all of the above in the Manuscript as it was not sufficiently well explained (in part, to avoid duplication with the earlier study, Rubio-Codina et al., 2016). The relevant paragraph (lines 166-181) now reads: “The ASQ-3, Denver-II and BDI-2 screeners were multi-dimensional and covered the entire age range. They all combined receptive and expressive language in one communication/language scale. Similarly, the BDI-2 motor scale combined fine and gross motor items; and the Denver-II fine-motor adaptive scale included both fine motor and cognitive items. The ASQ-3 problem solving scale comprises cognition. As reported earlier [10], all tests were administered following manual instructions except the ASQ-3, which we modified as follows. Given the low literacy levels of some caregivers, caregiver-completed items were administered by interview in order to ensure all mothers understood the questions similarly. Furthermore, the child was tested if the caregiver did not know the answer. In addition, whenever the scale ceiling was reached, we added the next three non-overlapping items from subsequent questionnaires. This reduced the number of children on the test ceiling by 10.5–15.5% to levels of 1.7–4.8%, depending on the domain, thus increasing the variability in development captured by the test. Similar adaptations have been used elsewhere [11]. Following test manuals, the Denver-II and BDI-2 were mostly collected by direct child assessment; although up to 39% of the items in the Denver-II (mainly in the personal-social and language scales) can be obtained by caregiver report and a few BDI-2 items may also be collected by caregiver report or tester observation.”

7. Line 163, 261. The Denver fine motor-adaptive subscale does not really include cognitive items in the usual sense of problem solving, matching and memory -- stacking cubes and scribbling with a pencil don’t reflect cognitive development, which is the important facet of mental development. This is the downside of the Denver; the other being a caregiver report which entails too many potential biases to be used in most nurturing care programs.

Response: Thanks. We say in the paper that the Denver-II fine motor-adaptive scale comprises some fine motor and cognitive items. We think there is evidence supporting the claim that it contains cognition. The Denver-II manual describes the fine motor-adaptive scale as including the “following areas of function: eye-hand coordination, manipulation of small objects, and problem-solving” (Frankenburg et al., 1990, p. 1). Moreover, there are several items in the Denver-II scale that are very similar to items in the Bayley cognitive scale. For example, “look for yarn that falls out of sight”, “takes 2 cubes” (holding one in each hand), “removes pellet”. In prior analysis of concurrent validity (Rubio-Codina et al., 2016), we found somewhat higher correlations between the Denver-II fine motor-adaptive scale and the Bayley-III cognitive scale than with the Bayley-III fine motor scale in children younger than 30 months. We also show, in the current paper, that the scale correlates well with later IQ, better or similar to the cognitive scales of the ASQ-3 or the BDI-2 and only slightly less well than the Bayley-III cognitive scale. For example, the Bayley-III cognitive score in the age range 31-42 months correlates 0.474 and 0.436 with later IQ and school achievement, respectively, compared with 0.422 and 0.438 for the Denver-II fine motor-adaptive scale. Therefore, we used the Denver-II fine motor-adaptive scale for both cognition and fine motor analyses. We have explained this in the Statistical Analysis section lines 282-287 and in the Discussion as a limitation, lines 508-512. These paragraphs read:

Lines 282-287: “We used the Denver-II fine motor-adaptive scale for both cognition and fine motor analyses, as there is some evidence the scale includes both fine motor and cognitive items. Similar items to some of the Denver items appear in the Bayley-III cognitive scale, and in prior analysis of concurrent validity [10], somewhat higher correlations were found between the Denver-II fine motor-adaptive scale and the Bayley-III cognitive scale than with the Bayley-III fine motor scale in children under 30 months.”

Lines 508-512: “Another problem is that the Denver-II does not have a separate cognitive scale per se. However, the fine motor-adaptive scale combines cognitive and fine motor items and we have shown that it correlated well with later IQ, better or similar to the cognitive scales in other short tests and similar to the Bayley-III cognition for children 19 months and older.”

Finally, as explained in the response to Comment 6 of Reviewer 1, the Denver-II is not primarily a caregiver report test and only a minority of items accept caregiver report. 

8. Lines from 186 are a repeat of lines 92 – 95.

Response: Thanks for pointing this out. We have edited both paragraphs to avoid repetition. They now read: 

Lines 77-81: “The Bayley-III was found to be the most expensive and longest test to give. Whilst also long and expensive, the BDI-2 took, on average, 20 minutes less to administer than the Bayley-III; and the Denver-II and the ASQ-3 took a third of the time or less, being intermediate in time and cost. The single-domain tests were quickest, taking no more than 8 minutes, on average, and were considerably less expensive (the WHO-Motor was free).”

Lines 197-203: “The Bayley-III was the most expensive test ($1,025 US per kit; $4.89 US per child fee, at the time of assessment) and its administration took 83 minutes on average. The BDI-2 took 63 minutes and was also expensive ($405.70 US per kit; $3.08 US per child fee), whilst the Denver-II (27 minutes) and ASQ-3 (20 minutes) were intermediate both in time and cost (Denver-II: $200 US per kit; $0.45 US per child fee and ASQ-3: $275 US per kit; no per child fee). As expected, the single-domain tests were quickest (6-8 minutes) and cheapest (SFs: $90 US per kit including both forms, $1 US per child fee; WHO: free).”

9. The inter-observer reliability data suggest that there was no practice effect for children from test to re-test. Is this so?

Response: Test-retest reliabilities after 6-14 days reached the usually found acceptable levels: we have added the range in lines 325-326: “Test-retest reliabilities after 6-14 days were also good ([ICC =0.39-0.88], ICC > 0.6 for most tests).” In 16 subjects, each given the 7 subtests of the WISC-V, 77 (68.8%) improved, 13 (11.6%) obtained the same score, and 22 (19.6%) decreased. This indicates a small practice effect, as expected, although it varied by subtest. Similarly, with regards to the 3 tests composing the achievement score, for the 16 subjects, on 22 occasions (45.8%) the score improved, on 4 (8.3%) it remained the same, and on 22 (45.8%) it decreased. We have not updated the text with these details, owing to space restrictions.

Results

10. The tables are useful for comparing correlations of WISC and Achievement with earlier measures.

I did not see correlations of the measures with child’s age and grade level for the older children. These correlations would also address concurrent validity. This is different from partialling age in the analyses (or standardizing within age).

Response: Thanks for this suggestion. We have added the correlations of the raw scores of the components of the FSIQ and Achievement scores in Supplementary Table S2. All correlations are highly significant (all P <0.000). Correlations of the achievement subtests with age and grade are generally larger than those of the seven subtests that compose the FSIQ in the WISC-V. Note that there is no raw score for the FSIQ score as such, as, by construction, it is the sum of the externally standardized scores of the subtests. Similarly, by construction, there is no raw score of the achievement score. 

We have added a reference to Supplementary Table S2 in the Manuscript in lines 322-323: “Raw scores of all middle childhood tests and subtests, reported in S1 Table, increased with age and school grade (S2 Table).”

11. It is problematic not to be able to see all raw scores. In a few cases, raw scores were given but in other cases only internally standardized scores were provided. Raw scores with ranges are useful in knowing the nature of this sample and how it compares with other samples in LMIC. Please provide raw scores for variables in table 1.

Response: Thanks also for this suggestion. We report raw scores of the Bayley-III, the components of the FSIQ in the WISC-V, and the components of the achievement score in Supplementary Table S1. We provide means and SDs for the entire sample and for children in battery A and in battery B, alongside a test of the difference between batteries. Minimum and maximum values are also reported for all children. 

We have added a reference to Supplementary Table S1 in the Manuscript in lines 322-323: “Raw scores of all middle childhood tests and subtests, reported in S1 Table, increased with age and school grade (S2 Table).”

Given this new table, we have removed all raw scores from Table 1 in the Manuscript to avoid duplicating the information. 

12. Could you present the correlations of height-for-age, not only stunting, with other measures?

Response: We have added height-for-age to the analysis (please see Tables 1 and 3). We had performed the analysis earlier and debated whether to report it or not. We decided against, as results are qualitatively the same as those from stunting, but we agree with the Reviewer and have now reported both measures. Thanks for the suggestion! 

Discussion

13. Any researcher looking at Table 3 would zoom in on the Bayley cognitive subscale as showing the strongest correlations. Some of the others may not be significantly different in correlation size, but they are noticeably lower.

Response: The study was based on the idea that the Bayley-III was the gold standard and, as expected, the Bayley-III has the highest predictive validity and we have already pointed this out (see line 340, previous lines 306-307) and reiterated it in the first line of the Discussion (line 429, previous line 377). The statement was also one of our starting hypotheses (see lines 112-113, previous lines 120-121). So we think the point is already covered. We were however surprised by how relatively good the correlations were with some of the other tests. 

14. The Discussion para starting line 403 could be more balanced. The Denver lacks a cognitive scale and this reduces its appeal. The suggestion to use only cognitive and language (receptive if children are inhibited) scales is helpful and will simplify the decision for many researchers (did people think otherwise?). This cuts the time to 40 min for the Bayley. Many researchers already use non-professionals with no/little psychology background; this also simplifies the decision. Also, you did not address the important fact that parents participating in a psychosocial stimulation program will be influenced by this and so their reports of child cognition and language will be biased. So caregiver report screeners can be used only in health, nutrition and cash transfer programs. These points need to be raised here.

Response: We have addressed the point that we think the fine motor-adaptive scale in the Denver-II has a cognitive component in our response above to Reviewer 1, Comment 7. 

The use of non-professionals with no/little psychology background requires extensive training with lots of practice tests. This has implications for costs. Although using non-professionals is certainly feasible, it is likely to pose a risk to accuracy for long and complex tests that require a lot of manipulatives such as the Bayley-III (Fernald, Prado, Kariger, & Raikes, 2017). It is generally recognized that tests as complex as the Bayley require a tester with higher levels of education than that required by the short tests, which also has cost implications.

We agree that an important point is that maternal report is likely to be biased where mothers have been involved in a stimulation program. However, as explained above in our response to Reviewer 1, Comment 6, the Denver-II is mostly by direct administration (Frankenburg et al., 1990) and hence any potential bias should be limited. However, we have inserted a comment on the potential bias of maternal recall as a special problem when the caretakers have been involved in parenting interventions in the Discussion, which reads: “Caregiver reports might be better suited for the evaluation of population-based indicators and less convenient to evaluate psychosocial stimulation programs as they may suffer from recall bias. Moreover, where mothers have been involved with the intervention, they may be biased towards making optimistic claims of their children’s development.”, lines 471-475.

15. Line 417. There is a problem with the FCI when used with children 6 to 48 months of age. Bornstein (2015) showed that prevalence for several items asymptotes at a high rate at 4 months of age, and other items like counting and drawing asymptote at a low rate at 4 years. So not all items are relevant for low-income countries at all these ages.

Response: We did not evaluate the use of any instrument for the 2 ages mentioned (4 months and 4 years). 

Bornstein showed that caregiving practices vary substantially across countries and child ages. However most psychosocial interventions improve stimulation in the home (Hamadani et al., 2019; Attanasio et al., 2014; Tofail et al., 2013; Walker et al., 2004), and it has been shown to mediate child development effects in some cases (Attanasio et al 2019). Obviously, more research is necessary for use in all countries. We have made this point in lines 487-489: “We hypothesize that adding the FCI to the Denver-II cognitive and language scales as suggested above might improve the sensitivity of a program evaluation and needs to be investigated elsewhere.”; and lines 495-497: “Finally, the above findings could be extrapolated to urban Colombia and possibly urban areas in other Latin American countries. Further studies would be required before extrapolating to rural areas and other LMICs, especially in Africa and Asia”.

16. Line 423. FCI is a continuous score and stunting is dichotomized. Why not make both continuous or both dichotomous?

Response: Thanks for the suggestion. We have added height-for-age to the analysis. Nonetheless, we prefer to keep stunting as well since this is the indicator that has been repeatedly used as a global ECD proxy indicator (Black et al., 2017; Grantham-McGregor et al., 2007), is internationally accepted, and we were interested in recommending the least change from what it is currently used. Furthermore, stunting is presently defined as below -2 SD of a healthy well-fed population. We would not be able to determine a cut-off for the FCI to define poor stimulation, analogous to that used for stunting, because we have no data on the normal distribution of the FCI from a relatively advantaged population. We have not discussed this in the paper, owing to space restrictions. 

17. Limitations should be expanded. The Malawi Developmental Assessment measure was excluded here. Limitations of caregiver reports could be expanded. The nature of this sample – urban and involved in programs.

Response: The Malawi Development Assessment Tool (MDAT) was about to be published when the study began and was not well known (Gladstone et al., 2010). We agree it would be useful to evaluate tests currently being used in Africa in addition to the ones we examined. However, time and budgetary constraints prevented us from doing so. There are several other tests that are now used that we were unable to evaluate. 

We have added a comment on the external validity being limited at best to urban Latin America, which was not well highlighted previously in lines 495-497: “Finally, the above findings could be extrapolated to urban Colombia and possibly urban areas in other Latin American countries. Further studies would be required before extrapolating to rural areas and other LMICs, especially in Africa and Asia”.

However, the study sample was not involved in programs and was not a convenience sample in any way. Rather, it was a sample representative of the poorest (first) three out of the six socio-economic sectors (strata) in which the city of Bogota is divided. These sectors account for 85% of the city’s population and comprise low- and lower-middle-income households. To ensure representativeness, neighborhoods and blocks were selected by random probability, stratifying by sector. Within blocks, children were identified by a door-to-door census and selected for study inclusion by random probability, stratifying by age. This is explained in current lines 130-137, which have been revised for clarity, and now read “The study was conducted in the poorest three, out of six, socio-economic sectors (‘estratos’ in Spanish) of Bogota. These three sectors, defined by location and the quality of housing and infrastructure, comprise low- and lower-middle-income households and account for 85% of the city’s population. In 2011, we enrolled 1,533 children aged 6-42 months living in blocks in these sectors. Within each sector stratum, blocks were selected by random probability, weighting by the proportion of women in fertile age. Within each selected block, all children 6-42 months were identified through a door-to-door census and a subsample was randomly selected for study inclusion, stratifying by age.”

Reviewer #2: 

The authors have done an interesting study and collected a lot of data in a longitudinal project for which congratulations are in order! Important is also that they collected data on child development, but also on family care indicators, I have only a few comments on the paper.

1. The ASQ is developed to be answered by parents, but an interview was used instead. Did the authors also have some data available to check the reliability of the interview procedure with a sole parent report? Could there be a difference? Could the information also be provided by a day care worker? Perhaps the authors could pay attention to this aspect in their discussion.

Response: Thanks for raising this point. Unfortunately, we have no data to compare the performance of the ASQ-3 by parental interview vs. parental-completion. Face-to-face interviews may increase certain biases, such as desirability or acquiescence (tendency to say ‘yes’) biases. They may also increase recall biases since parents have less time to remember what the child does or does not do in the context of an interview. At the same time, administration of the test by interview is likely to improve the comprehension of the items (by reducing ambiguity in the interpretation of what each item enquires about) and to increase the comparability of response across the sample, provided that the interviewers/testers had been adequately standardized during the training. We are not commenting on these biases in the Manuscript since we do not have any data to support these hypotheses and owing to space restrictions.

In the study, however, the low literacy level of some caregivers made it difficult for them to complete the form independently. We therefore preferred to give the ASQ-3 by interview to all participants in order to standardize the administration. This situation is not uncommon in LMICs. We have clarified this point in lines 171-173, which now read: “Given the low literacy levels of some caregivers, caregiver-completed items were administered by interview in order to ensure all mothers understood the questions similarly”. 

The ASQ-3 developers encourage the use of the tool by childcare programs and report on its suitability for use by childcare professionals. However, in LMICs a large proportion of young children do not attend childcare centers. In our sample, for example, the average childcare attendance rate is 31.13%, increasing from levels of about 4.1% for children under 12 months, to 18.5% for children between 1 and 2 years, to levels of 47.5% for children over 2 years. Therefore, for programmatic evaluation and population-level reporting in LIMCs, we would suggest assessment of child development in the home environment and not at the childcare center. For simplicity, we are not commenting on this point in the Manuscript either.

2. No Columbian norms were available for any of the assessments; did all the assessments have norms based on the USA population? Would the authors expect stronger relationships over time when Columbian norms would be available?

Response: The Bayley-III and all short tests considered have norms for the US population, except for the WHO-Motor. However, these norms are all based on different norming samples and use different norming strategies—this is, each one of the tests adjusts for age in different ways. Some of the norming strategies used produce externally standardized scores and some (particularly for the screener tests) produce cut-off scores, which are used to identify children at risk for developmental delay. Therefore, as noted in the Manuscript (lines 418-419), an advantage of using internally standardized scores for the analysis as done is that it handles age effects consistently across all short tests, the Bayley-III and the FCI, which facilitates the comparison of correlations between pairs of tests. 

It is difficult to say whether correlations would be stronger if Colombian norms were available for all tests analyzed, with the data available. If Colombian norms were to be constructed using the same sample and methodology for all tests, it is likely that results would remain largely unaffected since the process of constructing the Colombian norms would be very similar to the one currently performed—namely, to standardize all tests in a similar fashion. From the robustness tests we have carried out using externally standardized scores from the FSIQ, we know that results are little altered (lines 423-427, previous lines 372-375). 

We have not modified the text further with regards to this point. 

3. An important reason to use early developmental tests or screeners, is to identify children at risk for later developmental problems. The authors state that their study was not designed to result in sensitivity and specificity data, as they did not study subgroups at risk. Indeed, population specific norms are important to base such information on. In the discussion they do make suggestions for combinations of assessments to be used for further pilot studies to acquire population based indicators of poor development. Do they think that the same assessment batteries could provide indicators of poor development in general, even global, populations, or that specific instruments would be needed in subgroups of children at risk?

Response: This is an important question, but our data cannot answer it: as rightly noted, the number of children at risk for any disability in the sample is too low to draw any conclusion. A separate study would be needed to determine whether these tests could be used to identify high-risk children in LMICs and it is likely that they could, as several were specifically designed for screening. We have clarified this in the Introduction, lines 118-120: “The study was neither designed nor powered to examine the sensitivity or specificity of the screeners in identifying high risk children [10]. Furthermore, obviously disabled children were excluded as well as children with Bayley scores below 70. Therefore, our results have no implications regarding the use of screeners to identify potentially high-risk children in need of further assessment.” and in the Discussion, lines 498-503: “An important point is that some of the short tests have been shown to have good sensitivity and specificity when identifying children in LMICs at high-risk of disability [54]. By design, our study did not address this issue: the study sample was insufficient, not representative of high-risk populations (e.g. premature or low birthweight children), and obviously disabled children were excluded [10]. Therefore, our findings and recommendations are not generalizable to these subpopulation groups.”

Minor points

4. The legend in Table 1 regarding missing data is not clear: I read as if 477 children had missing data for being stunted: is that correct? 

Response: Thanks very much for pointing this out. nA=477 is the number of children in Battery (subsample) A with information on height-for-age. There is actually only one child with missing information. The legend was indeed unclear and has been revised and it now reads: “a Missing data for some variables. Sample sizes for these are: height-for-age and stunted (nA=477); mother's age (nA=453, nB=433); father's education (nA=456, nB=424).” in lines 319-320.

5. ASQ cognition probably reflect the Problem solving dimension?

Response: Yes, it does. We have clarified this in the Materials and methods section: “The ASQ-3 problem solving scale comprises cognition” (lines 169-170). Thanks. 

Reviewer #3: 

This manuscript focuses on the important topics of: (1) the lack of measures of children's early childhood development for use in low and middle income countries and (2) the uncertain predictive validity of existing measures, including the inexpensive short form or single domain measures often used in survey research. The study therefore followed a large (> 1,000) sample of children from Bogota Columbia from 6-42 months in 2011 to ages 6-8 in 2016.

Broadly, the study is well designed, executed and reported, although addressing several issue would strengthen the manuscript.

1) The authors note in several places that some measures are “expensive” and others fall in the middle or are “cheapest.” Can numbers be attached to these statements, even if rounded or a range, to help the reader benchmark what is expensive and what is cheap?

Response: We have added details on purchase costs in the Procedures section, lines 197-203, which now read: “The Bayley-III was the most expensive test ($1,025 US per kit; $4.89 US per child fee, at the time of assessment) and its administration took 83 minutes on average. The BDI-2 took 63 minutes and was also expensive ($405.70 US per kit; $3.08 US per child fee), whilst the Denver-II (27 minutes) and ASQ-3 (20 minutes) were intermediate both in time and cost (Denver-II: $200 US per kit; $0.45 US per child fee and ASQ-3: $275 US per kit; no per child fee). As expected, the single-domain tests were quickest (6-8 minutes) and cheapest (SFs: $90 US per kit including both forms, $1 US per child fee; WHO: free).”

Further details on purchase and administration costs, as well as on the requirements to train all tests investigated, were provided in prior work (Rubio-Codina et al., 2016, Table 1). We had limited the inclusion of such details in this paper to avoid duplication.

2) The Bayley-III is considered the gold standard. To what extent has its reliability and validity for low/middle income and Spanish speaking children?

Response: The Bayley Scales have been widely used in research studies in low- and middle-income countries among very different populations. Many of these studies report on the performance (reliability and validity) of the test within the population of interest, as we had previously done for the current study (Rubio-Codina et al., 2016; Rubio-Codina et al. 2015). However, as far as we know, there is no formal study of the Bayley-III on a representative population of a country for the development of norms outside of the US. Since March 2015, an official version of the Bayley-III can be bought in Spanish from the publisher, Pearson. Yet, it is unclear from the web description whether this version has been normed in a Spanish speaking population or whether it is a mere translation (most likely, the latter). In this study, we used the Bayley-III as our gold standard given that it is one of the most commonly used tests for the assessment of development for children 1-42 months-of-age and the Bayley Scales have been used in more than 25 LMICs (Fernald et al., 2017; Frongillo, Tofail, Hamadani, Warren, & Mehrin, 2014). 

3) What are the authors considering as “acceptable” predictive validity (p. 6)?

Response: The acceptable predictive validity would vary by initial and later age. We have removed the word acceptable from previous line 117 and have added the exact correlations reported in the paper referenced instead. The paragraph now reads (lines107-110): “Similarly, a maternal report language test for children 12-18 months, developed locally and based on the MacArthur-Bates Communicative Development Inventories [29], significantly predicted IQ at age 5 (r =[0.37-0.41]) [30]”. Thanks for identifying this imprecision. 

4) Although broadly the authors well describe the study design and implementation, including sample loss, a few things are not clear:

a) How many age/sector strata were there?

Response: Initially, we selected 4 socio-economic sector strata (the 4 lowest out of 6), although data were only collected on children from the first 3 sectors (except for 12 children from sector 4). There were 4 age strata: children 6-14 months, children 15-23 months, children 24-32 months, and children 33-42 months. We have clarified this in lines 137-139, which now read: “Strata sizes (originally, 4 sector strata and 4 age strata: 6-14, 15-23, 24-32 and 33-42 months-of-age) were computed to allow for detection of differences in child development between socio-economic sector and age groups.”

b) What defined study eligibility?

Response: Children were eligible if:

(i) They were 6-42 months of age.

(ii) They lived in one of the selected blocks from one of the selected neighborhoods. Both neighborhoods and blocks were randomly selected, weighting by the proportion of women in fertile age and stratifying by sector. All neighborhoods and blocks were in sectors 1 to 4 (out of 6). However, due to high refusal rates, sector 4 (middle income) was dropped from the analysis. Nonetheless, the 12 children living in sector 4 who had already been enrolled in the study were kept in the analysis. 

(iii) They had no obvious disabilities. Note that 1.1% of children were later dropped if they had a Bayley score <70. 

(iv) Singletons only.

If in one household, more than one child was eligible, one of them was randomly selected for study inclusion. We have slightly re-written the Study design and participants section to clarify these criteria: “In 2011, we enrolled 1,533 children aged 6-42 months living in blocks in these sectors. Within each sector stratum, blocks were selected by random probability, weighting by the proportion of women in fertile age. Within each selected block, all children 6-42 months were identified through a door-to-door census and a subsample was randomly selected for study inclusion, stratifying by age. Strata sizes (originally, 4 sector strata and 4 age strata: 6-14, 15-23, 24-32 and 33-42 months-of-age) were computed to allow for detection of differences in child development between socio-economic sector and age groups. The final study sample included 12 children from a fourth sector (middle-income), initially included in the study but subsequently dropped due to high refusal to study participation. Twins and children with obvious disabilities were excluded; and in households with more than one eligible child, one was randomized into the study.” In lines133-143. Criterion (iii) has specifically been included in lines 119-120: “Furthermore, obviously disabled children were excluded as well as children with Bayley scores below 70”. Figure 1 reports the number of children excluded by each criterion.

c) How many children were eligible?

Response: In early childhood, 1,533 children were eligible and enrolled in the study but only 1,330 were administered the Bayley-III and 1,311 Bayley-III scores were analyzed, as indicated in lines 144-148, and in Figure 1.

In middle childhood, all 1,330 children who had been tested on the Bayley-III in early childhood were eligible for study inclusion. However, the WISC-V and school achievement tests were only administered to 940 of these children, as reported in the Results section (line 307). Most of the sample loss between early and middle childhood was due to migration as reported in lines 309-310 and in Figure 1. 

We have therefore not modified the text on these lines but hope that the edits to the definition of eligibility (see Reviewer 3, Response 4.b. above) have contributed to clarity on this point.

d) What was the response rate among eligible children? Was this different by strata?

Response: Yes, response rates to study participation were higher for lower socio-economic sectors. The extreme case was sector 4, where it was very difficult to even identify eligible children by means of the door-to-door census. On many occasions, we were banned access to the residential blocks where sector 4 families live. Therefore, at one point in the enrollment stage, the research team decided to drop sector 4 from the study and distribute households in sector 4 to sector 1 and, in a higher proportion, to sectors 2 and 3. Nonetheless, the 12 children living in sector 4 who had already been enrolled, were kept for analysis. This is explained in detail in prior work (Rubio-Codina et al., 2015). 

e) Clarifying the above will also help explain the statement in lines 143-145 regarding what seems to be one strata (sector but not age is clear) that was dropped due to low response rate.

Response: We hope that the edits throughout the first paragraph of the Study design and participants subsection have clarified this sentence. We did not get into more details due to space limitations and given it has been reported in prior work (Rubio-Codina et al., 2015). 

f) The above might be incorporated into Figure 1, preceding the n=1,533. Also, are the blocks in Figure 1 synonymous with strata?

Response: No, blocks are a subdivision of strata. This has now been clarified in the text (lines 134-135: “Within each sector stratum, blocks were selected by random probability, weighting by the proportion of women in fertile age”) and in Figure 1. 

g) The authors appear to use the bootstrap methods to adjust for strata (e.g., note to Table 2). Making that clear when the use of strata in the design is presented would help the reader. Also, was there any oversampling within/across strata, suggesting sampling weights would also be needed (Heering, West & Berglund, Applied Survey Data Analysis, Chapman & Hall)

Response: We used bootstrapped P values in all statistical inference. These have been computed stratifying by the design strata: socio-economic sector and age. This has been clarified in the Statistical analysis section (lines 290-293: “Bootstrapped P values [41], computed stratifying by the design strata (socio-economic sector and age), were used in all inference and to compare whether the predictive validity of a test differed significantly across the three age groups examined.”) and in the footnotes to Tables 2 (line 333), 3 (line 360) and 4 (line 337). In the correlation analyses (Tables 2 and 3), computing P values in this way is a refinement over the use of standard P values. In line with this, results are the same if regular P values are used. However, in the comparison of the predictive validity of each pair of tests (i.e. the analysis presented in Table 4), the use of bootstrapped P values is important because we are comparing correlations that come from different estimations and sometimes even different samples. The bootstrap method accounts for this (Efron, 1982). 

As indicated in the response to Comment 4.d. of Reviewer 3, in early childhood, higher study participation rates were found for lower socio-economic sectors; and the total sector sizes were modified with respect to what was originally planned. In previous work we had investigated this issue in detail and had showed that weighted and unweighted analysis offered unaltered results (Rubio-Codina et al., 2015). We are not delving into this issue in this paper for simplicity as it is already complicated and due to space limitations. Furthermore, we are not examining the effect of socio-economic variation nor other possible modifiers; and as shown in Table 1, the sample was well-balanced across age groups. 

g) As I understand it, the PPVT and TVIP identify the child’s approximate word knowledge and then show words around that target level. It is thus not clear why a subset of words were selected from the TVIP? Is this allowable in the standard instructions for the TVIP, or was permission received from the TVIP developers to make that change and call the scores TVIP scores?

Response: The reviewer’s understanding of how the PPVT and TVIP work is correct. Using existing nationally representative data for Colombia from the ELCA study , we selected the 75 (out of the 125) words in the TVIP that would be expected to be in the children’s age range and excluded ones unlikely to have any variation (i.e. either all passed or failed). After piloting, we re-ordered them by difficulty for this population in order to reduce testing time and make it easier to administer. Instead of following the standard method of administering the test by using the start and stop rules (which are relatively difficult to train and use during administration), we had all children start at word item 1 and we kept them on the test until three consecutive errors were made. This also simplified the record forms to be used. The probability of a child 6-8 years of being administered a word in any of the 50 remaining words in the test was very close to zero. We have clarified this in the text in lines 214-218, which now read: “These words were selected using existing data from urban children of the same age and socio-economic status from a longitudinal nationally representative survey (ELCA) [37]. We first selected the words in the relevant age range that showed sufficient variability and ordered them by difficulty; we then made final decisions after piloting. The aim was to simplify test administration and minimize testing time.”

We did request permission to Pearson Inc, the test publisher, to do this, and paid the corresponding license. Throughout the text, we are referring to these scores as “receptive vocabulary, based on TVIP” or “receptive vocabulary, “TVIP” (i.e. “TVIP” in quotes)”; purposively, we are not explicitly calling them TVIP scores. 

h) Why was principal component rather than factor analysis used? Does the MC-HOME have a standard scoring? If so, why wasn’t that standard scoring used?

Response: The MC-HOME does have a standard scoring procedure consisting of adding up all items (binary indicators). In fact, this is how we had constructed the MC-HOME total score. The FCI total score for play materials and play activities was also constructed by adding up binary indicators for the 8 varieties of play materials and the 7 varieties of play activities (with cutoffs varying depending on the empirical prevalence of each indicator). In addition, both scores were internally standardized using the procedure described in the Statistical analysis section. This way of constructing FCI and MC-HOME scores was chosen as it is consistent with the way in which the FSIQ and school achievement scores had been constructed, which facilitated comparisons. 

In Table 1, we are referring to the FCI and MC-HOME scores as “Total scores, internally standardized”, which is consistent with the methodology explained in the paragraph above. This methodology, however, does not correspond with the text in lines 228-231 in the old version of the Manuscript, which was incorrect, and has now been modified, see lines 245-247, which now read: “FCI and MC-HOME total scores were constructed by adding up binary indicators, with cutoffs varying depending on the empirical prevalence for each indicator for the FCI.” 

The mistake was due to the fact that we had used FCI and MC-HOME scores constructed by polychoric principal component analysis in earlier versions of the paper (for consistency with earlier work: Rubio-Codina, Attanasio, & Grantham-McGregor, 2016; Rubio-Codina & Grantham-McGregor, 2019), but had decided to use “total internally standardized scores” in the end for simplicity and for comparability with the way in which the FSIQ and school achievement scores were computed. However, we had forgotten to update the text and are very grateful to the Reviewer for helping us identify this omission. 

i) Were results sensitive to whether raw scores or full sample standardized scores were used, rather than age-standardized scores? What is the conceptual rationale for using age-standardized scores? (This question is returned to below).

Response: Raw scores increase with age and this correlation may well vary by test (see for example correlations in S1 Table). Any comparison of concurrent and predictive validity must partial out the effect of age (so as to avoid confounding the association between the tests with that related to age of the child at the time of the assessment, as pointed out by Reviewer 3 in Comment 7). This can either be done by computing partial correlations (which partial out the effect of age) or by removing the effect of age beforehand. If all scores had equivalent externally standardized scores (i.e. scores computed using norms for Colombia or norms for the US), we could use these scores in the analysis. Given that this is not the case (please see our response to Reviewer 2, Comment 2), we opt for standardizing scores internally. The main advantage of the method used is that it handles age effects consistently across all short tests, the Bayley-III and the FCI, which facilitates the comparison of correlations between pairs of tests. It is also less sensitive to outliers/small sample sizes than methods traditionally used to internally standardize scores (i.e. using interval-specific (for example, monthly) means and SDs to compute z-scores). We have revised the Statistical analysis, lines 262-271, which now read: “Therefore, we internally standardized the raw scores over age using age-conditional means and SDs, computed non-parametrically, after removing testers’/interviewers’ effects, as done in previous work [10]. This is, for each value of the residual of the raw scores on tester or interviewer dummies, we constructed a Z-score by subtracting the age-conditional mean and dividing by the age-conditional SD, both computed using local polynomial regressions. Unlike using norms from the reference populations (i.e. externally standardized scores) for each test, this standardization method handles age consistently across tests, which facilitates comparisons. It is also less sensitive to outliers/small sample sizes than methods traditionally used to internally standardize scores, which usually use interval-specific (for example, monthly) means and SDs to compute z-scores.” 

We believe the age spread in the sample, of 36 months, is too large to compute full sample standardized scores. Moreover, using full sample standardized scores would not permit the analysis of the data by age, which we think is important precisely given the large age spread and the fast pace of development from ages 6 to 42 months (as well as the differential predictive ability found by age). This has been addressed in the Introduction in lines 99-103: “As with concurrent validity [10], and given the broad age range of the children in the study sample (36 months) and fast pace of development from 6 to 42 months, the analysis of predictive validity was performed by 12-months-of-age intervals (6-18, 19-30 and 31-42 months-of-age). These intervals were the smallest we thought possible, given the available sample sizes.”; and line 113: “…tests and measures typically used as ECD proxies), given at three different age ranges, to…”.

Results are robust to the standardization method used and this is now explicitly stated in lines 423-427: “Analyses were repeated using FSIQ externally standardized scores, scores internally standardized using traditional methods, ASQ-3 scores using the original 6-item questionnaires, dropping outliers <-2 SD in the FSIQ distribution, and weighting by the inverse probability of being assessed in middle childhood to correct for loss at follow-up. Results were little altered in all cases.”

j) The authors refer to removing testers/interviewers effects “as done previously” and provide a citation. For the paper to “stand on its own” this adjustment should be briefly described so that it is transparent to the readers of this paper. Again, sensitivity to this adjustment would also be of value.

Response: We have added an explanation to this sentence in the Statistical analysis section, lines 262-271. The relevant paragraph is copied in our response to the previous comment. A reference to the robustness of findings to alternative standardization methods has been added at the end of the Results section in lines 423-427, which are also copied in our response to the previous comment. 

k) The point on lines 248-250 that the age-standardization process has the advantage of handling age consistently across tests makes sense, but it is not clear how the approach is “less sensitive to small samples.” By the latter do you mean that use calculated the means and SDs non-parametrically? Even if this is the meaning of the statement, it is not clear how norms from reference populations would be more sensitive to small samples, since often norming samples are large and likely as large or larger than your within-age samples. Clarifying the meaning of your statement would help.

Response: The Reviewer’s argument is correct and the point is well-noted. This has been clarified in lines 269-271; “It is also less sensitive to outliers/small sample sizes than methods traditionally used to internally standardize scores, which usually use interval-specific (for example, monthly) means and SDs to compute z-scores.” Thank you. 

l) Analysis within 12-month-of-age ranges is sensible, but as a reader I wanted this plan to be more explicit earlier in the paper, with the age ranges (how many, min/max age) explicitly defined and its rationale and connection to the design offered

Response: Thanks. We have stated that the analysis is by 12 months-of-age ranges and added a justification for the choice of interval ranges in the Introduction in lines 99-103: “As with concurrent validity [10], and given the broad age range of the children in the study sample (36 months) and fast pace of development from 6 to 42 months, the analysis of predictive validity was performed by 12-months-of-age intervals (6-18, 19-30 and 31-42 months-of-age). These intervals were the smallest we thought possible, given the available sample sizes.”; and in the study aims in line 113: “tests and measures typically used as ECD proxies), given at three different age ranges, to”. Age groups were defined given the broad age range of the children in the study sample and the fast pace of development from 6 to 42 months. The intervals chosen were the smallest we thought possible. The proportion/number of children in each age group by battery are provided in Table 1. The minimum and maximum age are not reported as they correspond to the intervals’ lower and upper limits. 

m) The statement on p. 258-260 seems to be combining two ideas, and relates to the request of clarifying why you chose to age-standardize. You seem to be implying that computing correlations within 12-month-age-bands is equivalent to controlling for age. However, the analysis approach assumes correlations are the same within 12-month-age bands and allows them to differ across 12-month-age-bands. It seems there would be a conceptual motivation for looking at analyses within 12-month-age-bands, as discussed below.

Response: The analyses is on internally standardized scores within 12 months-of-age bands. This means that (i) the age effect included in raw scores has been netted out (please see response to Reviewer 3, Comment 4.i.); and (ii) correlations are forced to be the same within 12 month-age bands but differ across 12-month-age-bands. Therefore, the two sentences refer to two different concepts, which we have tried to clarify in lines 278-282: “The ability of all tests, the FCI, height-for-age, and stunting to predict FSIQ and school achievement was analyzed by 12-months-of-age intervals, computing Pearson correlations within each age range. In all correlations, the internally standardized scores were used, which is equivalent to computing partial correlations controlling for testers/interviewers and age flexibly.”

n) In lines 270-272 the authors note that sample sizes varied across tests. A table that summarizes the tests, and any sample loss that is built in (due to eligibility for the test) and that is not (due to refusals or data errors) would be helpful. Such a table could also contain the names, acronyms, and study-defined types of the tests (incorporating aspects of the glossary at the start of the paper). If the latter type of test-level missing was extensive in some cases, did the authors examine how those with and without scores differ and consider addressing it with multiple imputation or FIML?

Response: The flow diagram in Figure 1 details the information requested by the Reviewer. This is, it includes the eligibility criteria, any occurrence of sample loss (detailing the reasons), and the subsample of analysis for each test. 

We would like to clarify that all children in the analysis sample were tested on the Bayley-III; and their height, weight, and home FCI were collected. Approximately half of these children were randomly assigned to Battery A (ASQ-3, Denver-II, and SFI or SFII) and the remaining half were randomly assigned to Battery B (BDI-2, WHO-Motor), as explained in the Methods section and as illustrated in Figure 1. We have clarified this in lines 145-147: “approximately, half the children were assessed on the short tests in battery A and the remaining half on those in battery B.”

As explained, not all tests were available for all children 6-42 months. More specifically, the SFI is available for children 8-18 months; the SFII for children 19-30 months; and the WHO-Motor was only analyzed for children 6-15 months, since it showed limited variability for older children. Hence, not all tests were administered to all children, but only to children in the adequate age range, which does generate differences in subsample size by test. Test/analysis subsample sizes are also reported in Table 3 and in Figure 1. 

Following the Reviewer’s suggestion, we have added the tests’ names and acronyms to the legend of Figure 1 (which is included in the main text and not with the Figure, as required by the journal’s formatting guidelines), in lines 150-160: “ASQ-3 (Ages and Stages Questionnaires, third edition); Denver-II (Denver Developmental Screening Test, second edition); SFI (MacArthur-Bates Communicative Development Inventories, Short-Form I in Spanish); SFII (MacArthur-Bates Communicative Development Inventories, Short-Form II in Spanish); BDI-2 (Battelle Developmental Inventory screener, second edition); WHO-Motor (World Health Organization Gross Motor Milestones); Bayley-III (Bayley Scales of Infant and Toddler Development, third edition); FCI (Family Care Indicators); WISC-V (Wechsler Intelligence Scale for Children, fifth edition); WM-III (Woodcock-Muñoz Test of Achievement, third edition); “TVIP” (Peabody Picture Vocabulary Test-Revised, Spanish version, adapted/selected subset of words). a Children 8-18 months. b Children 19-30 months. c Children 6-15 months.”

We have also edited lines 270-272 (current lines 299-302) to clarify what was meant, as it could lead to confusion, as pointed out by the reviewer. Thank you. These lines now read: “To further explore the effect of age under 19 months in the Bayley-III, the FCI, height-for-age, and stunting, which were available for children in both batteries, we repeated the analyses by 6-months-of-age intervals on enrollment. For the other tests, given to children in either battery A or battery B, sample sizes were considered too small to subdivide. ”

Finally, we did not attempt to impute test scores using multiple imputation or FIML or any other method since these are our main outcomes. We did however re-run the analysis correcting for sample loss (by weighting all calculations by the inverse probability of being given the test) and found consistent results, as noted in lines 423-427: “Analyses were repeated using FSIQ externally standardized scores, scores internally standardized using traditional methods, ASQ-3 scores using the original 6-item questionnaires, dropping outliers <-2 SD in the FSIQ distribution, and weighting by the inverse probability of being assessed in middle childhood to correct for loss at follow-up. Results were little altered in all cases.” In prior work (Rubio-Codina et al. 2015), we had also assessed the robustness of the concurrent validity results by weighting the data by the probability of sample inclusion, which varied by socio-economic sector, as noted in Comment 4.d. of Reviewer 3. 

o) What does “Migrants were tracked whenever feasible” mean? (line 281).

Response: It means that all households that had migrated were followed to the extent that it was possible to follow them. This is to say, (i) we could find out about the household’s new residence; and (ii) the new residence was in Bogota, or within less than 1-hour distance by bus from the city. Unfortunately, we did not have funds to track households any further than this. This has been clarified in lines 311-312: “Migrants who could be located and lived within 1-hour from Bogota (by bus) were tracked.” 

p) Table 2 lists “n = 2000 replications” which appears to correspond to bootstrapping. What was the number of children on which each value was based?

Response: Correlations in Table 2 were calculated on the entire sample (N = 937 children). We have added this to the Table footnote (line 331) and apologize for the omission!

q) Did the authors test whether the correlations differed significantly across the age groups in Table 3? Adding confidence intervals would both show the uncertainty of estimation and allow readers to see which estimates overlapped across age groups.

Response: We have added a Supplementary Table (S3 Table) with the results of the tests of the comparison of the predictive ability across age groups. For consistency with the rest of the analysis, the statistical significance of these differences has been investigated with bootstrapped P values. We have updated the Statistical analysis in lines 290-293: “Bootstrapped P values [41], computed stratifying by the design strata (socio-economic sector and age), were used in all inference and to compare whether the predictive validity of a test differed significantly across the three age groups examined.” and Results in lines 385-388: “The increase of the correlations across age groups was significant between the youngest age group (children 6-18 months) and the oldest (children 30-41 months) for the Bayley-III and the Denver-II cognitive and language scales (S4 Table). For the Bayley-III scales the increase was also significant between the youngest and the middle age group (children 19-24 months)”; lines 398-399: “The increase in predictive validity between the SFI (children 6-18 months) and the SFII (children 19-24 months) was statistically significant (S4 Table)”; line 405 “This increase was statistically significant (S4 Table).”; and lines 409-410: “The predictive validity of height-for-age and stunting increased with age, although the increase was not statistically significant (S4 Table).”

5) The authors might attend to the following in the discussion:

a) The authors provide recommendations to readers about which tests they might use, based on the results. Doing so is helpful, but as a reader I felt at times the recommendations focused on relative rather than absolute predictive validity (e.g., p. 405-406). The authors might remind the reader that the predictive validity is low-to-moderate.

Response: Thanks very much. This is an important point. The first two paragraphs of the Discussion do explain that the predictive validity values found are in the low-to-moderate range. As suggested by the Reviewer, we have added a line in the Discussion (line 461: “…predictive ability to the Bayley-III, although it was low-to-moderate, as indicated earlier.”) and the Conclusions (line 524 “…after this age, predictive validity values were only low-to-moderate at best.”) to re-emphasize this point. 

b) In lines 417-422, the authors might be careful to acknowledge that the FCI may proxy for other aspects of household or community resources that would be the better proximal cause for intervention, given this study is simply focusing on correlations.

Response: We agree that the FCI is related to household resources and it is known to be related to maternal education and household wealth; but neither of these are targeted by ECD interventions so the improvement found in child development reflects an improvement in maternal behavior towards the child and in the number and quality of the interactions with the child. The FCI is a direct measure of these. We have clarified in the paragraph that we were referring to ECD interventions—see lines 484-487: “Whilst individual item performance can vary depending on the context and children’s age, home environment quality has been identified as a relevant protective factor [25,26] and the FCI has often improved with ECD interventions [51–53].”

6) Figure 2 does not well “stand on its own”

Response: Thanks very much for all these very useful suggestions. We have revised Fig 2, which we believe is now clearer, as a result.

a) The acronyms should be defined in the note (or refer the reader to where to find the definitions).

Response: The acronyms are now defined in Fig 1 footnote (lines 150-160). We have added a note in Fig 2 footnote (line 351), referring the reader to Fig 1. 

b) The Y axes are not labelled.

Response: We have labeled the Y axes. 

c) The n’s on which the values in the Figure are based should also be clear either in each the legends/titles/axes or the table notes (or tell the reader where to find the n’s in another table that explicitly adds and lists them).

Response: The n’s are in Table 3. We have added a note in Fig 2 footnote (line 351), referring the reader to Table 3. 

d) The Y axis or table note should also make clear what is being correlated with what: that each correlation is between the kids in the younger age group with their own scores 5 years later. For that reason, it would be preferable on the X axis to list the follow-up along with the initial age-range, as those together define the meaning of the correlation (e.g., B: 6-18 m, F: ~ 6y; B: 19-30 m, F ~ 7y; B: 31-42 m, F ~ 8y).

Response: We have relabeled the X axes as suggested and have also added a note to the Fig 2 footnote, explaining what each correlation depicted means and the relationship between the age groups. The relevant lines are 345-350: Each plot in Fig 2 shows, for each developmental domain, the average correlation between scores in early childhood with FSIQ scores in middle childhood, by age group. The youngest age group are children 6–18 months in early childhood and 6 years in middle childhood; the middle age group are children 19–30 months in early childhood and 7 years in middle childhood; and the oldest age group are children 31–42 months in early childhood and 8 years in middle childhood.”

7) This point just noted about labelling the X axis relates to a broader conceptual question that should receive more attention throughout the paper. The study design starts with children in three age bands, centered on age 1, age 2, and age 3. These children are followed up 5 years later at ages 6, 7, and 8, respectively. Thus the size of the correlations depend on both how well the measures used in early childhood capture each latent construct at those early ages, and how well the measures used in middle childhood capture each latent construct at those later ages. In other words, the lower correlations for the “6-18 months” group may to some degree reflect how well the middle childhood tests measure outcomes for 6 year olds versus 8 year olds. If the study could have followed children to age 8, the meaning would have been further strengthened. Presumably that wasn’t done in 2018, but the meaning of the values presented should be clear (as in the labels of Figure 2) and the rationale for the design and its interpretation should be clearer throughout the paper. More attention to the reliability and validity of the middle childhood measures (including the WISC, but also school achievement - are 6, 7, and 8 year olds all in schools of similar type in this context?) for these ages, and especially in lower/middle income countries and for Spanish speaking children, would be of value. 

Response: Thanks for raising this important point. It is true that the analysis of predictive validity of the early childhood tests is confounded by the child’s age in middle childhood, as well as by the quality of the middle childhood assessments (to the extent to which such quality varies with age). Unfortunately, for budgetary and logistical reasons, we were not able to test all children at ages 6, 7 and 8 years in middle childhood. We therefore focused on the time between the 2 test sessions being equal for all age groups. 

However, we have investigated whether the quality of the assessments in middle childhood varied with age—please see the following two Tables (not presented in the Manuscript for simplicity and owing to space restrictions). The first one shows that the internal consistency (Cronbach’s alpha) of all subtests that compose the FSIQ and the school achievement score remains stable between ages 6, 7 and 8 years; thus, indicating stable quality. However, as shown in the second Table, we do observe slightly higher correlations with socio-economic variables and amongst tests in middle childhood for children 7 and 8 years than for children age 6 years, which would suggest that the middle childhood assessments worked better for slightly older children (which seems sensible and in line with the usual performance of tests). We have updated the text in the Results section (lines 326-328: “FSIQ and school achievement were associated with household characteristics and with each other as theoretically expected (Table 2); these correlations were higher for children 7 and 8 years than for the 6-year olds (not shown).”) to reflect this and have also added this limitation in the Discussion (lines 512-517: “A further limitation is that the predictive validity of the early tests might be partly confounded with the performance of the middle childhood tests since, in middle childhood, all children were tested 5.5 years after the early assessment. Performance of the middle childhood tests seems to increase with age according to some reliability and validity indicators (correlations of tests with each other and with socio-economic variables) but not to some others (internal consistency).”).

Internal consistency (Cronbach’s alpha) of middle childhood subtests by age

Correlations of middle childhood tests with concurrent Socio-economic variables and with each other by age

Regarding the comment of whether all schools are of similar type in the context, we note that 40% of the children in the study sample are in private primary schools and 60% are in public primary schools. These proportions are balanced by age of the child in middle childhood and by school grade. We have not added this information in the Manuscript due to space restrictions.

References

Araujo, C., Dormal, M., Grantham-McGregor, S. M., Lazarte, F., Rubio-Codina, M., & Schady, N. (2019). 

Home visiting at scale and child development, under review. 

Attanasio, O. P., Fernandez, C., Fitzsimons, E. O. A., Grantham-McGregor, S. M., Meghir, C., & Rubio-Codina, M. (2014). Using the infrastructure of a conditional cash transfer program to deliver a scalable integrated early child development program in Colombia: cluster randomized controlled trial. BMJ, 349(sep29 5), g5785–g5785. https://doi.org/10.1136/bmj.g5785

Bayley, N. (2006). Bayley Scales of Infant and Toddler Development–Third Edition: Technical manual. San Antonio, TX: Harcourt Assessment.

Black, M. M., Walker, S. P., Fernald, L. C., Andersen, C. T., DiGirolamo, A. M., Lu, C., … Committee, L. E. C. D. S. S. (2017). Early childhood development coming of age: science through the life course. The Lancet, 389(10064), 77–90. https://doi.org/10.1016/S0140-6736(16)31389-7

Efron, B. (1982). The Jackknife, the Bootstrap, and Other Resampling Plans. CBMS-NSF Regional Conference Series in Applied Mathematics (Vol. 38). Philadelphia, PA: SIAM. https://doi.org/http://dx.doi.org/10.1137/1.9781611970319

Fernald, L. C. H., Prado, E., Kariger, P., & Raikes, A. (2017). A Toolkit for Measuring Early Childhood Development in Low and Middle-Income Countries (Working Paper No. 122031). Washington, DC. Retrieved from http://documents.worldbank.org/curated/en/384681513101293811/A-toolkit-for-measuring-early-childhood-development-in-low-and-middle-income-countries

Frankenburg, W. K., Dodds, J., Archer, P., Bresnick, B., Maschka, P., Edelmann, N., & Shapiro, H. (1990). The DENVER II Technical Manual. Denver, CO: Denver Developmental Materials.

Frongillo, E. A. ., Tofail, F., Hamadani, J. D., Warren, A. M., & Mehrin, S. F. (2014). Measures and indicators for assessing impact of interventions integrating nutrition, health, and early childhood development. Annals of the New York Academy of Sciences, 1308(1), 68–88. https://doi.org/10.1111/nyas.12319

Gladstone, M., Lancaster, G. A., Umar, E., Nyirenda, M., Kayira, E., Van, N. R., … Smyth, R. L. (2010). The Malawi Developmental Assessment Tool (MDAT): The Creation, Validation, and Reliability of a Tool to Assess Child Development in Rural African Settings, 7(5). https://doi.org/10.1371/journal.pmed.1000273

Grantham-McGregor, S., Cheung, Y. B., Cueto, S., Glewwe, P., Richter, L., & Strupp, B. (2007). Developmental potential in the first 5 years for children in developing countries. The Lancet, 369(9555), 60–70. https://doi.org/10.1016/S0140-6736(07)60032-4

Hamadani, J. D., Mehrin, S. F., Tofail, F., Hasan, M. I., Huda, S. N., Baker-Henningham, H., … Grantham-McGregor, S. (2019). Integrating an early childhood development programme into Bangladeshi primary health-care services: an open-label, cluster-randomised controlled trial. The Lancet Global Health, 7(3), e366–e375. https://doi.org/10.1016/S2214-109X(18)30535-7

Newborg, J. (2005). Battelle Developmental Inventory-2nd Edition. Rolling Meadows, IL: Riverside Publishing.

Rubio-Codina, M., Attanasio, O., & Grantham-McGregor, S. (2016). Mediating pathways in the socio-economic gradient of child development: Evidence from children 6-42 months in Bogota. International Journal of Behavioral Development, 40(6), 483–491. https://doi.org/10.1177/0165025415626515

Rubio-Codina, Marta, Araujo, M. C., Attanasio, O., Muñoz, P., & Grantham-McGregor, S. (2016). Concurrent Validity and Feasibility of Short Tests Currently Used to Measure Early Childhood Development in Large Scale Studies. Plos One, 11(8), e0160962. https://doi.org/10.1371/journal.pone.0160962

Rubio-Codina, Marta, Attanasio, O., Meghir, C., Varela, N., & Grantham-McGregor, S. (2015). The Socioeconomic Gradient of Child Development: Cross-Sectional Evidence from Children 6–42 Months in Bogota. Journal of Human Resources, 50(2), 464–483. https://doi.org/10.3368/jhr.50.2.464

Rubio-Codina, Marta, & Grantham-McGregor, S. (2019). Evolution of the wealth gap in child development and mediating pathways: Evidence from a longitudinal study in Bogota, Colombia. Developmental Science, (e12810.), 1–15. https://doi.org/10.1111/desc.12810

Tofail, F., Hamadani, J. D., Mehrin, F., Ridout, D. A., Huda, S. N., & Grantham-McGregor, S. M. (2013). Psychosocial Stimulation Benefits Development in Nonanemic Children but Not in Anemic, Iron-deficient Children. The Journal of Nutrition, 143(6), 885–893. https://doi.org/10.3945/jn.112.160473.Three

Walker, S. P., Chang, S. M., Powell, C. A., & Grantham-McGregor, S. M. (2004). Psychosocial intervention improves the development of term low-birth-weight infants. Journal of Nutrition, 134, 1417–1423.

---

## [Decision Letter · Decision Letter 1]

4 Nov 2019

PONE-D-19-20029R1

Predictive validity in middle childhood of short tests of early childhood development used in large scale studies compared to the Bayley-III, the Family Care Indicators, height-for-age and stunting: A longitudinal study in Bogota, Colombia

PLOS ONE

Dear Dr Rubio-Codina,

Thank you for submitting your manuscript to PLOS ONE. After careful consideration, we feel that it has merit but does not fully meet PLOS ONE’s publication criteria as it currently stands. Therefore, we invite you to submit a revised version of the manuscript that addresses the points raised during the review process.

We would appreciate receiving your revised manuscript by Dec 19 2019 11:59PM. To enhance the reproducibility of your results, we recommend that if applicable you deposit your laboratory protocols in protocols.io, where a protocol can be assigned its own identifier (DOI) such that it can be cited independently in the future. For instructions see: http://journals.plos.org/plosone/s/submission-guidelines#loc-laboratory-protocols

We look forward to receiving your revised manuscript.

Kind regards,

Thach Duc Tran, M.Sc., Ph.D.

Academic Editor

PLOS ONE

Reviewers' comments:

Reviewer's Responses to Questions

**Comments to the Author**

1. If the authors have adequately addressed your comments raised in a previous round of review and you feel that this manuscript is now acceptable for publication, you may indicate that here to bypass the “Comments to the Author” section, enter your conflict of interest statement in the “Confidential to Editor” section, and submit your "Accept" recommendation.

Reviewer #1: (No Response)

Reviewer #2: All comments have been addressed

2. Is the manuscript technically sound, and do the data support the conclusions?

Reviewer #1: Yes

Reviewer #2: Yes

3. Has the statistical analysis been performed appropriately and rigorously? 

Reviewer #1: Yes

Reviewer #2: Yes

4. Have the authors made all data underlying the findings in their manuscript fully available?

Reviewer #1: Yes

Reviewer #2: Yes

5. Is the manuscript presented in an intelligible fashion and written in standard English?

Reviewer #1: Yes

Reviewer #2: Yes

6. Review Comments to the Author

Reviewer #1: PONE-D-19-20029R1

Predictive validity in middle childhood of short tests of early childhood development used in large scale studies compared to the Bayley-III, the Family Care Indicators, height-for-age, and stunting: A longitudinal study in Bogota, Colombia

The study presents analyses of mental development data from a cohort 6 to 42 months from Bogata showing data on predictive validity. The purpose is to provide data that would allow researchers to select short measures of early child development, with evidence that they are sound alternatives to the Bayley III in predicting WISC and academic scores in middle childhood. The methodology and statistical analyses were rigorously conducted. One limitation is that the findings are limited to descriptive/ longitudinal purposes; the authors cannot convincingly address program evaluation, though they would like to. We do not know how sensitive is the Denver to evaluate a nutrition-WASH-stimulation program. A second limitation is that researchers will shortly have a short global measure and a longer program evaluation measure to use instead of the Denver Screening Test or other existing alternatives. Already they have the MDAT which was not included in this study and not mentioned in the ms.

Abstract and elsewhere

Evaluation of interventions and predictive validity of such tests entail two separate issues.

Predictive validity is the main phrase in the title; however the first sentence of the Abstract concerns the evaluation of interventions at scale. Evaluation is not really “hindered by the lack of developmental tests” because large-scale evaluations can validly assess representative samples to compare with controls. Your reply to this comment was arguably limited to purchases from the US, and did not consider the important value for LMIC in training local researchers and the value in translation and adaptation. Parent-report measures similar to those used in this study are subject to bias if the intervention concerns psychosocial stimulation: mothers who receive messages about child development now have beliefs about when developmental milestones occur and mothers who learn about interacting with their child will have more accurate observations of when milestones are attained (see WB toolkit p52). So use of the ASQ instead of a direct child assessment provides weak support. Frankly, the data here address predictive validity and not evaluations of interventions; reference to evaluation of interventions in the Introduction and Discussion should be dropped as it misguidedly encourages the use of screeners and parent reports in interventions. If you think that the Denver or another measure has been effectively used to evaluate a large unconditional cash transfer program, this can be mentioned in the Discussion as additional information about the test being advocated.

Introduction and Enrollment

Lines 51-53. This sentence should be revised: “Whilst existing diagnostic tests such as the Bayley Scales [5,6] are sensitive to ECD interventions [2,7–9], they are very expensive, take a long time to administer and require highly trained testers with a certain level of technical expertise [2].” Only in some countries are they expensive, take a long time to deliver and require highly trained testers … It might be more accurate to state, “According to our experience, …they are very expensive…” I would have to say that only in full do they take a long time, and rarely have I seen highly trained testers doing the training. I wonder how many other researchers had to spend 6 weeks training psychology graduates on the Bayley III. You might want to consider that this is your experience; colleagues in Peru had the same experience so I don’t doubt your word. But it is not a feature of the test and not necessarily the experience of others [I’m not even sure that the authors of the WB 2017 toolkit have the experience of training others on the Bayley].

Methods

Lines 166-181. Thank you for adding more about whether tests were direct or parent report and how many items were parent report if the test included both. Appreciate also the information about including higher items on the screeners to avoid ceiling. These are critical features of child assessment.

Line 282. Thank you for elaborating on the Denver motor-adaptive scale. The original Bayley [ref 5] Performance subscale combined cognitive and fine motor, so I can see how the Denver does so.

Nothing was said about adaptation of the measures, particularly the WISC, Bayley, BSD, Denver other than translation. Could you state that no adaptations were required?

Results

Thank you for clarifying improvement in retest scores.

Thank you for providing age correlations in S2.

Table S2. p cannot be <.0000. Better to write p<.0001.

Table 1. Internally standardized scores are not very meaningful. I can understand why you used them for analyses but they are not meaningful for readers who are usually given standardized Bayleys and raw HOME. Some of the raw scores were useful in S2 but it would be good to see raw HOME in order to compare with other studies (rather than internally standardized) and externally standardized scores for Bayley III and WISC. I noticed that the Bayley III standardized scores in the 2016 paper looked similar to US and UK means; they won’t always be this high in other regions of the world. Did you expect standardized IQ scores to drop in middle childhood?

For the other measures (ASQ, BSI, Denver) can you provide some central score or % above/below a cut-off? It is problematic that these tests do not allow for scores to be shown.

Discussion

Line 473. “recall bias” is probably not the right phrase to describe the bias of mothers who have attended a stimulation class. It may be more like an “observation bias” due to spending more time with their child, and a “desirability bias” due to knowing what milestones of development to expect.

Line 487 etc. My previous comment re the FCI was “prevalence for several items asymptotes at a high rate at 4 months of age, and other items like counting and drawing asymptote at a low rate at 4 years. So not all items are relevant for low-income countries at all these ages.” The ages being referred to are between 4m and 4 year, not simply those two ages. If high rates asymptote at 4m it means they have hit ceiling for all older children. If rates asymptote at a low level by 4 years, it means they are at floor for many months before this. Have you assessed variation in your FCI scores for your full sample by age? Thank you for pointing out that the FCI will have different means in Africa and Asia.

Line 488. One problem with using the FCI to evaluate a psychosocial program is that it assesses stimulation but not responsive caregiving.

The Denver Screening test appears to be the strongest substitute for the Bayley. It may have the same administration duration, requires caregiver report for 39% of the items, and costs less in the US. The authors did not provide information on length of training testers and adaptation required. This would help prospective researchers know what is required.

Limitations. Reasons for not using the MDAT should be mentioned in the limitations. I understand that you were not aware of its launch in 2010 but readers will wonder why it was ignored given its currently popularity.

Another limitation is that height and stunting may be less predictive in Colombia than in Africa because of the narrow range of values, particularly in the stunted range.

Reviewer #2: The authors have improved their manuscript and they sufficiently addressed my comments.

For their understanding: The Bayley III is also available in Dutch with Dutch norms, and we published the differences between the US norms and the Dutch norms in PLOS One.

7. PLOS authors have the option to publish the peer review history of their article (what does this mean?). If published, this will include your full peer review and any attached files.

Reviewer #1: No

Reviewer #2: Yes: Anneloes van Baar

---

## [Author Response · Author response to Decision Letter 1]

19 Dec 2019

PONE-D-19-20029

Predictive validity in middle childhood of short tests of early childhood development used in large scale studies compared to the Bayley-III, the Family Care Indicators, height-for-age, and stunting: A longitudinal study in Bogota, Colombia

Responses from the authors to the Reviewer’s comments 

We thank the Reviewers for reading through our first set of responses in detail and for their subsequent comments to continue improving the Manuscript. We respond below to the new set of reviewers’ comments. For clarity we retain the comments and add our response in italics. We have numbered the reviewers’ comments to facilitate cross-references to our responses. Please note that the references to the line numbers where edits have been made correspond to the version of the Manuscript without tracked changes (clean version). 

Reviewer Comments to the Author

Reviewer #1: PONE-D-19-20029R1

Predictive validity in middle childhood of short tests of early childhood development used in large scale studies compared to the Bayley-III, the Family Care Indicators, height-for-age, and stunting: A longitudinal study in Bogota, Colombia

The study presents analyses of mental development data from a cohort 6 to 42 months from Bogata showing data on predictive validity. The purpose is to provide data that would allow researchers to select short measures of early child development, with evidence that they are sound alternatives to the Bayley III in predicting WISC and academic scores in middle childhood. The methodology and statistical analyses were rigorously conducted. 

1. One limitation is that the findings are limited to descriptive/ longitudinal purposes; the authors cannot convincingly address program evaluation, though they would like to. We do not know how sensitive is the Denver to evaluate a nutrition-WASH-stimulation program. 

Response: This is an important limitation, but also one that is beyond the scope of this work. We have acknowledged this in lines 488-491, which have been rewritten to refer to impact evaluations of cash transfer programs using some of the short tests—the Denver-II in Nicaragua and the SFII in Ecuador— where both have been found to be sufficiently sensitive to identify intervention impacts. These lines now read: “Whilst both the Denver-II and the SFII have been found to be sensitive to the impact of cash transfer programs in Nicaragua (Macours, Schady, & Vakis, 2012) and Ecuador (Fernald & Hidrobo, 2011), respectively, further investigation of sensitivity to interventions for all short tests would be helpful in guiding the choice of tests.” In the immediately previous sentences (lines 478-488), we had emphasized that our recommendations on test choice were dependent on the purpose of use of the test. 

2. A second limitation is that researchers will shortly have a short global measure and a longer program evaluation measure to use instead of the Denver Screening Test or other existing alternatives. Already they have the MDAT which was not included in this study and not mentioned in the ms.

Response: Indeed, a short and long global measure for the assessment of child development, the Global Scales of Early Development or GSED, is being developed. This is exciting news for the discipline and the authors have been involved with their development. However, the GSED short and long forms will not be available for some time since both tests are still at different stages of piloting in six different countries. In addition, further piloting might be required in different areas of the world to ensure that their properties are maintained worldwide, across languages and cultures (i.e. that they are truly global measures). Furthermore, investigation of their predictive validity and response to interventions will take considerably longer to be available. The analysis performed in this work is therefore useful whilst these global measures are under development and fully validated, as noted in the Manuscript in lines 48-50 and lines 61-64. 

As explained, the MDAT was not included in the study since it was not published when the study began and was not well known at the time (Gladstone et al., 2010). In addition, time and budgetary constraints prevented us from using and evaluating other tests. We have added the need to use other tests/further investigation to be able to extrapolate findings outside urban Latin America as a limitation in lines 506-510, which now reads: “Finally, the above findings could be extrapolated to urban Colombia and possibly urban areas in other Latin American countries. Further studies would be required before extrapolating to rural areas and other LMICs, especially in Africa and Asia, which could additionally evaluate more short tests for these contexts, such as the Malawi Developmental Assessment Tool (MDAT) (Gladstone et al., 2010).” We have also added the reasons why the MDAT and other tests had not been used in lines 510-512: “The number of short tests included in the current study was limited due to time and budgetary constraints. We therefore chose tests that were commonly used in large surveys and evaluations at the time the study began”.

Abstract and elsewhere

3. Evaluation of interventions and predictive validity of such tests entail two separate issues. Predictive validity is the main phrase in the title; however the first sentence of the Abstract concerns the evaluation of interventions at scale. Evaluation is not really “hindered by the lack of developmental tests” because large-scale evaluations can validly assess representative samples to compare with controls. Your reply to this comment was arguably limited to purchases from the US, and did not consider the important value for LMIC in training local researchers and the value in translation and adaptation. Parent-report measures similar to those used in this study are subject to bias if the intervention concerns psychosocial stimulation: mothers who receive messages about child development now have beliefs about when developmental milestones occur and mothers who learn about interacting with their child will have more accurate observations of when milestones are attained (see WB toolkit p52). So use of the ASQ instead of a direct child assessment provides weak support. Frankly, the data here address predictive validity and not evaluations of interventions; reference to evaluation of interventions in the Introduction and Discussion should be dropped as it misguidedly encourages the use of screeners and parent reports in interventions. If you think that the Denver or another measure has been effectively used to evaluate a large unconditional cash transfer program, this can be mentioned in the Discussion as additional information about the test being advocated. 

Response: We agree with the Reviewer on the possible limitations of the use of screeners with parent reports in the evaluation of early childhood interventions and had already expressed this in lines 478-484. Furthermore, our results do not support the use of the ASQ-3, which is entirely parent report. They do support the use of the Denver-II, and in particular, the use of the language and cognitive scales of the test, which have at least 61% of the items directly administered to the child, as indicated in lines 179-182.

In this paragraph we also emphasize the importance of the purpose of the assessment when choosing a test (line 478) and the need for further investigation on the sensitivity to intervention of the short tests to further guide the choice of tests to use (lines 488-491). Yet, to avoid confusions we have dropped the relevant references to the use of the short tests for evaluations in the abstract (lines 18-19) and the introduction (lines 43-45) which now read, respectively: “However, progress is hindered by the lack of valid developmental tests feasible for use at large scale.” (abstract) and “However, progress of these efforts is impeded by the lack of reliable, valid, and easy-to-collect measures of ECD, particularly for children under age three years (Black et al., 2017; Fernald, Prado, Kariger, & Raikes, 2017; McCoy, Black, Daelmans, & Dua, 2016).” (introduction).

Thank you for the suggestion to refer to impact evaluation studies that have used any of the short tests. We have added two references that refer to impact evaluation studies that have used two of the short tests: these are the evaluation of cash transfer programs in Nicaragua (which used the Denver-II) and in Ecuador (which used the SFII) in lines 488-491, which read: “Whilst both the Denver-II and the SFII have been found to be sensitive to the impact of cash transfer programs in Nicaragua (Macours et al., 2012) and Ecuador (Fernald & Hidrobo, 2011), respectively, further investigation of sensitivity to interventions for all short tests would be helpful in guiding the choice of tests.” There is also recent data on the use of the Denver-II for evaluation in a large-scale program in China. One of the authors was involved with the study and selected the Denver-II test for evaluation, subsequent to the findings from the present study. A significant effect was found (analysis by Nobel Laureate, James Heckman) providing evidence of the Denver-II’ s sensitivity to intervention effects. Unfortunately, the paper is in preparation and presently unpublished, so we hesitate to cite it here. We cite it here only to provide information for the Reviewer and Editor. 

Introduction and Enrollment

4. Lines 51-53. This sentence should be revised: “Whilst existing diagnostic tests such as the Bayley Scales [5,6] are sensitive to ECD interventions [2,7–9], they are very expensive, take a long time to administer and require highly trained testers with a certain level of technical expertise [2].” Only in some countries are they expensive, take a long time to deliver and require highly trained testers … It might be more accurate to state, “According to our experience, …they are very expensive…” I would have to say that only in full do they take a long time, and rarely have I seen highly trained testers doing the training. I wonder how many other researchers had to spend 6 weeks training psychology graduates on the Bayley III. You might want to consider that this is your experience; colleagues in Peru had the same experience so I don’t doubt your word. But it is not a feature of the test and not necessarily the experience of others [I’m not even sure that the authors of the WB 2017 toolkit have the experience of training others on the Bayley].

Response: We are afraid we disagree with this point, as we have met researchers in many countries who find the Bayley difficult to give and expensive. We have rewritten the sentence in lines 50-54 as suggested and it now reads “Whilst existing diagnostic tests such as the Bayley Scales (Bayley, 1969, 2006) are sensitive to ECD interventions (Attanasio et al., 2014; Fernald et al., 2017; Hamadani, Huda, Khatun, & Grantham-McGregor, 2006; Nahar et al., 2009), according to our experience and that of many other researchers, they are very expensive, take a long time to administer and require highly trained testers with a certain level of technical expertise (Fernald et al., 2017).” This has been the experience of the many researchers we have consulted, some of whom are part of the GSED and DScore teams (Cavallera et al., 2019; Weber et al., 2019), as well as authors of the WB2017 toolkit. 

Methods

5. Lines 166-181. Thank you for adding more about whether tests were direct or parent report and how many items were parent report if the test included both. Appreciate also the information about including higher items on the screeners to avoid ceiling. These are critical features of child assessment.

Response: Thank you for suggesting it. We agree that these details add important information on the assessments used. 

6. Line 282. Thank you for elaborating on the Denver motor-adaptive scale. The original Bayley [ref 5] Performance subscale combined cognitive and fine motor, so I can see how the Denver does so.

Nothing was said about adaptation of the measures, particularly the WISC, Bayley, BSD, Denver other than translation. Could you state that no adaptations were required?

Response: With regards to the middle childhood assessments, the WISC-V was the only test that had to be translated. For the other tests, minor wording and phrasing adaptations had to be made after piloting to adjust the language to the local use of Spanish, as indicated. We have now clarified that no other adaptations were found necessary in lines 234-236: “We translated the WISC-V report forms and manuals and piloted the translations. All other test materials were available in Spanish and minor wording/phrasing modifications were made, after piloting, to reflect Colombian Spanish. No other adaptations were found necessary”. 

With respect to the enrollment assessments (Bayley-III, Denver-II, etc.), details had been provided in earlier work (see Appendix I in pages 40-42). Owing to space restrictions and to avoid duplication, we do not add these details to the current Manuscript. We have added a paragraph explaining whether any translation/adaptation was required and have provided the reference to the earlier work containing the full details in lines 198-203: “In preparation for testing, we translated and back-translated the Bayley-III; as well as the BDI-2 manual, and the WHO-Motor report forms and manual. Short tests in battery A were all available in Spanish. All translations and official Spanish versions were piloted and, subsequently, minor wording/phrasing modifications were made in order to reflect Colombian Spanish. Similarly, a few images had to be contextualized. Full adaptation details were provided earlier (Rubio-Codina, Araujo, Attanasio, & Grantham-McGregor, 2016).” 

Results

7. Thank you for clarifying improvement in retest scores.

Thank you for providing age correlations in S2.

Table S2. p cannot be <.0000. Better to write p<.0001.

Response: Thanks for pointing this out, we have modified the p-value in Table S2 as suggested. We have also modified the p-values in Table 4. We are glad that the other clarifications were helpful. 

8. Table 1. Internally standardized scores are not very meaningful. I can understand why you used them for analyses but they are not meaningful for readers who are usually given standardized Bayleys and raw HOME. Some of the raw scores were useful in S2 but it would be good to see raw HOME in order to compare with other studies (rather than internally standardized) and externally standardized scores for Bayley III and WISC. I noticed that the Bayley III standardized scores in the 2016 paper looked similar to US and UK means; they won’t always be this high in other regions of the world. Did you expect standardized IQ scores to drop in middle childhood?

Response: Externally standardized WISC-V scores are already provided in Table 1. We have added Bayley-III externally standardized scores (composites) in Table S1. However, we would caution against the use of external norms for samples in populations other than the norming population. 

In discussions with the authors of the HOME, we adapted the test substantially in order to make it more suitable for the study population. This is noted in lines 245-246: “In middle childhood, an adaptation of the Middle Childhood Home Observation for Measurement of the Environment (MC-HOME) (Bradley, Caldwell, Rock, Hamrick, & Harris, 1988; Caldwell & Bradley, 2003) was collected”. Therefore, comparisons of the HOME raw score in Bogota with the HOME raw scores in other studies might be somewhat misleading and, hence, we rather not show the HOME raw score in the Manuscript, particularly as the HOME is not one of the tests under study, but only acts as a ‘covariate’. However, in the next table, we report the HOME total raw score and by subscale for the Reviewer’s reference. 

9. For the other measures (ASQ, BSI, Denver) can you provide some central score or % above/below a cut-off? It is problematic that these tests do not allow for scores to be shown.

Response: We had provided raw scores of all tests in earlier work (Rubio-Codina, Araujo, Attanasio, & Grantham-McGregor, 2016, Table A1 in Appendix III, page 44). As noted in the Manuscript, the short tests used were mostly screeners and their mode of administration was modified, for some of the tests, so as to increase their ability to assess children along the developmental continuum. The language in some of the items had also been modified to ensure cultural appropriateness and functional equivalence (details of the adaptations are also provided in earlier work, Rubio-Codina, Araujo, Attanasio, & Grantham-McGregor (2016), Appendix I, pages 40-42). For some short tests, these modifications made it difficult to interpret external norms. For example, we used 9 items (as opposed to the original 6) in the ASQ and the cut-off points in the original test were no longer applicable and could be misleading. We had no plan to use them as screeners or validate cutoff points. The study design and sample were not appropriate for this. 

The purpose of the current paper is to assess the relative performance of the various short tests used to predict concurrent and future development against a gold standard (the Bayley-III). To this end, we think that the analyses carried out using internally standardized scores is more adequate. 

Discussion

10. Line 473. “recall bias” is probably not the right phrase to describe the bias of mothers who have attended a stimulation class. It may be more like an “observation bias” due to spending more time with their child, and a “desirability bias” due to knowing what milestones of development to expect.

Response: Thanks for raising this point and for the suggestions. We have relabeled and explained the sources of bias in lines 478-484, which now read “Caregiver reports might be better suited for the evaluation of population-based indicators and less convenient to evaluate psychosocial stimulation programs as they may suffer from “observation” bias, if mothers in the treatment group spend more time with the child and are more aware of the process of development/achievement of milestones as a result of the intervention. Moreover, intervened mothers may be biased towards making optimistic claims of their children’s development to report on intervention success (“desirability” bias).”

11. Line 487 etc. My previous comment re the FCI was “prevalence for several items asymptotes at a high rate at 4 months of age, and other items like counting and drawing asymptote at a low rate at 4 years. So not all items are relevant for low-income countries at all these ages.” The ages being referred to are between 4m and 4 year, not simply those two ages. If high rates asymptote at 4m it means they have hit ceiling for all older children. If rates asymptote at a low level by 4 years, it means they are at floor for many months before this. Have you assessed variation in your FCI scores for your full sample by age? Thank you for pointing out that the FCI will have different means in Africa and Asia.

Response: This is a very interesting point, thanks for the clarification. The following two tables show the variation of the FCI scores by component (type of play material and type of play activity) by age groups. A few of the items (in italics) show little variation over age and seem to be at a ceiling (“toys for moving around”, “singing songs”, “taking child outside the home place/go for a walk”). The remaining items show a gradient with age. 

SEE TABLES in RESPONSE DOCUMENT ENCLOSED

We have reconstructed the FCI play activities and play materials score without the 3 items that do not show any variation and have correlated the new score to IQ and school achievement at ages 6-8 years by age group in early childhood. The correlations remain very similar to those previously obtained, as shown in the next table (see greyed area and text in blue). Therefore, we do not think that the potential lack of variability in certain FCI items is of concern and prefer to use the instrument as it was designed. The fact that the predictive validity of the FCI score without these items improves slightly after 19 months might be an indication that our estimates of predictive validity are conservative. We thank the review for raising this point as it has been an interesting analysis to carry out and an aspect to keep in mind in future work. 

12. Line 488. One problem with using the FCI to evaluate a psychosocial program is that it assesses stimulation but not responsive caregiving.

Response: Thanks, we have added this in current lines 493-496, which now read: “It is free, quick (10-15 min) and easy to give, provides information on useful activities for parents (although not on responsive caregiving) and has been widely used in international surveys—most notably, UNICEF’s Multiple Indicator Cluster Surveys (MICS-ECDI, n.d.).”

13. The Denver Screening test appears to be the strongest substitute for the Bayley. It may have the same administration duration, requires caregiver report for 39% of the items, and costs less in the US. The authors did not provide information on length of training testers and adaptation required. This would help prospective researchers know what is required. 

Response: Details on training and adaptation were published earlier (Rubio-Codina, Araujo, Attanasio, & Grantham-McGregor, 2016; Rubio-Codina, Araujo, Attanasio, Muñoz, & Grantham-McGregor, 2016). There is a reference to both papers in lines 202-203 “Full adaptation details were provided earlier (Rubio-Codina, Araujo, Attanasio, & Grantham-McGregor, 2016).” and lines 210-212 “More details on the enrollment tests, their costs, and on the administration and training procedures were provided previously [10, 33]” of the current Manuscript. As noted in our response to Reviewer 1 comment 6, we have added some details on the adaptations needed for the early childhood assessments in lines 198-203. We hesitate to add any further details to avoid duplications and due to space restrictions. Please note that the Denver-II took a considerably shorter amount of time to administer than the Bayley-III (27 minutes vs. 83 minutes), as indicated in lines 204-210. 

14. Limitations. Reasons for not using the MDAT should be mentioned in the limitations. I understand that you were not aware of its launch in 2010 but readers will wonder why it was ignored given its currently popularity.

Response: We have added these reasons as limitations in the Discussion in lines 506-512, which read: “Finally, the above findings could be extrapolated to urban Colombia and possibly urban areas in other Latin American countries. Further studies would be required before extrapolating to rural areas and other LMICs, especially in Africa and Asia, which could additionally evaluate more short tests for these contexts, such as the Malawi Developmental Assessment Tool (MDAT) (Gladstone et al., 2010). The number of short tests included in the current study was limited due to time and budgetary constraints. We therefore chose tests that were commonly used in large surveys and evaluations at the time the study began.” 

15. Another limitation is that height and stunting may be less predictive in Colombia than in Africa because of the narrow range of values, particularly in the stunted range.

Response: Thanks. We agree with the Reviewer and believe that the point is covered in the paragraph corresponding to lines 501-505: “Furthermore, although stunting has been used as an indicator of inadequate child development globally (Black et al., 2017; Grantham-McGregor et al., 2007), the FCI was a better predictor of future overall intellectual functioning and school achievement in this population. If these findings are replicated in countries with different home environments and stunting severity and prevalence, combining both indicators would be a more effective population-based indicator.” and in the following paragraph in lines 506-512 (see response to Comment 14 above). 

Reviewer #2: 

1. The authors have improved their manuscript and they sufficiently addressed my comments.

For their understanding: The Bayley III is also available in Dutch with Dutch norms, and we published the differences between the US norms and the Dutch norms in PLOS One.

Response: We are very glad to hear that we have sufficiently addressed Reviewer’s 2 comments. We were not aware that the Bayley-III is also available in Dutch and has Dutch norms and thank the Reviewer for bringing it to our attention. We have not modified the Manuscript with regards to Reviewer’s 2 comments. 

References

Attanasio, O. P., Fernandez, C., Fitzsimons, E. O. A., Grantham-McGregor, S. M., Meghir, C., & Rubio-Codina, M. (2014). Using the infrastructure of a conditional cash transfer program to deliver a scalable integrated early child development program in Colombia: cluster randomized controlled trial. BMJ, 349(sep29 5), g5785–g5785. https://doi.org/10.1136/bmj.g5785

Bayley, N. (1969). Bayley Scales of Infant Development. New York: Psychological Corp.

Bayley, N. (2006). Bayley Scales of Infant and Toddler Development–Third Edition: Technical manual. San Antonio, TX: Harcourt Assessment.

Black, M. M., Walker, S. P., Fernald, L. C., Andersen, C. T., DiGirolamo, A. M., Lu, C., … Committee, L. E. C. D. S. S. (2017). Early childhood development coming of age: science through the life course. The Lancet, 389(10064), 77–90. https://doi.org/10.1016/S0140-6736(16)31389-7

Bradley, R. H., Caldwell, B. M., Rock, S. L., Hamrick, H. M., & Harris, P. (1988). Home Observation for Measurement of the Environment: Development of a Home Inventory for use with families having children 6 to 10 years old. Contemporary Educational Psychology, 13(1), 58–71. https://doi.org/10.1016/0361-476X(88)90006-9

Caldwell, B. M., & Bradley, R. H. (2003). HOME Inventory Administration Manual. Comprehensive Edition. (U. of Arkansas, Ed.). Print Design, Little Rock.

Cavallera, V., Black, M., Bromley, K., Cuartas, J., Eekhout, I., Fink, G., … Dua, T. (2019). The Global Scale for Early Development ( GSED ). Early Childhood Matters, 80–84.

Fernald, L. C. H., Prado, E., Kariger, P., & Raikes, A. (2017). A Toolkit for Measuring Early Childhood Development in Low and Middle-Income Countries (Working Paper No. 122031). Washington, DC. Retrieved from http://documents.worldbank.org/curated/en/384681513101293811/A-toolkit-for-measuring-early-childhood-development-in-low-and-middle-income-countries

Fernald, L. C., & Hidrobo, M. (2011). Effect of Ecuador’s cash transfer program (Bono de Desarrollo Humano) on child development in infants and toddlers: a randomized effectiveness trial. Social Science and Medicine, 72(9), 1437–1446. https://doi.org/0.1016/j.socscimed.2011.03.005

Gladstone, M., Lancaster, G. A., Umar, E., Nyirenda, M., Kayira, E., Van, N. R., … Smyth, R. L. (2010). The Malawi Developmental Assessment Tool (MDAT): The Creation, Validation, and Reliability of a Tool to Assess Child Development in Rural African Settings, 7(5). https://doi.org/10.1371/journal.pmed.1000273

Grantham-McGregor, S., Cheung, Y. B., Cueto, S., Glewwe, P., Richter, L., & Strupp, B. (2007). Developmental potential in the first 5 years for children in developing countries. The Lancet, 369(9555), 60–70. https://doi.org/10.1016/S0140-6736(07)60032-4

Hamadani, J. D., Huda, S. N., Khatun, F., & Grantham-McGregor, S. M. (2006). Psychosocial stimulation improves the development of undernourished children in rural Bangladesh. The Journal of Nutrition, 136(10), 2645–2652. https://doi.org/136/10/2645 [pii]

Macours, K., Schady, N., & Vakis, R. (2012). Cash Transfers, Behavioral Changes, and Cognitive Development in Early Childhood: Evidence from a Randomized Experiment. American Economic Journal: Applied Economics, 4(2), 247–273. https://doi.org/doi
http://dx.doi.org/10.1257/app.4.2.247.

McCoy, D. C., Black, M. M., Daelmans, B., & Dua, T. (2016). Measuring development in children from birth to age 3 at population level. Early Childhood Matters, 34–39.

MICS-ECDI. (n.d.). MICS UNICEF. Retrieved from http://mics.unicef.org/

Nahar, B., Hamadani, J. D., Ahmed, T., Tofail, F., Rahman, a, Huda, S. N., & Grantham-McGregor, S. M. (2009). Effects of psychosocial stimulation on growth and development of severely malnourished children in a nutrition unit in Bangladesh. European Journal of Clinical Nutrition, 63(6), 725–731. https://doi.org/10.1038/ejcn.2008.44

Rubio-Codina, M., Araujo, M. C., Attanasio, O., & Grantham-McGregor, S. (2016). Concurrent Validity and Feasibility of Short Tests Currently Used to Measure Early Childhood Development in Large Scale Studies: Methodology and Results (No. IDB-WP-723). IDB-WP-723. Washington, D.C. https://doi.org/10.1371/journal.pone.0160962

Rubio-Codina, M., Araujo, M. C., Attanasio, O., Muñoz, P., & Grantham-McGregor, S. (2016). Concurrent Validity and Feasibility of Short Tests Currently Used to Measure Early Childhood Development in Large Scale Studies. Plos One, 11(8), e0160962. https://doi.org/10.1371/journal.pone.0160962

Weber, A. M., Rubio-Codina, M., Walker, S. P., van Buuren, S., Eekhout, I., Grantham-McGregor, S. M., … Black, M. M. (2019). The D-score: a metric for interpreting the early development of infants and toddlers across global settings. Mimeo.

---

## [Decision Letter · Decision Letter 2]

8 Jan 2020

PONE-D-19-20029R2

Predictive validity in middle childhood of short tests of early childhood development used in large scale studies compared to the Bayley-III, the Family Care Indicators, height-for-age and stunting: A longitudinal study in Bogota, Colombia

PLOS ONE

Dear Dr Rubio-Codina,

Thank you for submitting your manuscript to PLOS ONE. After careful consideration, we feel that it has merit but does not fully meet PLOS ONE’s publication criteria as it currently stands. Therefore, we invite you to submit a revised version of the manuscript that addresses the points raised during the review process.

We would appreciate receiving your revised manuscript by Feb 22 2020 11:59PM. To enhance the reproducibility of your results, we recommend that if applicable you deposit your laboratory protocols in protocols.io, where a protocol can be assigned its own identifier (DOI) such that it can be cited independently in the future. For instructions see: http://journals.plos.org/plosone/s/submission-guidelines#loc-laboratory-protocols

We look forward to receiving your revised manuscript.

Kind regards,

Thach Duc Tran, M.Sc., Ph.D.

Academic Editor

PLOS ONE

Reviewers' comments:

Reviewer's Responses to Questions

**Comments to the Author**

1. If the authors have adequately addressed your comments raised in a previous round of review and you feel that this manuscript is now acceptable for publication, you may indicate that here to bypass the “Comments to the Author” section, enter your conflict of interest statement in the “Confidential to Editor” section, and submit your "Accept" recommendation.

Reviewer #1: (No Response)

2. Is the manuscript technically sound, and do the data support the conclusions?

Reviewer #1: Partly

3. Has the statistical analysis been performed appropriately and rigorously? 

Reviewer #1: Yes

4. Have the authors made all data underlying the findings in their manuscript fully available?

Reviewer #1: Yes

5. Is the manuscript presented in an intelligible fashion and written in standard English?

Reviewer #1: Yes

6. Review Comments to the Author

Reviewer #1: PONE-D-19-20029R2

Predictive validity in middle childhood of short tests of early childhood development used in large scale studies compared to the Bayley-III, the Family Care Indicators, height-for-age and stunting: A longitudinal study in Bogota, Colombia

Overall, the value of the paper is in demonstrating predictive validity of various measures of child development. The data addressing this goal make a novel contribution. It is less valuable in guiding researchers’ selection of a measure. This is because the current data are limited in various ways: 1) they are from urban Colombia and unlikely to generalize to rural contexts or to Africa or Asia; 2) actual data on the short tests are presented in internally standardized form and so uninterpretable to other researchers; 3) the short tests/screeners had to be modified to prevent ceiling effects and so are inaccessible to other researchers; 4) the popular MDAT was not included and the GSED will shortly be the preferred measure. I suggest that the aim of the current study be solely to provide predictive validity of the measures, though in the Discussion the authors could offer an opinion about use of the Denver if researchers need a short, easy-to-administer test in the near future.

“Data” is a plural noun; verbs should accord.

Some figure and table titles use a preposition that does not make sense to me. E.g. S3 Fig “… validity of Bayley-III and short tests at 6-42 months of school achievement at 6-8 years…”. The preposition “of” seems incorrect, and maybe should be “with” or “and”. This occurs in several other tables/figures, e.g. line 351 and 363 “of FSIQ”.

Line 51. I would delete the word “diagnostic” because none of the studies cited here is using the Bayley for diagnostic purposes; it is used as a research tool to assess child development. “Diagnosis” implies use of a test to identify problems needing treatment.

Lines 52-54. Citation [2] referring to the WB toolkit is actually a citation to the Pearson description of the Bayley, and not derived from Fernald et al’s personal experience with the Bayley. I don’t mind that this be left unrevised, but it makes the authors appear naïve.

Line 77. Who thinks that administering the Bayley at a center is the “ideal condition”? An unfamiliar center might make young children wary and distracted if they have not habituated to the surroundings.

Line 128. The authors responded to my concern about using these data to inform evaluations by saying that they agreed to delete such phrases. However, they need to delete the phrase “impact evaluations”. Possibly other data address the use of the Denver cognitive and language items in cash transfer, but the data from this study support its use for descriptive, not evaluative, designs.

Line 489. I appreciate the paragraph on the choice of a measure being linked to its objective. However, the final statement about “further investigations” requires citations of existing research. There are already many studies using these measures to evaluate interventions and it is clear that many are not sensitive. The ASQ, for example, is frequently used and shows good effect sizes, but because mother-report is biased the findings are suspect.

It is surprising that no adaptations except translation and minor wording had to be made on the enrollment tests. Is this because the setting is urban Latin America? The stimulus booklet and picture book of the Bayley would need extensive adaptation in countries of Africa and Asia. So the authors’ claim to be searching for an easy-to-adapt measure has not been tested in places where adaptation would be more extensive.

The authors are unable to present anything other than internally standardized scores for tests other than the Bayley. The reason given is that they were screening tests and so required the addition of items and other major modifications. This is a major barrier to other researchers using those measures (the adapted measures with additional items would have to be obtained from these authors) and a major limitation in presenting findings to be compared with findings based on different measures of development. So the authors’ guidance on selection of a short measure is limited. Without comparable and interpretable data on the different short-form measures, researchers will not be able to use this study in their future work. The authors have selected the Denver in future work in China, but without interpretable evidence other researchers would have to take their word on faith.

7. PLOS authors have the option to publish the peer review history of their article (what does this mean?). If published, this will include your full peer review and any attached files.

Reviewer #1: No

---

## [Author Response · Author response to Decision Letter 2]

3 Feb 2020

PONE-D-19-20029R2

Predictive validity in middle childhood of short tests of early childhood development used in large scale studies compared to the Bayley-III, the Family Care Indicators, height-for-age, and stunting: A longitudinal study in Bogota, Colombia

Responses from the authors to the Reviewer’s comments 

We thank Reviewer 1 for reading through our second set of responses in detail. We respond below to the new set of comments. For clarity we retain the comments and add our response in italics. We have numbered the reviewers’ comments to facilitate cross-references to our responses. Please note that the references to the line numbers where edits have been made correspond to the version of the Manuscript without tracked changes (clean version or Manuscript). 

Reviewer Comments to the Author

Reviewer #1: PONE-D-19-20029R2

Predictive validity in middle childhood of short tests of early childhood development used in large scale studies compared to the Bayley-III, the Family Care Indicators, height-for-age, and stunting: A longitudinal study in Bogota, Colombia

1. Overall, the value of the paper is in demonstrating predictive validity of various measures of child development. The data addressing this goal make a novel contribution. It is less valuable in guiding researchers’ selection of a measure. This is because the current data are limited in various ways: 1) they are from urban Colombia and unlikely to generalize to rural contexts or to Africa or Asia; 2) actual data on the short tests are presented in internally standardized form and so uninterpretable to other researchers; 3) the short tests/screeners had to be modified to prevent ceiling effects and so are inaccessible to other researchers; 4) the popular MDAT was not included and the GSED will shortly be the preferred measure. 5) I suggest that the aim of the current study be solely to provide predictive validity of the measures, though in the Discussion the authors could offer an opinion about use of the Denver if researchers need a short, easy-to-administer test in the near future.

Response: The limitations mentioned by the reviewer were already acknowledged in the Manuscript and/or have been addressed in prior responses to comments from Reviewer 1. Specifically, 

1.1. The need to use other tests/further investigation to be able to extrapolate findings outside urban Latin America was mentioned in lines 507-513: “Finally, the above findings could be extrapolated to urban Colombia and possibly urban areas in other Latin American countries. Further studies would be required before extrapolating to rural areas and other LMICs, especially in Africa and Asia, which could additionally evaluate more short tests for these contexts, such as the Malawi Developmental Assessment Tool (MDAT) (Gladstone et al., 2010).” 

1.2. We recognize that the choice of internally or externally standardized scores is sometimes controversial. However, we disagree that the results would be “uninterpretable” because internally standardized scores were used. Internally standardized are the preferred scores in order to facilitate the comparison of performance of the various short tests, as indicated in lines 270-277: “No test had norms for Colombia. Therefore, we internally standardized the raw scores over age using age-conditional means and SDs, computed non-parametrically, after removing testers’/interviewers’ effects, as done in previous work [10]. This is, for each value of the residual of the raw scores on tester or interviewer dummies, we constructed a Z-score by subtracting the age-conditional mean and dividing by the age-conditional SD, both computed using local polynomial regressions. Unlike using norms from the reference populations (i.e. externally standardized scores) for each test, this standardization method handles age consistently across tests, which facilitates comparisons”. Lack of norms for tests is common in LMICs. In this situation, one may equally argue that using externally standardized scores in countries where the test is not standardized would be difficult to interpret.

Please note that all prior reviewers of this and the earlier paper investigating concurrent validity (Rubio-Codina, Araujo, Attanasio, Muñoz, & Grantham-McGregor, 2016), concurred with the use of the internally standardized scores as the best practice for the type of analysis we carry out. 

1.3. It is not clear why these modifications are inaccessible to other researchers since they are explained in detail in previous work and in the Manuscript (see lines 172-180) “As reported earlier (Rubio-Codina, Araujo, Attanasio, Muñoz, et al., 2016), all tests were administered following manual instructions except the ASQ-3, which we modified as follows. A) Given the low literacy levels of some caregivers, caregiver-completed items were administered by interview in order to ensure all mothers understood the questions similarly. B) Furthermore, the child was tested if the caregiver did not know the answer. C) In addition, whenever the scale ceiling was reached, we added the next three non-overlapping items from subsequent questionnaires. This reduced the number of children on the test ceiling by 10.5–15.5% to levels of 1.7–4.8%, depending on the domain, thus increasing the variability in development captured by the test. Similar adaptations have been used elsewhere (Fernald, Kariger, Hidrobo, & Gertler, 2012).” It is worth highlighting the fact that these modifications have been used by other researchers in prior published work (Fernald et al., 2012). 

1.4. The reasons why the MDAT was not included in the study have already been explained in prior responses and in the Manuscript—please see lines 511-513: “The number of short tests included in the current study was limited due to time and budgetary constraints. We therefore chose tests that were commonly used in large surveys and evaluations at the time the study began” and in the paragraph copied in point 1) above. We had already acknowledged that the GSED will be available in the future in Manuscript lines 48-50 “Efforts are currently underway to develop global (i.e. culturally-neutral or very easy-to-adapt), valid, feasible, freely accessible population-based instruments, as well as individual-level instruments suitable to evaluate interventions, for children 0-3 years (Cavallera et al., 2019; McCoy, Black, Daelmans, & Dua, 2016).” However, the GSED will not be available immediately and these results can be helpful until then, as indicated in lines 61-64: “More generally, it is critical to identify readily available, reliable, valid and feasible ECD measures for use in large samples until the population-based and individual-level instruments currently under development become available and are shown to have both concurrent and predictive validity.” Moreover, the fact that the GSED is available does not necessarily imply that any of the prior existing tests are no longer used.

1.5. The aim of the study has been revised to highlight the study of predictive validity. However, since this can help guide the choice of tests, we have kept a mention to it and have rephrased lines 124-129 accordingly, which now read: “The study aimed to investigate the ability of the screeners (and other short tests and measures typically used as ECD proxies), given at three different age ranges, to predict later functioning across the developmental range, which can help to guide the choice of predictive, feasible and easy-to-use readily available instruments for the assessment of very young children’s development in large scale studies in LMICs, for either research purposes or as population-based indicators of ECD.”

2. “Data” is a plural noun; verbs should accord.

Response: We have revised the Manuscript for grammatical accuracy. 

3. Some figure and table titles use a preposition that does not make sense to me. E.g. S3 Fig “… validity of Bayley-III and short tests at 6-42 months of school achievement at 6-8 years…”. The preposition “of” seems incorrect, and maybe should be “with” or “and”. This occurs in several other tables/figures, e.g. line 351 and 363 “of FSIQ”.

Response: Thanks. We have revised the Manuscript for grammatical accuracy.

4. Line 51. I would delete the word “diagnostic” because none of the studies cited here is using the Bayley for diagnostic purposes; it is used as a research tool to assess child development. “Diagnosis” implies use of a test to identify problems needing treatment.

Response: The word diagnostic is used to describe the Bayley-III, which is indeed a diagnostic test. We are not implying that the studies listed are being used to ‘diagnose’ children, however, we have removed the word so that there is no misunderstanding. And inserted “Whilst existing full developmental tests such as the Bayley Scales (…)”.

5. Lines 52-54. Citation [2] referring to the WB toolkit is actually a citation to the Pearson description of the Bayley, and not derived from Fernald et al’s personal experience with the Bayley. I don’t mind that this be left unrevised, but it makes the authors appear naïve.

Response: We have left the reference since the WB tookit concurs with these points: “administration of these tests requires a certain level of technical expertise and may be cumbersome for assessments that require a lot of manipulatives (e.g., Bayley Scales of Infant Development).” (Fernald, Prado, Kariger, & Raikes, 2017, p. 60). The authors of the toolkit also refer to time, cost, and copyright issues as limitations of the Bayley-III for administration in the context of program evaluations and hypothesis-driven research (see p. 65), as we had explained in our first set of responses to reviewers. 

6. Line 77. Who thinks that administering the Bayley at a center is the “ideal condition”? An unfamiliar center might make young children wary and distracted if they have not habituated to the surroundings.

Response: A busy and noisy home with many interruptions may also distract the child. When comparing test scores, it is important that all children experience similar conditions as far as possible, that is the main reason for testing in centers because homes vary a lot and it is not possible to control interruptions. We have revised the sentence in lines 76-78, which now reads: “In contrast, the Bayley-III, which we considered our ‘gold standard’, was administered by psychologists at a center to minimize distractions and standardize the test experience as far as possible—and therefore, under preferable conditions.”

7. Line 128. The authors responded to my concern about using these data to inform evaluations by saying that they agreed to delete such phrases. However, they need to delete the phrase “impact evaluations”. Possibly other data address the use of the Denver cognitive and language items in cash transfer, but the data from this study support its use for descriptive, not evaluative, designs.

Response: Apologies for this omission. The words ‘impact evaluations’ have been deleted from line 129. Lines 123-129 now read: “The study aimed to investigate the ability of the screeners (and other short tests and measures typically used as ECD proxies), given at three different age ranges, to predict later functioning across the developmental range, which can help to guide the choice of predictive, feasible and easy-to-use readily available instruments for the assessment of very young children’s development in large scale studies in LMICs, for either research purposes or as population-based indicators of ECD.”

8. Line 489. I appreciate the paragraph on the choice of a measure being linked to its objective. However, the final statement about “further investigations” requires citations of existing research. There are already many studies using these measures to evaluate interventions and it is clear that many are not sensitive. The ASQ, for example, is frequently used and shows good effect sizes, but because mother-report is biased the findings are suspect.

Response: We have listed references to evaluations in LAC, the context of relevance, that have used the Denver-II and the SFI-II, which are the two tests we would suggest using (in addition to the FCI). Please see lines 489-492: “Whilst both the Denver-II and the SFII have been found to be sensitive to the impact of cash transfer programs in Nicaragua (Macours, Schady, & Vakis, 2012) and Ecuador (Fernald & Hidrobo, 2011), respectively, further investigation of sensitivity to interventions for all short tests would be helpful in guiding the choice of tests. ” We are not suggesting researchers use any of the other tests, and less so the ASQ-3, as it shows very poor predictive validity (and had showed very poor concurrent validity in our previous work). Therefore, we do not see a need to provide references to those. 

9. It is surprising that no adaptations except translation and minor wording had to be made on the enrollment tests. Is this because the setting is urban Latin America? The stimulus booklet and picture book of the Bayley would need extensive adaptation in countries of Africa and Asia. So the authors’ claim to be searching for an easy-to-adapt measure has not been tested in places where adaptation would be more extensive.

Response: We do not understand this point. Many of the enrollment tests had to be translated and pictures and wording had to be replaced and further modified after piloting. This was already explained in lines 199-204, in response to Reviewer 1, Comment 6 in our second set of revisions. These sentences read: “In preparation for testing, we translated and back-translated the Bayley-III; as well as the BDI-2 manual, and the WHO-Motor report forms and manual. Short tests in battery A were all available in Spanish. All translations and official Spanish versions were piloted and, subsequently, minor wording/phrasing modifications were made in order to reflect Colombian Spanish. Similarly, a few images had to be contextualized. Full adaptation details were provided earlier (Rubio-Codina, Araujo, Attanasio, & Grantham-McGregor, 2016).” As explained in our prior revision, and owing to space restrictions and to avoid duplication with the previous publication, we did not add more adaptation details of the enrollment tests in the current manuscript. A link to the paper containing a complete description of the adaptation details was provided in the previous response document for the Reviewer’s reference and is also available here (see Appendix I in pages 40-42).

10. The authors are unable to present anything other than internally standardized scores for tests other than the Bayley. The reason given is that they were screening tests and so required the addition of items and other major modifications. This is a major barrier to other researchers using those measures (the adapted measures with additional items would have to be obtained from these authors) and a major limitation in presenting findings to be compared with findings based on different measures of development. So the authors’ guidance on selection of a short measure is limited. Without comparable and interpretable data on the different short-form measures, researchers will not be able to use this study in their future work. The authors have selected the Denver in future work in China, but without interpretable evidence other researchers would have to take their word on faith.

Response: We disagree this is a major barrier to other researchers as explained in our response to Comment 1.2 and 1.3 above. However, we are adding the raw scores of the short tests in Table S1. 

References 

Cavallera, V., Black, M., Bromley, K., Cuartas, J., Eekhout, I., Fink, G., … Dua, T. (2019). The Global Scale for Early Development ( GSED ). Early Childhood Matters, 80–84.

Fernald, L. C. H., Kariger, P., Hidrobo, M., & Gertler, P. J. (2012). Socioeconomic gradients in child development in very young children: Evidence from India, Indonesia, Peru, and Senegal. Proceedings of the National Academy of Sciences, 109(Supplement_2), 17273–17280. https://doi.org/10.1073/pnas.1121241109

Fernald, L. C., & Hidrobo, M. (2011). Effect of Ecuador’s cash transfer program (Bono de Desarrollo Humano) on child development in infants and toddlers: a randomized effectiveness trial. Social Science and Medicine, 72(9), 1437–1446. https://doi.org/0.1016/j.socscimed.2011.03.005

Fernald, Lia C. H., Prado, E., Kariger, P., & Raikes, A. (2017). A Toolkit for Measuring Early Childhood Development in Low and Middle-Income Countries (Working Paper No. 122031). Washington, DC. Retrieved from http://documents.worldbank.org/curated/en/384681513101293811/A-toolkit-for-measuring-early-childhood-development-in-low-and-middle-income-countries

Gladstone, M., Lancaster, G. A., Umar, E., Nyirenda, M., Kayira, E., Van, N. R., … Smyth, R. L. (2010). The Malawi Developmental Assessment Tool (MDAT): The Creation, Validation, and Reliability of a Tool to Assess Child Development in Rural African Settings, 7(5). https://doi.org/10.1371/journal.pmed.1000273

Macours, K., Schady, N., & Vakis, R. (2012). Cash Transfers, Behavioral Changes, and Cognitive Development in Early Childhood: Evidence from a Randomized Experiment. American Economic Journal: Applied Economics, 4(2), 247–273. https://doi.org/doi
http://dx.doi.org/10.1257/app.4.2.247.

McCoy, D. C., Black, M. M., Daelmans, B., & Dua, T. (2016). Measuring development in children from birth to age 3 at population level. Early Childhood Matters, 34–39.

Rubio-Codina, M., Araujo, M. C., Attanasio, O., & Grantham-McGregor, S. (2016). Concurrent Validity and Feasibility of Short Tests Currently Used to Measure Early Childhood Development in Large Scale Studies: Methodology and Results (No. IDB-WP-723). IDB-WP-723. Washington, D.C. https://doi.org/10.1371/journal.pone.0160962

Rubio-Codina, M., Araujo, M. C., Attanasio, O., Muñoz, P., & Grantham-McGregor, S. (2016). Concurrent Validity and Feasibility of Short Tests Currently Used to Measure Early Childhood Development in Large Scale Studies. Plos One, 11(8), e0160962. https://doi.org/10.1371/journal.pone.0160962

---

## [Decision Letter · Decision Letter 3]

10 Feb 2020

PONE-D-19-20029R3

Predictive validity in middle childhood of short tests of early childhood development used in large scale studies compared to the Bayley-III, the Family Care Indicators, height-for-age, and stunting: A longitudinal study in Bogota, Colombia

PLOS ONE

Dear Dr Rubio-Codina,

Thank you for submitting your manuscript to PLOS ONE. After careful consideration, we feel that it has merit but does not fully meet PLOS ONE’s publication criteria as it currently stands. Therefore, we invite you to submit a revised version of the manuscript that addresses the points raised during the review process.

We would appreciate receiving your revised manuscript by Mar 26 2020 11:59PM. To enhance the reproducibility of your results, we recommend that if applicable you deposit your laboratory protocols in protocols.io, where a protocol can be assigned its own identifier (DOI) such that it can be cited independently in the future. For instructions see: http://journals.plos.org/plosone/s/submission-guidelines#loc-laboratory-protocols

We look forward to receiving your revised manuscript.

Kind regards,

Thach Duc Tran, M.Sc., Ph.D.

Academic Editor

PLOS ONE

Reviewers' comments:

Reviewer's Responses to Questions

**Comments to the Author**

1. If the authors have adequately addressed your comments raised in a previous round of review and you feel that this manuscript is now acceptable for publication, you may indicate that here to bypass the “Comments to the Author” section, enter your conflict of interest statement in the “Confidential to Editor” section, and submit your "Accept" recommendation.

Reviewer #1: (No Response)

2. Is the manuscript technically sound, and do the data support the conclusions?

Reviewer #1: Partly

3. Has the statistical analysis been performed appropriately and rigorously? 

Reviewer #1: Yes

4. Have the authors made all data underlying the findings in their manuscript fully available?

Reviewer #1: Yes

5. Is the manuscript presented in an intelligible fashion and written in standard English?

Reviewer #1: Yes

6. Review Comments to the Author

Reviewer #1: PONE-D-19-20029 R3

Predictive validity in middle childhood of short tests of early childhood development used in large scale studies compared to the Bayley-III, the Family Care Indicators, height-for-age, and stunting: A longitudinal study in Bogota, Colombia

The authors have made some of the requested revisions and resisted others. I suppose some of the difference is between a statistical preference for drawing conclusions and giving advice based on the statistical analyses, and a preference for interpreting the methods and analyses in terms of child development within the context of an intervention.

So we can agree to disagree on whether it is better to test a child in his/her home where they have habituated to the "distractions" or to test in a controlled strange environment. Maybe you have to know about the "strange situation" to understand the difference.

We can also agree to disagree on whether Fernald was expressing an expert opinion on the Bayley or simply writing what Pearson Assessment has on their materials.

However, my statement that the authors should not state as an aim that they would provide guidance on which short measure to use was not sufficiently addressed. My point was not that you failed to mention the limitations, but that the limitations were sufficiently severe as to reduce the value of your advice on selection of a measure for others' research.

1.2. I understand that internally standardized scores make for good statistical comparisons across measures but they do not provide transparent information about the children's mental development using these measures. This study could be more than an exercise in statistical comparison if you want to provide guidance on selection of measures. The request for actual scores was answered by providing a supplementary table. The Denver scores were provided and I can see that they are aligned with the Bayley's in a ways that are similar to other papers and so are more transparent.

1.3. I can understand that other members of your team (e.g. Fernald) and other co-authors might be able to follow your non-standardized instructions to avoid ceiling on screening measures, but for obvious reasons other researchers will/have not been able to. Consequently, while your predictive data might be rigorous, others using these measures do not have the required adaptations to yield such valid data.

1.4. I understand why you did not include the MDAT or refer to the GSED, but that does not eliminate the limitation that you advise people to use the Denver while the MDAT may have been a better test.

The authors left unchanged their aim as: "The study aimed to investigate the ability of the screeners (and other short tests and measures typically used as ECD proxies), given at three different age ranges, to predict later functioning across the developmental range, which can help to guide the choice of predictive, feasible and easy-to-use readily available instruments for the assessment of very young children’s development in large scale studies in LMICs, for either research purposes or as population-based indicators of ECD." I offered a compromise which the authors were unwilling to accept. I suggested that they not state as an aim that they would "guide the choice of predictive, feasible and easy-to-use readily available instruments …. for research purposes or as population-based indicators…" but rather provide their pros and cons in the Discussion to help guide researchers. At the same time, in the Discussion they should also mention the untested option of the MDAT and the future D-score. They could also inform researchers that the Denver II is now 27 years old and there might be a new version coming out soon. If researchers wanted a "guide to the choice of" good measures for their purpose, they would go to the World Bank Toolkit and its Inventory, not to a single study conducted in Bogota. For all these reasons and those mentioned initially (1.2, 1.3, 1.4), I recommend that the authors not state as an aim that they guide researchers' selection of a measure for research. Their Method is simply too limited to do this properly or convincingly. They can address this in the Discussion.

7. PLOS authors have the option to publish the peer review history of their article (what does this mean?). If published, this will include your full peer review and any attached files.

Reviewer #1: No

---

## [Author Response · Author response to Decision Letter 3]

15 Mar 2020

PONE-D-19-20029R3

Predictive validity in middle childhood of short tests of early childhood development used in large scale studies compared to the Bayley-III, the Family Care Indicators, height-for-age, and stunting: A longitudinal study in Bogota, Colombia

Responses from the authors to the Reviewer’s comments 

We thank Reviewer 1 for reading through our third set of responses. We respond below to the new set of comments. For clarity, we retain the reviewer’s comments and number them to facilitate cross-references to our responses. Any changes to the text in the Manuscript are copied below in italics. Please note that the references to the line numbers where edits have been made correspond to the version of the Manuscript without tracked changes (clean version of the Manuscript). 

Reviewer Comments to the Author

Reviewer #1: PONE-D-19-20029R3

Predictive validity in middle childhood of short tests of early childhood development used in large scale studies compared to the Bayley-III, the Family Care Indicators, height-for-age, and stunting: A longitudinal study in Bogota, Colombia

The authors have made some of the requested revisions and resisted others. I suppose some of the difference is between a statistical preference for drawing conclusions and giving advice based on the statistical analyses, and a preference for interpreting the methods and analyses in terms of child development within the context of an intervention. 

So we can agree to disagree on whether it is better to test a child in his/her home where they have habituated to the "distractions" or to test in a controlled strange environment. Maybe you have to know about the "strange situation" to understand the difference.

We can also agree to disagree on whether Fernald was expressing an expert opinion on the Bayley or simply writing what Pearson Assessment has on their materials.

1. However, my statement that the authors should not state as an aim that they would provide guidance on which short measure to use was not sufficiently addressed. My point was not that you failed to mention the limitations, but that the limitations were sufficiently severe as to reduce the value of your advice on selection of a measure for others' research.

Response: We have omitted the part of the aim that claimed to provide guidance for the choice of instruments for the assessment of very young children’s development in large scale studies. The revised study aim, in lines 126-129 now reads: “The study aimed to investigate the ability of the screeners and other short tests and measures, frequently used to evaluate interventions or measure child development at population level, given at three different age ranges, to predict later functioning in intelligence and school attainment.” 

Similarly, in the Discussion, we have deleted the reference to providing guidance for the choice of tests in lines 496-499, which now read: “Whilst both the Denver-II and the SFII have been found to be sensitive to the impact of cash transfer programs in Nicaragua [12] and Ecuador [13], respectively, further investigation of sensitivity to interventions for all short tests would be helpful.”

2. I understand that internally standardized scores make for good statistical comparisons across measures but they do not provide transparent information about the children's mental development using these measures. This study could be more than an exercise in statistical comparison if you want to provide guidance on selection of measures. The request for actual scores was answered by providing a supplementary table. The Denver scores were provided and I can see that they are aligned with the Bayley's in a ways that are similar to other papers and so are more transparent.

Response: We are pleased that this is now acceptable. 

3. I can understand that other members of your team (e.g. Fernald) and other co-authors might be able to follow your non-standardized instructions to avoid ceiling on screening measures, but for obvious reasons other researchers will/have not been able to. Consequently, while your predictive data might be rigorous, others using these measures do not have the required adaptations to yield such valid data.

Response: We are sorry that the description of the modifications to the ASQ-3 were not clear. It was in fact similar to that used in the previous publication. We have modified the description in lines 174-184 to further clarify it and piloted it with colleagues and hope it is now clear: “As reported earlier [10], all tests were administered following manual instructions except the ASQ-3, which we modified as follows. Given the low literacy levels of some caregivers, caregiver-completed items were administered by interview in order to ensure all mothers understood the questions similarly. Furthermore, the child was tested if the caregiver did not know the answer. In addition, whenever the scale ceiling was reached in the appropriate questionnaire for the age of the child, we added the next three more difficult items from the subsequent questionnaire excluding items that already occurred in the age-appropriate test. This reduced the number of children on the test ceiling by 10.5–15.5% to levels of 1.7–4.8%, depending on the domain, thus increasing the variability in development captured by the test. Similar adaptations have been used elsewhere [11].” 

4. I understand why you did not include the MDAT or refer to the GSED, but that does not eliminate the limitation that you advise people to use the Denver while the MDAT may have been a better test.

Response: We were saying that the Denver-II was the most appropriate out of the tests we examined—not of all available tests. We have made this clearer throughout the text: 

Lines 473-476 in the Discussion: “Over age 18 months, of the tests investigated, the Denver-II appeared to be the best candidate for use at scale, showing the closest predictive ability to the Bayley-III, although it was low-to-moderate, as indicated earlier.” 

Lines 553-554 in the Conclusion: “The SFII from 19-30 months and the Denver-II from 19-42 months were the most feasible and valid short tests of those investigated.”

Also, we were only suggesting that the Denver-II could be used in urban areas in Latin America, as indicated in lines 515-516: “Finally, the above findings could be extrapolated to urban Colombia and possibly urban areas in other Latin American countries.”

Subsequent to the publication of our previous paper on the concurrent validity of these tests, another study has shown that the Denver-II was also appropriate for Brazil so we have referenced this in lines 516-517: “Following our earlier study of these tests’ concurrent validity [10], it was shown that the Denver was also appropriate for Brazil [55].” 

5. The authors left unchanged their aim as: "The study aimed to investigate the ability of the screeners (and other short tests and measures typically used as ECD proxies), given at three different age ranges, to predict later functioning across the developmental range, which can help to guide the choice of predictive, feasible and easy-to-use readily available instruments for the assessment of very young children’s development in large scale studies in LMICs, for either research purposes or as population-based indicators of ECD." I offered a compromise which the authors were unwilling to accept. I suggested that they not state as an aim that they would "guide the choice of predictive, feasible and easy-to-use readily available instruments …. for research purposes or as population-based indicators…" but rather provide their pros and cons in the Discussion to help guide researchers.

Response: Please see our response to Comment 1 above. As explained, we have dropped this aim 

6. At the same time, in the Discussion they should also mention the untested option of the MDAT and the future D-score. 

Response: Thanks for the suggestion. We have now mentioned several tests that could be evaluated in future including the MDAT and the GSED in lines 519-525: “The number of short tests included in the current study was limited due to time and budgetary constraints. Short tests likely to be more suitable for Africa and Asia, such as the Malawi Developmental Assessment Tool (MDAT) [56] could be similarly evaluated. Moreover, several tests are currently under development including population-based and individual-level instruments for children 0-36 months [3,4] using a new approach, the D-score, which summarizes overall development using one interval scale [57]. After further piloting, these may be appropriate for use globally [4]. ” 

7. They could also inform researchers that the Denver II is now 27 years old and there might be a new version coming out soon. 

Response: Since the designer of the test has died, at present, there are no plans for a new version, but the manual and record forms are available for the publisher’s website (http://denverii.com/), so we are not mentioning this.

8. If researchers wanted a "guide to the choice of" good measures for their purpose, they would go to the World Bank Toolkit and its Inventory, not to a single study conducted in Bogota. For all these reasons and those mentioned initially (1.2, 1.3, 1.4), I recommend that the authors not state as an aim that they guide researchers' selection of a measure for research. Their Method is simply too limited to do this properly or convincingly. They can address this in the Discussion.

Response: We have dropped the aim. Please see our response to Comment 1 above. We have also added to the conclusions, the following (lines 559-560): “These findings need to be examined in other regions, when recently developed tests, could be included.”

---

## [Decision Letter · Decision Letter 4]

23 Mar 2020

Predictive validity in middle childhood of short tests of early childhood development used in large scale studies compared to the Bayley-III, the Family Care Indicators, height-for-age, and stunting: A longitudinal study in Bogota, Colombia

PONE-D-19-20029R4

Dear Dr. Rubio-Codina,

We are pleased to inform you that your manuscript has been judged scientifically suitable for publication and will be formally accepted for publication once it complies with all outstanding technical requirements.

With kind regards,

Thach Duc Tran, M.Sc., Ph.D.

Academic Editor

PLOS ONE

Additional Editor Comments (optional):

Reviewers' comments:

Reviewer's Responses to Questions

**Comments to the Author**

1. If the authors have adequately addressed your comments raised in a previous round of review and you feel that this manuscript is now acceptable for publication, you may indicate that here to bypass the “Comments to the Author” section, enter your conflict of interest statement in the “Confidential to Editor” section, and submit your "Accept" recommendation.

Reviewer #1: All comments have been addressed

2. Is the manuscript technically sound, and do the data support the conclusions?

Reviewer #1: (No Response)

3. Has the statistical analysis been performed appropriately and rigorously? 

Reviewer #1: (No Response)

4. Have the authors made all data underlying the findings in their manuscript fully available?

Reviewer #1: (No Response)

5. Is the manuscript presented in an intelligible fashion and written in standard English?

Reviewer #1: (No Response)

6. Review Comments to the Author

Reviewer #1: (No Response)

7. PLOS authors have the option to publish the peer review history of their article (what does this mean?). If published, this will include your full peer review and any attached files.

Reviewer #1: No

---

## [Editor Report · Acceptance letter]

7 Apr 2020

PONE-D-19-20029R4 

Predictive validity in middle childhood of short tests of early childhood development used in large scale studies compared to the Bayley-III, the Family Care Indicators, height-for-age, and stunting: A longitudinal study in Bogota, Colombia 

Dear Dr. Rubio-Codina:

I am pleased to inform you that your manuscript has been deemed suitable for publication in PLOS ONE. Congratulations! Your manuscript is now with our production department. 

With kind regards,

on behalf of

Dr. Thach Duc Tran 

Academic Editor

PLOS ONE